# Learning through experience: Episodic memory representation for cognitive agents

## Abstract

As the demand for intelligent robots and cognitive agents rises, the ability to retain and utilize past experiences through episodic memory has become crucial, especially for social companion robots that rely on previous interactions for task execution. To address this, we introduce Episodic Memory for Cognitive Agents (EMCA), a novel framework that advances knowledge representation by integrating real-world interactions. EMCA enables agents to adapt to complex environments by learning from tasks, interacting with humans, and processing multimodal data—such as speech, vision, and non-verbal cues—without pretraining on specific scenarios. EMCA models episodic memory through a graph-based structure, allowing for incremental storage and retrieval of experiences. Each interaction or event enriches the memory graph, supporting continuous learning and adaptation without extensive retraining. This human-like memory formation optimizes the agent's ability to retrieve relevant information for tasks like localization, planning, and reasoning based on prior experiences. Unlike conventional models relying on temporal markers or recurrent patterns, EMCA encodes data like human memory, allowing reasoning across diverse scenarios regardless of temporal patterns. The framework dynamically builds a memory graph with semantic and temporal connections based on the agent's experiences, promoting flexible temporal reasoning. It also introduces mechanisms for clustering new memories and a dynamic retrieval policy that adjusts based on context or query type, ensuring robustness even in unpredictable scenarios. Empirical tests show EMCA adapts effectively to real-world data, offering reliability and flexibility in dynamic environments.

## 1 Introduction

Episodic memory, introduced by Tulving (1972) Tulving (2002), refers to the recollection of personal experiences anchored to specific times and locations. Unlike semantic memory, which contains general knowledge, episodic memory retains detailed information about events, including temporal, spatial, emotional, and contextual aspects. Tulving's framework organizes episodic memory into components: time, place, characters, and events. Each episode, as shown in Figure 1, is a synthesis of these components tied to a specific time.

Building on this framework, we propose a model where episodic memories are organized as a graph, with Each episode $i$ is represented as a node: $\text{Episode}_i = \{\mathbf{C}_i, \mathbf{T}_i, \mathbf{L}_i, \mathbf{e}_i\}$.

Here, $\mathbf{C}_i$ denotes the characters, $\mathbf{T}_i$ the temporal aspects, $\mathbf{L}_i$ the spatial location, and $\mathbf{e}_i$ the events. The graph is connected by two types of edges: semantic edges $\mathbf{S}(v_i, v_j)$, linking nodes with shared components, and temporal edges $\mathbf{T}(v_i, v_j)$, establishing the sequence of episodes. A dynamic clustering approach is applied to group similar episodes based on temporal and contextual similarities, optimizing retrieval efficiency.

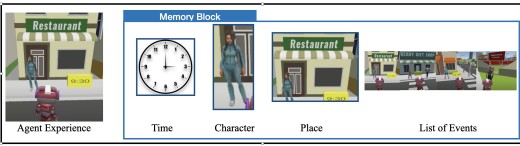

Figure 1: Representation of a single episode (node) in episodic memory, integrating time, character, place, and events.

Our retrieval system supports three query types: "what" (contextual), "when" (temporal), and "where" (spatial), as outlined by Stephen et al., and Holland and Smulders (2011), enabling human-

like memory recall. This is particularly valuable for applications in social companion robotics, aiding elderly or memory-impaired individuals.

For such a cognitive agent it is important that it posesses the ability to recall episodic memories is essential for human cognition, linking personal experiences to specific temporal and spatial contexts. Existing memory models often struggle with continuous, time-series data, which limits their ability to simulate episodic recall effectively. Many of these systems fail to store dialogues as multimodal data, which prevents them from capturing the rich, context-dependent nature of human memory. Additionally, most existing approaches store a single experience as one isolated episode and lack a mechanism for retrieving information across multiple experiences, hindering their ability to integrate knowledge over time. Furthermore, current episodic memory systems are typically restricted to performing a specific, predefined task, limiting their flexibility and adaptability. In contrast, our system is designed to be more versatile, capable of handling a variety of tasks and dynamically adapting to new scenarios, making it far more suited for real-world applications that require memory integration across different contexts and time periods. By integrating multimodal data and temporal information into the episodic memory framework, our model enables experience localization, recommendation, and episodic question answering. It provides a robust foundation for adaptable, scalable systems capable of operating without frequent retraining, applicable to real-world scenarios such as social companion robots and autonomous task planning systems.

**Contributions:**

1. Temporal connections are managed without complex pattern learning, enabling adaptive reasoning and retrieval of subgraphs from past experiences.

2. The system incrementally stores and retrieves episodic memories, dynamically clustering them based on temporal and contextual affinities.

3. A multi-edge graph framework optimizes path traversals for dynamic memory retrieval and personalized recommendations across subjective timescales.

4. A new dataset is introduced to improve episodic memory question answering, enhancing the agent's ability to respond to queries based on past events.

Our model's versatility is demonstrated through comparisons with existing systems that require retraining. It handles various dataset types, including visual, multimodal, and text-based data, and excels in temporal reasoning even without explicit timestamps, addressing complex memory retrieval tasks across diverse applications.

## 2 RELATED WORKS

Episodic memory in multimodal systems has seen significant progress. Xiong et al. Xiong et al. (2016) proposed a memory framework centered on question-answer pairs, later enhanced by Han et al. Han et al. (2019) with transformer-based networks and reinforcement learning, yet constrained by predefined queries, limiting real-world use.

Temporal Graph Networks (TGNs) by Rossi et al. Rossi et al. (2020) facilitate learning on dynamic graphs, finding applications in recommendations and social media. Associative memory models, such as Hopfield networks Ramsauer et al. (2021), allow content-based retrieval but struggle with irregular data. Sarıgun Sarıgün (2023) addresses dynamic temporal graphs, while TempoQR Mavromatis et al. (2021) excels in structured datasets with explicit event timing, unlike the contextual inference required by real-world agents.

Episodic memory is also key in temporal localization for language queries, extending beyond VideoQA Xu et al. (2021) to more complex scenarios. Techniques like 2D-TAN Zhang et al. (2020), VSLNet-L Zhang et al. (2020), and RELER Liu et al. (2022) achieve video content localization but operate independently for each video. Our model advances these by integrating audio-visual and contextual inputs for comprehensive, context-aware outputs.

EMQA Datta et al. (2022) enriches VideoQA through episodic memory for improved responses. Memory-Augmented Neural Networks (MANNs) Santoro et al. (2016), including STM Le et al. (2020), DNC Xiong et al. (2016), and Rehearsal Memory Zhang et al. (2021), are models that make use of memory system which store memory to answer questions from videos ,also Bärmann et al. develop memory graphs for long-term retentionBärmann et al. (2024) in order to verbalize data

for answering questions from prior memory . These models are trained using predefined question-answer pairs and do not store audio or sound data, which makes them less suitable for supporting episodic memory.Anokhin et al. (2024) merges semantic and episodic memory and is similar to human memory but suffers from traversal inefficiencies due to complexity. Edge et al. Edge et al. (2024) enhance retrieval-augmented generation (RAG) with graph indexing in academic domains, and Mavromatis et al. Mavromatis & Karypis (2024) integrate LLMs and GNNs for multi-hop QA, limited by static clusters. Our dynamic multi-edge graph approach adapts to multimodal, time-series data by creating new clusters as the agent gains new experiences, enabling efficient retrieval and enhanced flexibility in unstructured environments.

## 3 METHODOLOGY

EMCA's methodology for collecting and structuring episodic experiences is inspired by cognitive psychology, specifically the 'what', 'where', and 'when' (WWW) components of episodic memory. This approach highlights the agent's ability to independently capture multimodal data—visual and auditory—to build a comprehensive understanding of its environment. As depicted in Figure 2, the system comprises two primary stages: **Experience Memory Collection**, where the agent autonomously compiles and stores experiences in a knowledge base without pretraining, and **Memory Retrieval**, where it recalls relevant experiences for question answering and reasoning. This method enhances adaptability by utilizing prior experiences to inform decision-making. The episodic memory framework is applicable in diverse contexts, such as memory localization and experience-based recommendation. Detailed methodology is given in Appendix B

### 3.1 PROCESSING OF AUDIO DATA IN EPISODIC MEMORY

Audio data, including dialogues and acoustics, is crucial for constructing episodic memory. Dialogue's provide linguistic and contextual information, while acoustics capture environmental and emotional cues. These elements are integrated as $A(t) = D(t) + C(t)$, where $A(t)$ is the total audio data at time $t$, with $D(t)$ representing the dialog and $C(t)$ representing acoustics.

#### 3.1.1 EXTRACTION OF ACOUSTIC DATA USING MEL SPECTROGRAMS

Acoustic data is transformed into Mel spectrograms, which emphasize perceptually relevant frequencies. The Mel spectrogram $M(t, f)$ is computed as $M(t, f) = \log \left( \sum_k |X(t, k)|^2 \cdot H(f, k) \right)$. where $X(t, k)$ is the magnitude of the STFT at time $t$ and frequency $k$, and $H(f, k)$ is the Mel filter bank mapping linear frequencies to the Mel scale.

#### 3.1.2 EXTRACTION OF VERBAL CUES FROM AUDIO DATA

Verbal cues are extracted by applying the Short-Time Fourier Transform (STFT) and converting the spectrum to the Mel scale as $M(f) = 2595 \log_{10} \left( 1 + \frac{f}{700} \right)$. The Mel spectrogram is then derived as $\text{MelSpec}(m, t) = \log \left( \sum_{f_{\text{low}}}^{f_{\text{high}}} |S(f, t)|^2 M(f) + \epsilon \right)$, where $\epsilon$ is a small constant to prevent issues with the logarithm.

#### 3.1.3 INTEGRATION OF ACOUSTIC AND VERBAL FEATURES

The final audio representation integrates acoustic features with transcribed dialogue: $T_{\text{audio}}(t) = T_{\text{acoustics}}(t) + T_{\text{dialogue}}(t)$.capturing both tonal properties and linguistic meaning. By applying tagging techniques to the processed audio data, we extract and associate place, character, and time information. These features are then saved as an embedding, forming a unified representation for episodic memory.

### 3.2 PROCESSING OF VISUAL DATA IN EPISODIC MEMORY

Visual data processing starts by transforming each frame $F_i$ into a tensor and extracting global and local features. The scene representation is then obtained as $V_{\text{scene}} = \frac{1}{N} \sum_{i=1}^{N} V_{\text{embed}}(F_i)$.

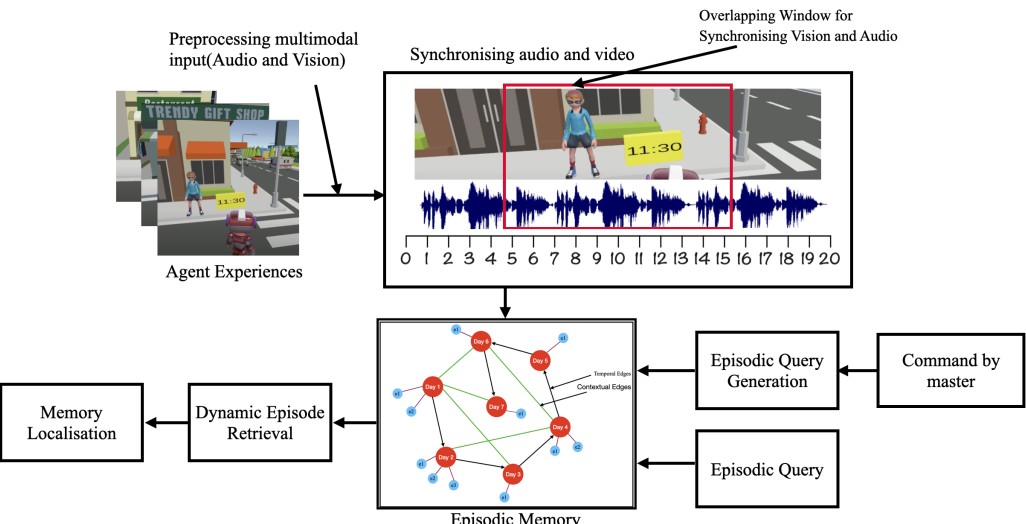

Figure 2: Methodology:The agent collects multimodal experiences through vision and audio, which are then synchronized by aligning them within specific time windows. The fused episode is stored in episodic memory which is represented in the form of a graph that, capturing details such as characters, time, place, and events and has both spatial and temporal edges. Upon receiving a query, the model retrieves the relevant episode or set of episodes from memory and generates an appropriate response.

where $N$ represents the number of frames in the scene. To extract $V_{\text{time}}$ and $V_{\text{place}}$, a convolution network processes the feature map $F_{\text{feature}}(x, y)$, generating probability maps for the text center lines (TCL) and text regions (TR): $\begin{pmatrix} P_{\text{TCL}}(x, y) \\ P_{\text{TR}}(x, y) \end{pmatrix} = \sigma \left( \begin{pmatrix} W_{\text{TCL}} \\ W_{\text{TR}} \end{pmatrix} \cdot F_{\text{feature}}(x, y) \right)$. Thresholding is applied to filter relevant text regions using the condition

$$P_{\text{filtered}} = \{(x, y) \mid P_{\text{TCL}}(x, y) \geq T_{\text{TCL}} \text{ and } P_{\text{TR}}(x, y) \geq T_{\text{TR}}\}.$$

Recognized text is processed with a softmax layer for classification: $\hat{y}_t = \text{Softmax}(W \cdot h_t + b)$. from which $V_{\text{time}}$ and $V_{\text{place}}$ are derived. Character information ($V_{\text{character}}$) is extracted by associating text and visual features through techniques such as Named Entity Recognition (NER) or vision-language embeddings. Together, $V_{\text{scene}}$, $V_{\text{time}}$, $V_{\text{place}}$, and $V_{\text{character}}$ form a comprehensive multimodal representation that captures contextual details for reasoning or retrieval tasks.

### 3.2.1 MERGING AND SYNCHRONIZING DATA

In the final stage, processed audio and visual data are synchronized to a common timestamp, forming a unified representation: $\mathbf{T_M} = (\mathbf{T}_{\text{audio}}, \mathbf{T}_{\text{visual}})$. This ensures temporal alignment between the modalities. The audio and visual embeddings are then concatenated into a joint multimodal embedding: $\mathbf{E}_{\text{combined}} = \mathbf{E}_{\text{audio}} \oplus \mathbf{E}_{\text{visual}}$, which is stored in episodic memory, with each node representing key experience aspects such as time, location, characters, and events.

This integrated embedding enhances memory recall and event-based analysis. Following concatenation, the embedding is encoded to capture entities like place, time, and characters for each episode. The methodology for capturing these entities is supported by prior studies, as shown in the appendix.

### 3.3 EPISODIC MEMORY REPRESENTATION

Each node in the episodic memory represents a day, with subnodes capturing the activities of that day. The main node summarizes the day's events, while subnodes encode specific event details. Joint embeddings, integrating place, character, and event information, are stored at both the event and day levels. This hierarchical structure enables efficient encoding and retrieval of both events and their details. There are two types of edges: contextual edges, which con-

nect nodes based on similarities in place, character, and events, and temporal edges, which link nodes temporally to reflect the sequence of events. Figure 3 shows an episodic memory graph.

### 3.3.1 CLUSTER DEFINITIONS

When an agent receives an episodic experience, it assigns the experience to multiple clusters based on location, events, and characters: Location Cluster $\mathbf{C}_l$, Character Cluster $\mathbf{C}_c$, and Event Cluster $\mathbf{C}_e$. Each entity within a cluster is uniquely identified, facilitating efficient memory organization and retrieval in alignment with the hierarchical structure of episodic memory, where nodes represent days and subnodes capture activities and events.

### 3.3.2 DYNAMIC CLUSTERING

When an agent gets an experience Text and visual embeddings are utilized to restore feature details and improve decision-making. Spatial and character embeddings are derived from similarities between locations and characters across episodes. For each new episode $E_n$, these embeddings are evaluated and integrated into existing clusters using an attention mechanism: $\text{attention}_X(E_n, E_i) = \frac{\mathbf{V}_{X_n} \cdot \mathbf{V}_{X_i}}{\|\mathbf{V}_{X_n}\| \|\mathbf{V}_{X_i}\|}$, where $X \in \{\text{location}, \text{char}\}$. This dynamic clustering process organizes memory based on spatial, character, and event similarities. New episodes that do not fit into existing clusters create new identifiers, ensuring continuous cluster expansion.

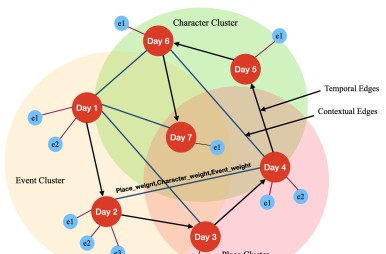

Figure 3: The diagram illustrates an episodic memory graph that organizes actions and key events hierarchically. Nodes represent memories enriched with textual, visual, and acoustic data, including Mel spectrograms. Connections reflect shared locations, characters, and events, aiding efficient recall and analysis. Clusters are formed using attention mechanisms on embeddings related to shared characters, places, and events, enabling effective grouping of related episodes.

Event clusters are formed by modeling coherence among consecutive utterances within dialogues, with each dialogue $D = (u_1, u_2, \ldots, u_{|D|})$ consisting of $|D|$ utterances. Instead of relying on expert annotations, we infer event structures by assuming that utterances within the same event exhibit higher coherence than those spanning multiple events. A contrastive learning objective is used to maximize coherence within event-related snippets while minimizing coherence across different events. The contrastive coherence loss is defined as: $L_{\text{contrastive}} = \sum_i [\log p(\text{positive}_i) - \log p(\text{negative}_i)]$, where positive and negative examples are selected based on coherence metrics such as ROUGE score, facilitating unsupervised event clustering. This approach allows for the segmentation of dialogues into relevant event clusters, improving the system's ability to retrieve and reason about past experiences, enhancing context-aware decision-making. Once the new episode $E_n$ is assigned to one or more clusters, the cluster definitions are updated as follows: $\mathbf{C}_X^j = \mathbf{C}_X^j \cup \{E_n\}$ (10), where $X$ refers to location, character, or event. If $E_n$ does not fit into any existing cluster, new identifiers are created in the cluster: $\mathbf{C}_X^{\text{new}} = \{E_n\}$ (11), where $X \in \{\text{location}, \text{char}, \text{event}\}$.

### 3.3.3 EDGE CONNECTION

Contextual edges between episodes are formed by analyzing shared clusters based on common characters, locations, or events. The similarity between two episodes $E_1$ and $E_2$ is determined by the number of overlapping clusters, with an edge created if two or more clusters are shared. The edge weight is computed by combining the location, character, and event similarities: $\text{Weight}(E(N_1, N_2)) = \mathbf{L}(N_1, N_2) \| \mathbf{C}(N_1, N_2) \| \mathbf{E}(N_1, N_2)$, where $\mathbf{L}(N_1, N_2)$, $\mathbf{C}(N_1, N_2)$, and $\mathbf{E}(N_1, N_2)$ represent location, character, and event similarities, respectively. The primary connection between episodes is established by the maximum similarity across these features: $\text{Link}(E(N_1, N_2)) = \arg\max(\{\mathbf{L}(N_1, N_2), \mathbf{C}(N_1, N_2), \mathbf{E}(N_1, N_2)\})$.

**Temporal Edges** are defined by the time relationship between consecutive episodes. The temporal connection between episodes $E_{t-1}$ and $E_t$ is represented as $\mathbf{T}_{\text{edge}}(E_{t-1}, E_t) = 1$. Temporal edges help handle missing timestamps, as no traditional statistical methods are used to estimate them. Instead, each episode is treated as representing a distinct day, and the temporal indexer is updated

based on visual or dialogue-based date capture. This ensures that the timestamp of any episode is adjusted accordingly, with subsequent episodes indexed relative to this structure. For instance, the "before" node is one day prior, and the "after" node is one day later. Even without external timestamps, the model understands the temporal order of episodes via temporal edges, maintaining consistency in the passage of time as the agent processes the information.

### 3.3.4 Dynamic Episode Retrieval

Figure 4 illustrates dynamic edge traversal for retrieving relevant memories using character, location, event, and temporal weights. The agent classifies the query $q$ using language models to determine whether it is a "what", "when", or "where" query. "What" queries focus on events, "when" queries on temporal details, and "where" queries on locations. This classification allows the agent to assign the appropriate context and efficiently retrieve relevant memories. Temporal entities (e.g., weeks, months, years) are processed by subtracting fixed intervals from the current date: $7n$ days for weeks, $30m$ days for months, and $365y$ days for years, where $n$, $m$, and $y$ are positive integers.

The similarity score between the query $q$ and a set $D_u$ of memory entries is given by $S_s = \sum_{e \in D_u} \frac{q \cdot e}{\|q\| \|e\|}$. For each neighbor $v$ of node $u$, the weight $W_{uv}$ is computed as $W_{uv} = \sum w(u, v)$. If $W_{uv} > \theta$, the query set is updated as $Q \leftarrow Q \cup (v, W_{uv})$.

Temporal edges $T_{\text{edge}}(E_{t-1}, E_t)$ maintain the sequence of events without re-evaluating the entire graph, with date and time represented as separate nodes for indexing. Explicit timestamps in queries map directly to temporal nodes, while contextual queries traverse the graph based on query type: event weights for "what," temporal weights for "when," and location weights for "where" queries. This framework (Figure 4) efficiently retrieves memory clusters, providing task-specific outputs: free-form text for episodic QA and recommendations, memory nodes for experience localization, and goal directives for RL agents (AppendixE.4).Additional details of dynamic node retrieval are given in Appendix B.5

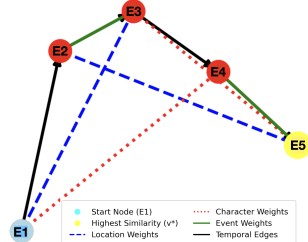

Figure 4: Dynamic edge traversal. Episodic graph representation of event relationships with different edge weights. The start node (E1) is highlighted in cyan, and the node with the highest similarity (E5) is in yellow. Edge colors and styles denote distinct relationship types: temporal edges (solid black), location weights (dashed blue), character weights (dotted red), and event weights (solid green). The legend clarifies the significance of the nodes and edges in the graph.

### 3.4 Impact of Node Removal on Connections

When an episode, such as $E_2$, is removed from the dynamic graph as part of the forgetting mechanism, its associated semantic links are eliminated, disrupting connections. However, the temporal edges are readjusted, allowing $E_3$ to connect to the next relevant node, preserving the chronological structure. Consequently, $E_3$ updates its temporal link to $E_1$, maintaining event sequence continuity. This selective removal of semantic associations, while preserving temporal coherence, eliminates the need for a full graph reevaluation. The remaining nodes retain contextual relevance, enabling the agent to reason based on prior experiences. Thus, implementing a forgetting mechanism does not compromise the integrity of the episodic memory graph. We have tested node removal mechanism by using frequency based weight decay to prune nodes as described in Appendix C. Results after node pruning are given in E.2.2

## 4 Recommendation Using EMCA

When assisting individuals with memory impairments, a cognitive agent utilizes past interactions to provide personalized support by extracting a **personalized cluster** $C_p$, representing the relevant memory subgraph for the individual. This subgraph consists of episodes, actions, and events related to the person:

$$C_p = \{G_p \mid \text{episodes associated with person } p\}$$

where $G_p = (V_p, E_p)$ includes episodes $V_p$ and edges $E_p$, preserving temporal and contextual information.

The agent then identifies **event clusters** $C_e$, representing actions performed by the individual over time:

$$C_e = \{E_e \mid \text{events associated with episodes in } G_p\}$$

To provide recommendations, the agent identifies the most recent event sequence $\mathcal{S}_t = (S_1, S_2, \ldots, S_t)$ and searches the memory subgraph for a matching sequence, determining the next action $S_{t+1} = \text{argmax}_{S \in G_p} \mathbb{I}(\mathcal{S}_t \subset \mathcal{S})$.

If multiple matches occur, the most frequent next event is selected: $S_{t+1} = \text{mode}(\{S_{t+1} \in \mathcal{S} \mid \mathcal{S}_t \subset \mathcal{S}, \mathcal{S} \in G_p\})$ This method allows the agent to generate recommendations even without recurring patterns, leveraging past episodic data to support individuals with memory impairments. Temporal graph networks and Hopfield memory networks rely on pattern recognition to predict future actions. In contrast, our approach allows the agent to make predictions and provide recommendations without depending on explicit pattern-based mechanisms, offering more flexible and adaptive support for individuals with memory impairments.

## 5 DATASET

We propose a comprehensive dataset framework designed to evaluate and enhance episodic memory systems in artificial agents. This framework integrates multiple datasets, including a custom set of episodic questions based on the TV series The Big Bang Theory, spanning all nine seasons (181 episodes). The aim is to assess memory recall and narrative understanding in complex scenarios.

We introduce the Agent Dataset, a 10-episode time-series dataset created in Unity3D, where a virtual agent performs tasks and interacts with characters in realistic environments, simulating the role of companion robots. This dataset emphasizes the importance of multi-sensory inputs and task execution, challenging the agent to process and integrate information from dialogues and visual cues to maintain task order and achieve context-driven objectives.

Additionally, we adapted the **Ego4D dataset**, restructuring its activity sequences into simulated chronological episodes to address the original absence of time-series data—portraying an agent performing a series of activities over 30 days. We also combined group activity videos designed for active speaker recognition. This transformation enables episodic queries such as "Where did I place the agricultural tool on the last day of farming?", enhancing the ability to localize and retrieve temporal experiences effectively.

Together with the **PerLTQA** Du et al. (2024) and **LLQA** Dolan & Brockett (2005) datasets, which test essential episodic memory dimensions—**"what"** (context), **"when"** (time), and **"where"** (place)—this framework forms a robust benchmark for evaluating advanced episodic memory capabilities in AI systems.

**Data Annotation**: The data was carefully annotated to tag scene information and identify characters in dialogues, ensuring that the model could recognize character presence and understand related events. This included explicitly tagging scene details for location identification and differentiating characters present in the scene versus those mentioned. Events within dialogues were also meticulously annotated to capture key details, facilitating effective memory representation beyond simple summaries. Capturing these essential details is crucial for episodic memory tasks, as it allows the agent to recall past experiences accurately. Each episode was annotated with 10 what, when, and where questions.

**Data Statistics**: The dataset includes a distribution of question types: temporal questions make up 24%, spatial questions 38%, contextual questions 18%, multimodal questions (integrating visual and auditory information) 10%, and dialogue-based questions 10%. These detailed annotations enable the model to handle temporal, spatial, and contextual elements, as well as multimodal inputs, ensuring comprehensive event recognition and effective interaction.

## 6 EXPERIMENTS AND EVALUATION

We evaluate our episodic memory cognitive agent (EMCA) on downstream tasks such as episodic memory question answering, benchmarking it against state-of-the-art graph-based and memory models. The agent's performance is tested on memory localization and multimodal memory-based

visual QA. An ablation study compares clustered and non-clustered approaches while assessing the impact of modality removal on episodic memory. These experiments leverage the episodic memory dataset, with EMCA implemented using Whisper, CLIP, and BERT backbones. Additional details are provided in the AppendixD.

### 6.1 EVALUATION METRICS

The performance was assessed using *recall accuracy* for episodic memory question answering (QA), defined as Episodic Recall $= \frac{\text{Number of Correctly Answered Questions}}{\text{Total Questions}}$. Additionally, the mean Intersection over Union (mIOU) score was employed to evaluate episodic memory localization.

### 6.2 COMPARISON WITH SOTA GRAPH MODELS AND EPISODIC MEMORY QA MODELS

We conducted experiments comparing our approach with state-of-the-art models, including EMR, GraphRag, GNN Rag, TempoQA, and Arigraph, for retrieving relevant information from datasets based on episodic questions. The table below presents a comparative analysis across various datasets, assessing contextual, temporal, spatial, and overall performance metrics.

EMR and Dynamic MemQA are pioneering approaches for episodic memory in multimodal QA, while Arigraph integrates semantic and episodic memory for human-like recall, and TempoQA manages temporal data for time-sensitive event comprehension. GraphRag and GNN Rag utilize graph-based memory and retrieval-augmented generation to handle complex data structures. These models were selected to benchmark EMCA against diverse memory models, graph-based RAG methods, and graph structures, including knowledge graphs and temporal graphs. Figure presents the time taken (in seconds) by different methods for various datasets, including Big Bang Theory, PerLTQA, Agent, and LLQA. Our

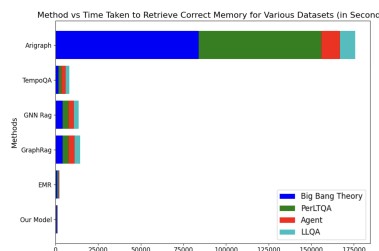

Figure 5: Comparison of Retrieval Times for Various Methods: The chart displays the time taken (in seconds) to retrieve correct memories for four datasets: blue for Big Bang Theory, green for PerLTQA, red for Agent, and cyan for LLQA. The stacked bars represent the cumulative retrieval time for each method

model demonstrates significantly faster retrieval times compared to other approaches, particularly when contrasted with methods like Arigraph and TempoQA, which are slower due to their reliance on more computationally intensive processes.

Table 1 demonstrates EMCA's superior performance across four datasets, highlighting its advanced multimodal capabilities and efficient storage and retrieval mechanisms. By integrating textual, visual, and acoustic data, EMCA handles diverse queries effectively, achieving high recall accuracy in contextual, temporal, and spatial questions.

EMCA's hierarchical clustering organizes episodic memory by characters, locations, and events, enabling precise retrieval and reducing noise. This dynamic approach outperforms static graph models like GraphRag and GNN Rag, which lack adaptability. EMCA's ability to integrate multimodal data and retrieve targeted memories makes it uniquely suited for episodic memory-based question answering. Additional results with memory models are shown in Appendix E.1.

### 6.3 EPISODIC MEMORY LOCALIZATION

We evaluated our model's performance in episodic memory localization using time-series-based questions. Table 2 compares our results with state-of-the-art (SOTA) models recognized for their effectiveness in multimodal data handling within the Ego4D dataset. Metrics include IOU@0.3 with Recall@1 (R@1), IOU@0.5 with Recall@5 (R@5), and mean IOU (mIOU).

While SOTA models excel in multimodal data tasks, they struggle with video localization using time-series data, which is critical for maintaining accurate, context-aware recall in episodic memory systems. By saving data as a time series, our approach ensures precise temporal alignment,

| Dataset | Method | Contextual | Temporal | Spatial | Total |
|---|---|---|---|---|---|
| **Big Bang Theory** | **Ours** | **75** | **80** | **76** | **78** |
| | Dynamic Memory QA | 5 | 8 | 7.5 | 10 |
| | EMR | 14 | 10 | 13 | 15 |
| | GraphRag | 32 | 30 | 30 | 30 |
| | GNN Rag | 25 | 24 | 23 | 27 |
| | TempoQA | 21 | 20 | 19 | 25 |
| | Arigraph | 25 | 24 | 25 | 26 |
| **Perltqa** | **Ours** | **90** | **95** | **89** | **90** |
| | Dynamic Memory QA | 30 | 15 | 21 | 40 |
| | EMR | 31 | 20 | 23 | 45 |
| | GraphRag | 35 | 40 | 42 | 38 |
| | GNN Rag | 60 | 51 | 57 | 55 |
| | TempoQA | 45 | 50 | 51 | 52 |
| | Arigraph | 55 | 75 | 71 | 79 |
| **Agent Dataset** | **Ours** | **77** | **90** | **90** | **86** |
| | Dynamic Memory QA | 33 | 30 | 31 | 37 |
| | EMR | 36 | 32 | 33 | 40 |
| | GraphRag | 31 | 20 | 29 | 27 |
| | GNN Rag | 50 | 51 | 55 | 53 |
| | TempoQA | 40 | 45 | 31 | 37 |
| | Arigraph | 52 | 53 | 53 | 51 |
| **LLQA** | **Ours** | **86** | **85** | **86** | **86** |
| | Dynamic Memory QA | 10 | 20 | 15 | 20 |
| | EMR | 15 | 23 | 20 | 23 |
| | GraphRag | 24 | 23 | 22 | 21 |
| | GNN Rag | 49 | 45 | 40 | 46 |
| | TempoQA | 42 | 41 | 45 | 45 |
| | Arigraph | 50 | 51 | 42 | 52 |

Table 1: Comparison of Recall Accuracy for Different Question Types Across Datasets

| Method | IOU = 0.3 R@1 | IOU = 0.5 R@5 | mIOU |
|---|---|---|---|
| **2D-TAN** | 4.32 | 2.60 | 5.62 |
| **VSLNet** | 8.09 | 7.03 | 7.65 |
| **CONE** | 10.55 | 7.54 | 9.04 |
| **RELER** | 12.89 | 8.14 | 10.51 |
| **SPOTEM** | 18.13 | 13.43 | 15.78 |
| **Ours** | **26.46** | **25.5** | **25.98** |

Table 2: Performance comparison on episodic memory localization.

minimizes outdated recall, and enhances context comprehension. Furthermore, integrating visual and dialogue modalities provides a comprehensive understanding of interactions and events, significantly improving episodic memory localization. Visualization of Episodic memory localization is as given in Appendix E.3

## 6.4 ABLATION STUDIES

We validated our EMCA approach by systematically removing different modalities from the architecture: first the visual module, then the speech module, and finally the music module. Performance was evaluated on videos from The Big Bang Theory and the Agent Dataset by posing episodic questions. The episodic recall capacity, as detailed in 6.2, was assessed based on these crafted episodic questions. Additionally, we measured retrieval time as the number of episodes increased.

| Modalities | Big Bang Theory | Agent Dataset |
|---|---|---|
| **Full** | 78 | 86 |
| **No Vision** | 16 | 15 |
| **No Acoustics** | 65 | 40 |
| **No Dialogues** | 36 | 20 |

Table 3: Comparison of Modalities.

| Method | Retrieval Time (Average) |
|---|---|
| **Dynamic Traversal** | 5.6 |
| **BFS** | 8.9 |
| **DFS** | 11 |
| **DFS + BFS** | 7.75 |

Table 4: Graph Traversal Retrieval Times.

| Number of Episodes | Retrieval Time (ms) |
|---|---|
| 10 | 9.11 |
| 50 | 11.0 |
| 100 | 11.5 |
| 181 | 12.0 |

Table 5: Retrieval Times vs. Episodes.

As shown in Table 3, retaining all modalities is essential for agents operating in multimodal environments to make informed decisions and reason effectively based on past events.

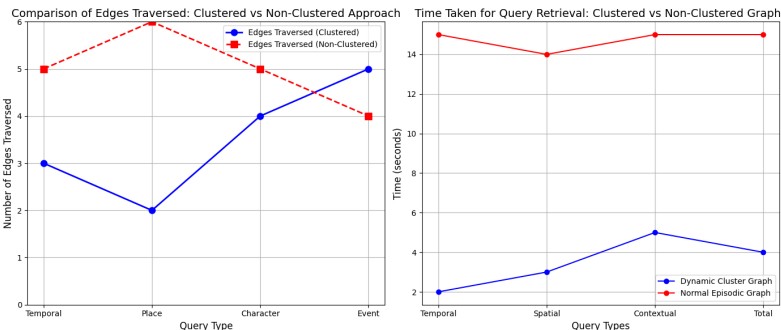

Figure 6: Comparison of the Number of Edges Traversed and Retrieval Time (ms) by Query Type for Clustered and Non-Clustered Approaches

**Assessment of Retrieval Time Across Traversal Methods**

We evaluated the time complexity of our dynamic graph traversal method against traditional techniques, summarized in Table 4. The dynamic traversal method demonstrated the lowest average retrieval time, outperforming BFS, DFS, and their combined approach. When explicit timestamps are provided, queries are treated as temporal, leading to faster retrieval compared to event-based queries.

**Comparing clustered and non-clustered graphs in terms of number of path traversals required and time taken to retrieval query** As shown in Figure 6, the dynamic clustered graph significantly reduces query retrieval time and the number of edges traversed compared to the non-clustered graph, demonstrating the efficiency of clustering in optimizing graph traversal. Dynamic edges and clusters enhance adaptability, improving performance across various query types. Table 5 shows a modest increase in retrieval time as the number of episodes grows, highlighting the clustering mechanism's role in keeping retrieval times low. These results emphasize the scalability and efficiency of our method. Replacing an episodic memory graph with a knowledge graph introduces significant complexity, especially when incorporating temporal features. For instance, an episodic graph represents three days of experiences with just three nodes, one per day, while a temporal knowledge graph may require 15–20 nodes and edges, drastically increasing structural complexity. This added intricacy hampers memory retrieval and action prediction, making them less efficient than the streamlined architecture of episodic graphs. In visual domains, where patterns and regularities are prevalent, knowledge graphs excel by leveraging these consistencies. However, in dynamic scenarios like conversations, interactions are often unique, necessitating frequent creation of new relations. This dynamic evolution further complicates the knowledge graph and reduces practical efficiency. Additional results, including ablation studies and node pruning effects, are detailed in Appendices E.2.1 and E.2..

# 7 CONCLUSION, LIMITATIONS, AND SOCIAL IMPACT

The EMCA system is designed to understand and process temporal timescales in a manner similar to human cognition. It dynamically organizes events along subjective timescales, allowing it to track and retrieve memories based on the relative importance of events rather than fixed timestamps. This enables the system to adapt to varying time-frames, understanding how past experiences may influence present contexts. By mimicking human-like temporal reasoning, EMCA can handle queries about "when" events occurred in both short-term and long-term memory, adjusting its responses based on the perceived significance of past events, much like human memory recalls important moments more vividly than routine occurrences.

**Social Impact:** The Episodic Memory Cognitive Agent has the potential to serve as a valuable social agent, particularly for individuals with memory-related disorders. However, without robust data protection mechanisms, there is a risk of data breaches, which could have serious privacy implications.

**Limitations:** Currently, EMCA lacks a forgetting mechanism and the ability to identify key events, as humans do, based on factors such as surprise, novelty, or emotional significance. These aspects will be addressed in future work.

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

## A    APPENDIX

The appendix provides a comprehensive overview of the research, including a detailed methodology, the forgetting mechanism, implementation specifics, and additional results. It explains the approach used in the study, covering data preprocessing, model architecture, and the integration of episodic and knowledge graph representations. The section on the forgetting mechanism discusses the strategies employed to manage memory capacity, ensuring the system efficiently retains relevant information. Implementation details include the experimental setup, hardware and software configurations, dataset preparation, training procedures, and evaluation metrics used. Furthermore, the appendix presents additional results, including extended analysis of model performance under different conditions, comparisons across multiple datasets, and visualizations of graph structures, offering a deeper understanding of the approach and validating its generalization.

## B    DETAILED METHODOLOGY

EMCA's methodology for collecting episodic experiences in robotic cognition draws inspiration from the human brain's mechanisms for encoding sensory information, particularly the distinct roles of the occipital and temporal lobes. In human cognition, the occipital lobe processes visual stimuli, while the temporal lobe is responsible for auditory information. Despite these processes occurring in specialized regions, the brain synchronizes these sensory inputs within a unified temporal framework, enabling the formation of cohesive and contextually rich memories. EMCA replicates this principle by employing separate pipelines for processing visual data (e.g., spatial and object recognition) and auditory data (e.g., speech and environmental sounds), which are then temporally aligned to construct a coherent representation of the agent's environment.

The system comprises two core stages: **Experience Memory Collection** and **Memory Retrieval**. During Experience Memory Collection, the agent autonomously gathers and stores multimodal experiences in a structured knowledge base without requiring prior training. In the Memory Retrieval phase, the agent leverages these stored experiences to answer queries, contextualize new events, and reason effectively based on past interactions. By integrating visual and auditory modalities within the same temporal window, EMCA achieves a sophisticated level of synchronization and contextual understanding, akin to human episodic memory.

This approach enhances the adaptability and decision-making capabilities of the agent by enabling it to draw upon prior experiences. Furthermore, the episodic memory system facilitates advanced applications such as **memory localization** associating specific memories with spatial and temporal contexts—and **personalized recommendations** based on historical data. The integration of biologically inspired memory encoding principles with robotic cognition underscores EMCA's potential for advancing human-like reasoning in artificial systems.

### B.1    PROCESSING OF AUDIO DATA

### B.2    PROCESSING OF AUDIO DATA IN EPISODIC MEMORY

Audio data is a critical component in constructing episodic memory, comprising dialogues and acoustics. Dialogues provide semantic information, capturing the exchange of language, intentions, and contextual meaning, while acoustics contribute environmental and emotional cues, such as tone, pitch, and ambient sounds. Together, these elements enable a comprehensive understanding of an episode, as the semantic content of dialogues combines with the situational context offered by acoustics. The integration of these elements can be represented as:

$$A(t) = D(t) + C(t),$$

where $A(t)$ is the audio data at time $t$, $D(t)$ represents dialogues, and $C(t)$ denotes acoustics. This unified representation reflects the complementary roles of both components in capturing the richness of episodic experiences. By processing and encoding dialogues and acoustics simultaneously within the same temporal window, the system ensures that both the linguistic and environmental aspects of an event are preserved, facilitating accurate retrieval and reasoning. This approach mirrors human

cognitive processes, where the brain's temporal lobe processes auditory signals and integrates them with contextual understanding, thus enhancing the episodic memory system's fidelity and utility.

### B.2.1 EXTRACTION OF ACOUSTIC DATA USING MEL SPECTROGRAMS

Acoustic data plays a crucial role in episodic memory, capturing non-verbal and environmental auditory cues that enhance contextual understanding. To process acoustic data, Mel spectrograms are employed, which provide a time-frequency representation of audio signals while emphasizing perceptually relevant frequencies. The process begins by segmenting the audio signal into overlapping frames, followed by applying the Short-Time Fourier Transform (STFT) to obtain the frequency spectrum. The resulting spectrum is then mapped to the Mel scale, a scale that approximates the human auditory perception of frequency. This transformation allows for the focus on the frequency range that the human ear is most sensitive to, providing a more relevant representation of the acoustic environment.

The Mel spectrogram $M(t, f)$ is computed using the following formula:

$$M(t, f) = \log \left( \sum_k |X(t, k)|^2 \cdot H(f, k) \right), \tag{1}$$

where $X(t, k)$ represents the magnitude of the STFT at time $t$ and frequency $k$, and $H(f, k)$ is the Mel filter bank that maps the linear frequency $k$ to the Mel scale frequency $f$. The resulting $M(t, f)$ represents the log-scaled Mel spectrogram at time $t$ and Mel frequency $f$.

This approach of transforming the audio signal into a Mel spectrogram allows the system to capture both the temporal and frequency domain features of the audio. By encoding these features, including tone and environmental sounds, the system enhances its ability to understand the acoustic aspects of an episode. This is analogous to how the human brain integrates auditory information with situational contexts to form a cohesive episodic memory. We use the Mel spectrogram for its low-level features, which are essential for developing effective policies in future reinforcement learning (RL) agents, enabling them to better interpret and interact with their auditory environments.

### B.2.2 EXTRACTION OF VERBAL CUES FROM AUDIO DATA

Verbal cues, integral to understanding dialogues and interactions within episodic memory, are extracted from the audio signal by processing the speech content. The extraction begins with the computation of the Short-Time Fourier Transform (STFT) to capture the frequency and time-domain characteristics of the audio. These raw spectral features are then mapped to the Mel scale, aligning with the auditory processing capabilities of the human ear, which focuses more acutely on certain frequency ranges. The formula for this spectral processing is as follows:

$$S(f, t) = \sum_{n=0}^{N-1} s_i(n) w(n - t) e^{-j2\pi fn/N} \tag{2}$$

where $S(f, t)$ represents the frequency-domain representation of the signal, with $s_i(n)$ being the signal at time step $n$, $w(n - t)$ the windowing function, and $e^{-j2\pi fn/N}$ the frequency component.

Subsequently, the power spectrogram $S(f, t)$ is converted to the Mel scale, a logarithmic transformation designed to better reflect the frequency response of the human auditory system:

$$M(f) = 2595 \log_{10} \left( 1 + \frac{f}{700} \right) \tag{3}$$

This conversion ensures that the low frequencies, which are more perceptible to the human ear, are more heavily weighted, reflecting natural auditory attention mechanisms.

The final Mel spectrogram is derived by aggregating the energy within the frequency bands that correspond to the Mel scale:

$$MelSpec(m,t) = \log \left( \sum_{f_{low}}^{f_{high}} |S(f,t)|^2 M(f) + \epsilon \right) \tag{4}$$

where $f_{low}$ and $f_{high}$ are the frequency bounds, and $\epsilon$ is a small constant to avoid computational issues with log of zero.

For further processing, the audio signal $A(t)$ is resampled to a standard rate of 16 kHz and divided into overlapping windows of 25 ms. An 80-channel log-magnitude Mel spectrogram $S_{Mel}(t,f)$ is then computed as follows:

$$S_{Mel}(t,f) = \log \left( \sum_{f_{low}}^{f_{high}} |S(t,f)|^2 \cdot M(f) \right) \tag{5}$$

To normalize the extracted Mel spectrogram, we apply a z-score normalization to ensure that the features have zero mean and unit variance:

$$S_{norm}(t,f) = \frac{S_{Mel}(t,f) - \mu}{\sigma} \tag{6}$$

where $\mu$ and $\sigma$ represent the global mean and standard deviation of the Mel spectrogram features across the entire dataset.

Once normalized, the Mel spectrogram is passed through a series of convolutional layers with GELU activations, which enable the network to extract high-level patterns from the spectrogram. The GELU activation function is defined as:

$$GELU(x) = x \cdot \Phi(x) \tag{7}$$

where $\Phi(x)$ is the cumulative distribution function of the standard normal distribution, providing a smooth, differentiable non-linearity that accelerates learning.

Finally, a Bidirectional Pre-trained Transformer (BPT) model transcribes the processed acoustic features into text, effectively integrating verbal cues from transcribed dialogues. This process allows for a more accurate and contextual understanding of verbal interactions within the episodic memory framework, enabling the system to recall and reason based on both acoustic and linguistic data.

### B.2.3    FINAL REPRESENTATION OF AUDIO DATA

For the final representation of audio input, it is essential to merge both acoustic and dialogue data into a unified form. Acoustic data, typically derived from the Mel spectrogram, captures the spectral features of speech, such as tone, pitch, and rhythm, while the dialogue data consists of the transcribed speech content. Both data types offer valuable, complementary information, and their combination enables a more complete understanding of the audio signal.

The process involves aligning the acoustic features and the transcribed dialogue to the same temporal framework, ensuring that both data sources correspond to the same timestamps. This temporal synchronization results in the final, combined audio representation, which can be expressed as:

$$T_{\text{audio}}(t) = T_{\text{acoustics}}(t) + T_{\text{dialogue}}(t) \tag{8}$$

Here, $T_{\text{audio}}(t)$ represents the complete audio representation at timestamp $t$, formed by the sum of the acoustic features, $T_{\text{acoustics}}(t)$, and the transcribed dialogue, $T_{\text{dialogue}}(t)$. The acoustic data provides information on the tonal and rhythmic properties of the speech, while the dialogue data encapsulates its linguistic meaning.

By merging these components at each timestamp, the system forms a rich, unified audio representation that integrates both the tonal nuances and the semantic content of the speech. This combined

representation serves as the final, holistic audio input for further processing, enhancing the system's capacity to understand and interpret spoken language in a more nuanced and context-aware manner.

### B.3 PROCESSING OF VISUAL DATA IN EPISODIC MEMORY

The processing of visual data begins with the extraction of features from individual video frames, a critical step in converting raw image data into meaningful representations suitable for downstream tasks. Each frame $F_i$ is subjected to a series of transformations, beginning with resizing and normalization to standardize the dimensions and scale of the image. This transformation converts the input frame into a tensor $T_i \in \mathbb{R}^{C \times H \times W}$, where $H$ and $W$ are the height and width of the transformed image, and $C$ denotes the number of color channels (e.g., RGB). The transformation is formally expressed as $T_i = T(F_i)$, where $T : \mathbb{R}^{H_0 \times W_0 \times C} \to \mathbb{R}^{C \times H \times W}$, with $(H_0, W_0)$ being the original dimensions of the frame.

Once the transformation is complete, each frame is further analyzed to extract both **global** and **local** features. Global features $G \in \mathbb{R}^D$ capture high-level semantic content of the frame, summarizing its overall scene representation. Local features $L \in \mathbb{R}^{D \times H_f \times W_f}$, on the other hand, represent spatially localized details, enabling the model to attend to specific regions of the frame, such as objects or key areas of interest. These local features may then undergo downsampling, where a pooling function $P$ is applied to reduce their spatial dimensions, yielding $L_p \in \mathbb{R}^{(H_p \times W_p) \times D}$, where $H_p$ and $W_p$ are the pooled spatial dimensions.

The final output consists of the global features $G$ and the processed local features $L_{\text{final}}$, where $L_{\text{final}} = P(L)$ if downsampling is applied, or $L_{\text{final}} = L$ if no downsampling is required. These representations encapsulate both high-level semantic content and localized spatial information, allowing for a comprehensive understanding of the visual input. To represent the entire video scene, the visual embeddings of all frames $F_i$ are aggregated using a mean operation, which serves to summarize the temporal sequence of frames into a single, fixed-size representation:

$$V_{\text{scene}} = \frac{1}{N} \sum_{i=1}^{N} V_{\text{embed}}(F_i) \tag{9}$$

Here, $N$ represents the total number of frames in the video. The aggregated embedding $V_{\text{scene}} \in \mathbb{R}^D$ encapsulates the collective visual information of the scene, serving as a representative feature for tasks such as video understanding, classification, and retrieval. Additionally, individual frame embeddings $V_{\text{embed}}(F_i)$ may be retained for detailed analysis, enabling finer-grained evaluation of the video's contents. This process effectively captures both local, fine-grained details and global, high-level scene information, facilitating the model's ability to understand and interpret complex visual scenes.

#### B.3.1 EXTRACTING TIME AND PLACE DETAILS FROM IMAGES

Upon receiving visual data, the agent applies a multi-step algorithm to detect and extract textual information. Initially, a convolution feature extraction network processes the image, producing a feature map $F_{\text{feature}}(x, y)$ that highlights potential text regions. The network subsequently generates probability maps for the text center line (TCL) and text regions (TR), denoted as $P_{\text{TCL}}$ and $P_{\text{TR}}$, through the following equations:

$$\begin{pmatrix} P_{\text{TCL}}(x, y) \\ P_{\text{TR}}(x, y) \end{pmatrix} = \sigma \left( \begin{pmatrix} W_{\text{TCL}} \\ W_{\text{TR}} \end{pmatrix} \cdot F_{\text{feature}}(x, y) \right) \tag{10}$$

where $\sigma$ represents the sigmoid function, and $W_{\text{TCL}}$ and $W_{\text{TR}}$ are learned weight matrices. A thresholding operation is applied to these maps, filtering out low-confidence regions:

$$P_{\text{filtered}} = \{(x, y) \mid P_{\text{TCL}}(x, y) \geq T_{\text{TCL}} \text{ and } P_{\text{TR}}(x, y) \geq T_{\text{TR}}\} \tag{11}$$

A striding algorithm is then employed to extract ordered points along the TCL, based on displacement defined by the radius $r$ and orientation $\theta$:

$$\text{Stride} = \Delta r \cdot (\cos(\theta), \sin(\theta)) \tag{12}$$

These points are used to reconstruct the text region, represented as an ordered sequence $\{(x_1, y_1), (x_2, y_2), \ldots, (x_n, y_n)\}$. The reconstructed text instances are then stored for further analysis, enabling the agent to extract relevant temporal and spatial information. Once the visual input is processed through the convolutional and recurrent layers, text recognition is performed by a softmax layer that predicts the probability distribution over the character set at each timestep. Mathematically, this can be expressed as:

$$\hat{y}_t = \text{Softmax}(W \cdot h_t + b) \tag{13}$$

where $\hat{y}_t$ is the predicted output at timestep $t$, representing the probability distribution over characters. $h_t$ is the hidden state of the LSTM at timestep $t$, and $W$ and $b$ are the learned weight matrix and bias vector, respectively. The softmax function converts the raw logits into a probability distribution, assigning likelihoods to the possible characters at each timestep. Once the text is recognized through the softmax layer, the next step involves temporal and place tagging to extract relevant contextual information. Temporal tagging refers to identifying time-related cues within the text, such as references to specific moments, durations, or sequences. Place tagging, on the other hand, involves detecting spatial information, such as locations or references to physical spaces. These tags help filter out unnecessary details and ensure that only the most pertinent information is retained for further processing. This approach enhances the ability to focus on time- and location-specific aspects of the data, which is critical for tasks such as event prediction or contextual understanding. By applying these filters, the system narrows the scope of relevant data, facilitating a more efficient and accurate interpretation of the visual input.

To obtain person embeddings, the first step is to perform person recognition in the visual data. This involves detecting and locating the person in the scene using a pre-trained model. After the person is identified, we crop the corresponding region of interest (ROI) from the original image to isolate the person. This cropped image is then processed through a feature extractor, which generates a unique representation of the person, commonly referred to as a person embedding. This embedding captures the distinctive visual characteristics of the person, and is stored in the memory for future reference or decision-making processes. This procedure ensures that only the relevant features related to the person are retained for further tasks, such as recognition or interaction.

### B.3.2 MERGING AND SYNCHRONIZING DATA

In the final stage, the processed audio and visual data are synchronized to a common timestamp, creating a unified representation:

$$\mathbf{T_M} = (\mathbf{T}_{\text{audio}}, \mathbf{T}_{\text{visual}})$$

This ensures temporal alignment between the two modalities, enabling coherent multimodal interaction. The audio and visual embeddings are then concatenated into a joint multimodal embedding:

$$\mathbf{E}_{\text{combined}} = \mathbf{E}_{\text{audio}} \oplus \mathbf{E}_{\text{visual}}$$

where $\oplus$ denotes the concatenation operation.

This joint multimodal representation is stored in episodic memory, where each node represents a specific experience, incorporating aspects such as time, location, characters, and events. By combining the visual and audio information, this integrated embedding enhances memory recall and supports detailed event-based analysis.

### B.4 EPISODIC MEMORY REPRESENTATION

Each node in the episodic memory represents a day, with subnodes capturing the activities of that specific day. The memory structure is hierarchical, where each day's main node consolidates the overall event, while subnodes encode the individual activities or events of that day. After generating the joint embedding that integrates place, character, and event information, these embeddings are further refined and organized within event nodes, which summarize the details of each event. The main node, which corresponds to the day, retains a higher-level summary of the events, while the

event nodes hold the specific details. This structure allows for efficient encoding and retrieval, enabling the agent to access a detailed account of both the broader context and specific interactions, improving the agent's decision-making and understanding of past experiences. The joint embedding, representing the integration of place, character, and event, is stored at both the event and day levels for efficient memory recall.

### B.4.1 DYNAMIC CLUSTERING

When an agent receives an episodic experience, it dynamically assigns the experience to clusters based on location, events, and characters. Each experience may belong to multiple clusters: the Location Cluster $\mathbf{C}_l$, Character Cluster $\mathbf{C}_c$, and Event Cluster $\mathbf{C}_e$. Each entity within these clusters is assigned a unique identifier, enabling efficient organization and retrieval of past experiences. This clustering mechanism works in conjunction with the hierarchical structure of episodic memory, where nodes represent days and subnodes capture specific activities and events, allowing for refined memory encoding and improved decision-making.

### B.4.2 CLUSTER INTEGRATION

Both text and visual embeddings are used to restore feature details and enhance decision-making. Spatial and character embeddings are derived based on the similarity between locations and characters across episodes. For each new episode $E_n$, these embeddings are evaluated and integrated into the existing clusters. The attention mechanism evaluates the relevance of the new episode to previously stored episodes using the formula:

$$\text{attention}_X(E_n, E_i) = \frac{\mathbf{V}_{X_n} \cdot \mathbf{V}_{X_i}}{\|\mathbf{V}_{X_n}\|\|\mathbf{V}_{X_i}\|}$$

where $X \in \{\text{location}, \text{char}\}$, allowing for a dynamic clustering process that organizes memory by spatial, character, and event-related similarities. When a new episode does not fit into an existing cluster, a new identifier is created, ensuring that clusters continuously expand to accommodate new data.

Event clusters are formed based on the content of dialogues, which exhibit high variability and can span multiple events. The system evaluates conversations at the event level. Each dialogue $D = (u_1, u_2, \ldots, u_{|D|})$ consists of $|D|$ utterances, each of which can be assigned to one or more events. Since event labels are often difficult to obtain without expert annotation or complex segmentation algorithms, we instead infer the event structure by modeling the coherence among consecutive utterances. The assumption is that utterances within the same event exhibit higher coherence than those spanning multiple events.

To capture this, we introduce a contrastive learning objective. A dialogue is broken into snippets, each consisting of a window of $k$ consecutive utterances. Positive examples of snippets are those within the same event, while negative examples are those spanning different events. The coherence between snippets is evaluated using a contrastive loss function. Specifically, for a dialogue $D$, the objective is to maximize the coherence between positive snippets and minimize the coherence between negative ones. The contrastive coherence detection is formalized as:

$$L_{\text{contrastive}} = \sum_i \left[\log p(\text{positive}_i) - \log p(\text{negative}_i)\right]$$

where positive and negative examples are selected based on coherence metrics, such as ROUGE score. This allows for unsupervised event detection and clustering.

The contrastive coherence detection, coupled with the learning of event relationships, allows for the effective segmentation of dialogues into relevant event clusters. Contextual edges between episodes are formed by analyzing shared clusters, which focus on common characters, locations, or events. The similarity between two episodes $E_1$ and $E_2$ is determined by the number of overlapping clusters, with an edge created if two or more clusters are shared.

The weight of the edge is calculated by combining the similarity scores of location, character, and event embeddings:

$$\text{Weight}(E(N_1, N_2)) = \mathbf{L}(N_1, N_2) \| \mathbf{C}(N_1, N_2) \| \mathbf{E}(N_1, N_2)$$

where $\mathbf{L}(N_1, N_2)$ represents the location similarity between two episodes, $\mathbf{C}(N_1, N_2)$ corresponds to the character similarity, and $\mathbf{E}(N_1, N_2)$ reflects the event similarity.

The primary connection, or "link," between two episodes is established by the maximum similarity among the location, character, or event features:

$$\text{Link}(E(N_1, N_2)) = \arg\max\left(\{\mathbf{L}(N_1, N_2), \mathbf{C}(N_1, N_2), \mathbf{E}(N_1, N_2)\}\right)$$

**Temporal Edges** are defined by the time relationship between consecutive episodes. The temporal connection between episodes $E_{t-1}$ and $E_t$ is represented as $\mathbf{T}_{\text{edge}}(E_{t-1}, E_t) = 1$, with The temporal weight is given by $\mathbf{T}_{\text{weight}}(E_{t-1}, E_t) = t - (t-1) = 1$.

### B.5 DYNAMIC EDGE TRAVERSAL

Dynamic edge traversal for retrieving relevant memories using character, location, event, and temporal weights. Based on the query $q$, the agent assigns tags: $P \leftarrow E[\text{People}]$, $L \leftarrow E[\text{Location}]$, $V \leftarrow E[\text{Event}]$, and $R \leftarrow E[\text{Temporal}]$. These tags enable the agent to focus on relevant aspects of the query, addressing the "What," "Where," and "When" elements that form the foundation of episodic memory.

Episodic memory tasks, which often center on answering "What-Where-When" (WWW) questions, are designed to capture the essence of episodic memory. Such tasks have been extensively used to study episodic(-like) memory in non-human animals, and similar methods can be applied to humans. In this context, participants are tasked with recalling specific events (what), associated locations (where), and their temporal sequence (when). These WWW tasks offer valuable insights into the mechanisms of episodic memory and the strategies, such as mental time travel, employed to solve them. For instance, studies have shown that participants actively memorizing WWW information often rely on episodic memory systems, whereas those passively encoding such information may engage alternative systems for where and when components.

Temporal entities, such as weeks, months, or years, are processed by subtracting fixed intervals from the current date. The number of weeks is calculated by subtracting $7n$ days, the number of months by subtracting $30m$ days, and the number of years by subtracting $365y$ days, where $n$, $m$, and $y$ are positive integers. This mechanism allows the agent to account for temporal relationships without explicit date references, enabling the handling of time-related queries in a flexible manner.

The similarity score between the query $q$ and a set $D_u$ of memory entries is computed as $S_s = \sum_{e \in D_u} \frac{q \cdot e}{\|q\|\|e\|}$. This score uses cosine similarity to assess the relevance of each memory entry in the context of the query.

For each neighbor $v$ of node $u$, the weight $W_{uv}$ is calculated by $W_{uv} = \sum w(u, v)$, where $w(u, v)$ aggregates the weights of shared features, such as characters, locations, and events. If $W_{uv} > \theta$, the query set is updated as $Q \leftarrow Q \cup (v, W_{uv})$. Temporal edges $T_{\text{edge}}(E_{t-1}, E_t)$ persist, allowing the agent to maintain continuity in the memory graph and reason contextually without re-evaluating the entire graph. This approach ensures the efficient retrieval of temporally connected events, supporting the dynamic traversal of episodic memories.

Location-tagged queries explore place clusters, character-tagged queries utilize character-based connections, and event-tagged queries trace event-related edges. This traversal strategy ensures that episodic memory is effectively leveraged to answer WWW queries, enabling the agent to recall not only the specific details of events but also their spatial and temporal context. These capabilities are crucial for supporting decision-making processes in scenarios that rely on detailed memory recall.

## C NODE PRUNING AND TEMPORAL EDGE UPDATING IN GRAPHS

In graph theory, pruning involves the selective removal of nodes while ensuring the structural integrity of the graph is maintained. A critical aspect of this process is the treatment of temporal edges, which represent time-dependent relationships between nodes. The temporal edge updating mechanism can be described as follows:

For a node $n$ to be pruned, temporal edges $E_{\text{temporal}}(n)$ are first identified. These edges connect $n$ to its temporal neighbors and are classified by the attribute edge_type = temporal. Once the temporal edges are identified, the temporal neighbors $T(n)$ are evaluated, resulting in two possible scenarios.

If the node $n$ has two temporal neighbors, denoted as $n_{\text{before}}$ and $n_{\text{after}}$, the edges $(n_{\text{before}}, n)$ and $(n, n_{\text{after}})$ are removed. To preserve the temporal relationship, a new edge $(n_{\text{before}}, n_{\text{after}})$ is introduced with the attribute edge_type = temporal:

$$E \leftarrow E \setminus \{(n_{\text{before}}, n), (n, n_{\text{after}})\} \cup \{(n_{\text{before}}, n_{\text{after}})\}.$$

This operation ensures that the temporal continuity between $n_{\text{before}}$ and $n_{\text{after}}$ is preserved after $n$ is removed.

If $n$ has only one temporal neighbor $n_{\text{connected}}$, the sole temporal edge $(n_{\text{connected}}, n)$ or $(n, n_{\text{connected}})$ is simply removed:

$$E \leftarrow E \setminus \{(n_{\text{connected}}, n)\} \quad \text{or} \quad E \leftarrow E \setminus \{(n, n_{\text{connected}})\}.$$

In this case, no new edge is added, as the temporal connection terminates with the removal of $n$.

After updating the temporal edges, the node $n$ is removed from the graph, ensuring it no longer contributes to the structure:

$$V \leftarrow V \setminus \{n\}.$$

Non-temporal edges connected to $n$ are retained without modification, preserving static relationships.

This process ensures that temporal relationships in the graph are maintained or updated appropriately, allowing for meaningful temporal reasoning and analysis even after node pruning. The approach safeguards the continuity of temporal information while ensuring that the graph remains coherent and analyzable.

## D   IMPLEMENTATION DETAILS

For event extraction in dialogues, we utilized a **Transformer-based BART** model initialized with pre-trained weights to effectively extract and summarize events within contextual boundaries. The architecture options included **BARTBASE**, with a 6-layer encoder-decoder and approximately 140 million parameters, and **BARTLARGE**, featuring a 12-layer encoder-decoder and 400 million parameters. Both configurations maintain a hidden size of 1024 and a feed-forward filter size of 4096, with dropout rates fixed at 0.1 across layers. The **Fairseq toolkit** was employed for training, with the **Adam optimizer** using warmup strategies. Learning rates were set at $4 \times 10^{-5}$ and $2 \times 10^{-5}$ for BARTBASE and BARTLARGE, respectively, with batch token limits set at 1100 tokens. Contrastive objectives were supported by a margin coefficient of 1, while hyperparameters for coherence and sub-summary objectives were tuned using a validation set. Our approach showed significant performance improvements compared to publicly available models trained on datasets such as **SAM-SUM** and **DialogueSUM**.

For visual processing, a **Vision Transformer (ViT)** served as the vision encoder, specifically adapted for video frame analysis from the **MSR-VTT dataset**. The encoder processed $224 \times 224$ video frames, segmented into patches of size 16, and embedded these into a 512-dimensional latent space. This 12-layer encoder had a width of 768 and utilized **LayerScale** (initialized at 0.1) for training stability. Advanced regularization methods, including stochastic depth with a variable drop_path_rate, were applied. The encoder was based on the "eva-clip-b-16" model and proved effective for extracting detailed spatial and temporal features essential for multimodal tasks.

For **LLaMA**-based models integrating vision and dialogue for character and place tagging, a multimodal configuration was employed. **ViT** was used for image processing while **LLaMA** handled dialogue inputs. The training included cross-entropy loss for character tagging, contrastive loss for image-text alignment, and incorporated **episodic memory** for QA tasks. Training leveraged the **AdamW optimizer**, a dropout rate of 0.2, and a **cosine annealing scheduler** for efficient learning.

**Temporal tagging** was configured with key hyperparameters for optimal performance: maximum sequence length of 128, batch size of 32, and a learning rate of $5 \times 10^{-5}$. Dropout was set at 0.1

to mitigate overfitting, and weight decay at 0.01 to improve generalization. Training spanned 10 epochs to ensure learning adequacy while avoiding overfitting.

For text detection, the **TextSnake** model was trained on the **SCUT-CTW1500** dataset using **SGD with Momentum** as the optimizer. The architecture combined **ResNet** and **FPN_UNet**, with training configurations involving a batch size of 64 and 8 workers for data loading. The validation batch size was set to 1 with 4 workers, and persistent workers were enabled. The training was set for 200 epochs with validation checks every 10 epochs.

The **QA system** was built using a **BERT** model fine-tuned on concatenated datasets, including **SQuAD**, **Wikipedia**, and **Reddit**, to enhance contextual understanding. The hyperparameters included a learning rate of $1 \times 10^{-5}$, a maximum sequence length of 512, and a document stride of 512. The training batch size was 8, with gradient accumulation steps of 2, spanning 2 epochs. Mixed-precision training was utilized with 'fp16' at **O2 optimization level** for efficiency. The final output was stored in the 'bart-squadv2' directory without saving models at each epoch.

Table 6: Hyperparameter Configuration for Model Implementations

| Model Component | Hyperparameter | Value |
|---|---|---|
| Event Extraction (BART) | Encoder-Decoder Layers | 6 (BASE), 12 (LARGE) |
| | Hidden Size | 1024 |
| | FFN Size | 4096 |
| | Dropout | 0.1 |
| | Learning Rate | $4 \times 10^{-5}$ (BASE), $2 \times 10^{-5}$ (LARGE) |
| | Max Tokens per Batch | 1100 |
| | Margin Coefficient | 1 |
| Vision Encoder (ViT) | Patch Size | 16 |
| | Resolution | $224 \times 224$ |
| | Latent Space Dim. | 512 |
| | Transformer Layers | 12 |
| | Width | 768 |
| | LayerScale Init. | 0.1 |
| | Dropout Path Rate | Configurable |
| QA System (BERT) | Learning Rate | $1 \times 10^{-5}$ |
| | Max Sequence Length | 512 |
| | Document Stride | 512 |
| | Train Batch Size | 8 |
| | Gradient Accum. Steps | 2 |
| | Epochs | 2 |
| | Mixed-Precision Opt. | fp16 (O2) |
| Temporal Tagging | Max Sequence Length | 128 |
| | Batch Size | 32 |
| | Learning Rate | $5 \times 10^{-5}$ |
| | Dropout | 0.1 |
| | Weight Decay | 0.01 |
| | Epochs | 10 |

# E ADDITIONAL RESULTS

## E.1 RESULTS WITH MORE MEMORY MODELS USING EGO4D DATASET

The table compares the recall accuracy of various models, including state-of-the-art (SOTA) methods, evaluated on the Ego4D dataset. Notably, Episodic Memory Verbalization leverages a graph-based memory model to store and retrieve information, making it uniquely suited for tasks requiring structured memory organization. The results highlight the importance of incorporating dialogues and time-series data into memory representations. Our method, achieving a recall accuracy of 81%, significantly outperforms other models, demonstrating the efficacy of our approach in storing and utilizing temporal and conversational context effectively.

| Method | Recall Accuracy |
|---|---|
| Episodic Memory Verbalization | 50% |
| Rehearsal Memory | 36% |
| STM | 30% |
| DNC | 35% |
| LT-CT | 50% |
| Ours | 81% |

Table 7: Comparison of Recall Accuracy Across Different Methods

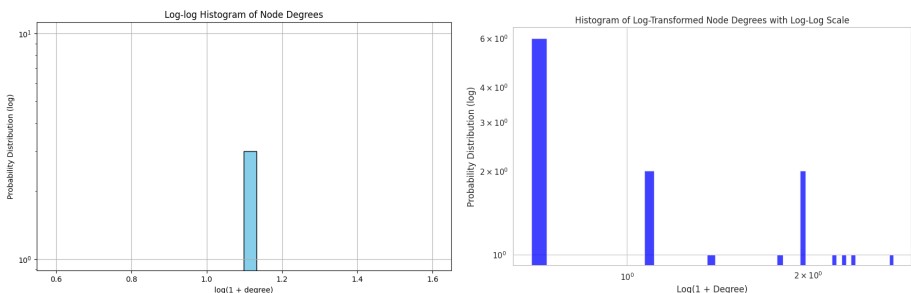

(a) Log Degree Distribution of the Episodic Graph

(b) Log Degree Distribution of the Knowledge Graph

Figure 7: Degree distribution indicates reduced connectivity in the episodic graph, promoting relevant memory transfer

### E.2 ABLATION STUDIES

We conducted an additional ablation study to assess the performance of the knowledge graph and episodic memory graph. This involved replacing the episodic memory structure with a knowledge graph and evaluating the impact of removing nodes from the graph.

#### E.2.1 COMPARISON WITH KNOWLEDGE GRAPHS

This section evaluates the replacement of episodic graphs with knowledge graphs in an episodic memory agent. Table 8 summarizes the comparison.

| Episodes | Episodic Graph (Nodes) | Knowledge Graph (Nodes) | Episodic Graph (Edges) | Knowledge Graph (Edges) |
|---|---|---|---|---|
| 3 | 3 | 12 | 3 | 20 |
| 10 | 10 | 50 | 10 | 90 |
| 20 | 20 | 110 | 20 | 220 |
| 50 | 50 | 280 | 50 | 600 |
| 100 | 100 | 600 | 100 | 1400 |

Table 8: Relationship Between Episodes and Graph Metrics for Episodic and Knowledge Graphs

Figure 7 illustrates that the episodic graph's lower connectivity facilitates efficient memory retrieval.

Table 9 compares the power-law characteristics of both graphs. The episodic graph shows a steep decay ($\alpha = 5.45$), indicating simplicity, while the knowledge graph's slower decay ($\alpha = 1.57$) reflects its complexity. Figure 8 reinforces that the episodic graph fits a power-law distribution strongly, while the knowledge graph does not.

In conclusion, the complexity of knowledge graphs hampers efficient memory retrieval. Conversely, episodic graphs enable quick extraction of relevant memories, facilitating faster interactions and improved reasoning.

#### E.2.2 RESULTS AFTER PRUNING

We examine the effectiveness of our pruning method by selectively removing nodes from a graph representation of the Big Bang Theory dataset, a network that captures interactions and relationships between characters. Specifically, we tested the pruning function by removing nodes with total connection weights below thresholds of 3 and 5. This approach helps us analyze how different pruning

| Metric | Episodic Graph | Knowledge Graph |
|---|---|---|
| Power-Law Exponent ($\alpha$) | 5.45 (Steep decay) | 1.57 (Slower decay) |
| Minimal Value ($x_{min}$) | 1.0 (Valid from degree 1) | N/A |
| Standard Error of $\alpha$ | 0.341 (Moderate precision) | 0.142 (Higher precision) |
| Log-Likelihood Ratio (R) | 299.15 (Strong positive value) | -0.88 (Exponential fits better) |
| p-value | $6.03 \times 10^{-172}$ (Very small) | 0.379 (No significant difference) |

Table 9: Power-Law Characteristics of Graphs

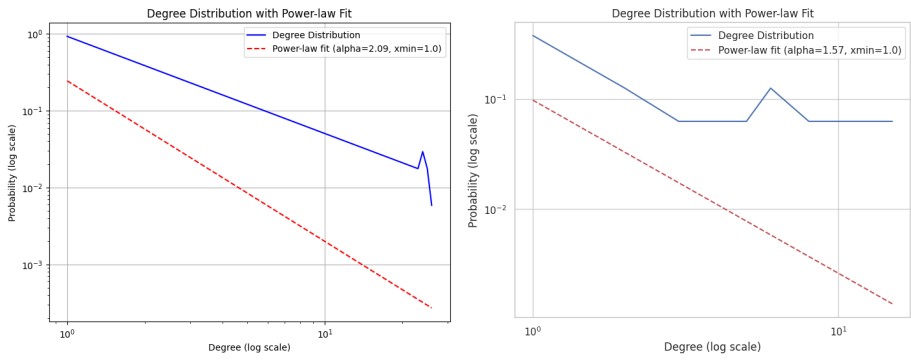

(a) Power Law Fit for the Episodic Graph     (b) Power Law Fit for the Knowledge Graph

Figure 8: Power-law analysis reveals a strong fit for the episodic graph, while the knowledge graph shows no significant power-law structure

intensities impact the overall structure and connectivity within the network, providing insight into the robustness and adaptability of the model when less significant nodes are removed.

| | What | When | Where | All |
|---|---|---|---|---|
| **Full Model** | 75 | 80 | 76 | 78 |
| **After pruning(Weight 1-3)** | 73 | 78 | 75 | 75 |
| **After pruning(Weight 1-5)** | 55 | 50 | 55 | 51 |

Table 10: Performance metrics of the model before and after pruning at different weight thresholds.

The pruning analysis shows that the graph constructed from the Big Bang Theory dataset is highly adaptable, maintaining its core structure and key temporal connections even after pruning. While it effectively captures both speech and visual details, the graph's performance is impacted by the use of LLM, resulting in slower processing and reduced reliability. This emphasizes the need for a balanced pruning approach to optimize performance while retaining essential features

### E.3 EPISODIC MEMORY LOCALIZATION

. Figure 9 is a visual representation of episodic memory localization. The tests focus on the agent's ability to process episodic memory queries, localize relevant data efficiently within time-series inputs, and respond accurately. Performance metrics such as localization accuracy, response time, and the ability to reason over sequential events are analyzed to validate the model's robustness and adaptability to dynamic tasks.

### E.4 SIMULATION TESTING

Models are inadequate for episodic memory localization when dealing with time-series and multi-modal data, including vision and dialogues.

We tested our model in Unity 3D with the agent interacting in a simulated environment. The agent gathered experiences and built episodic memory by navigating and interacting with characters. For example, after learning about a football registration task from a character, the agent uses "what," "when," and "where" questions to retrieve and act on the relevant memories. For instance: Where

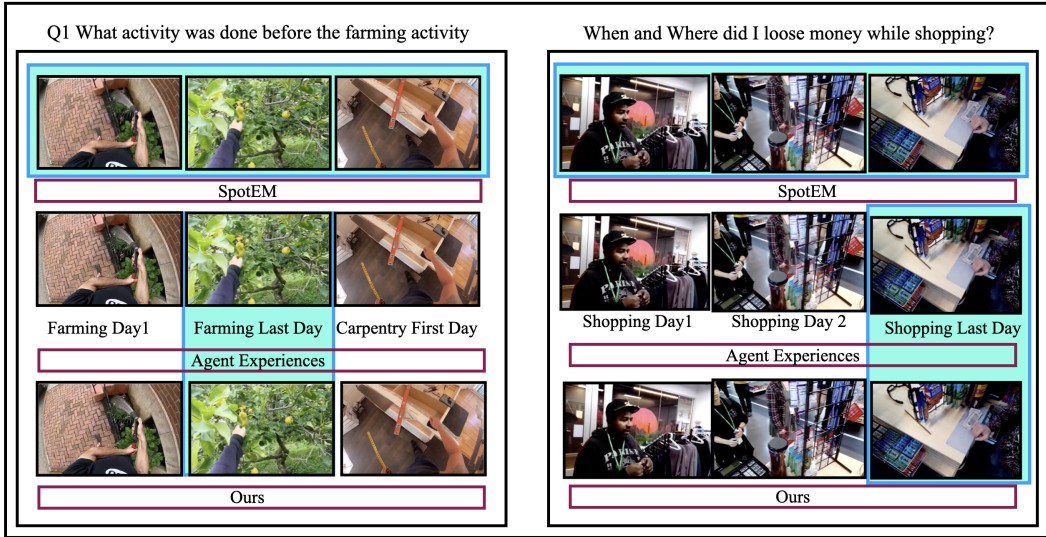

Figure 9: **Visualization of Model Performance with SpotEM on Time Series Data:** The blue window represents the localization window. The performance comparison reveals that SPOTEM, not designed for reasoning over multiple videos simultaneously, struggles with time series localization for episodic questions. In contrast, our model effectively identifies and retrieves the correct data chunk from memory to answer time series queries.

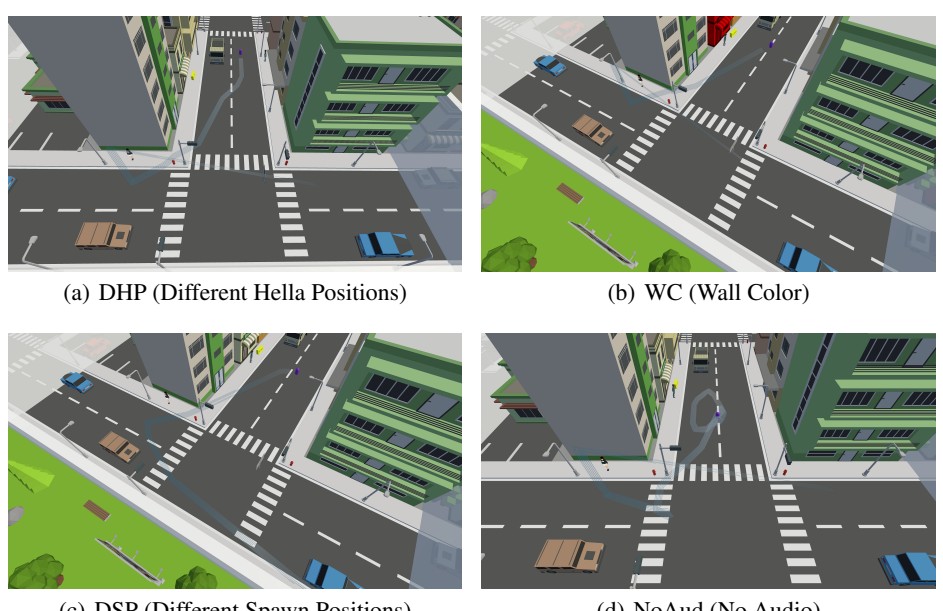

Figure 10: Agent trajectories for GTrXL model across various scenarios: (a) Different Hella Positions (DHP), (b) Wall Color (WC), (c) Different Spawn Positions (DSP), and (d) No Audio (NoAud).

should the agent go to complete the registration?, Where did the agent learn about the registration process?, When is the registration supposed to occur?, What are the steps for registration?.

The retrieved memory indicated that Hella had the registration details. The agent, unfamiliar with the football club, must seek Hella for more information. Figure **??** shows the memory chunk obtained. We made use of the the GTrXLParisotto et al. (2019) and TrXLDai et al. (2019) models across four and evaluated scenarios: different Hella positions (DHP), wall color changes (WC), different spawn positions (DSP), and no audio (NoAud). The success rate, defined as the percentage of successful episodes, was measured alongside efficiency and reward metrics. Higher rewards signify better performance in reaching goals and optimizing paths. We examine how audio and video data stored in the episodic memory graph help the agent identify accurate goal locations and develop an

**Memory**

**Episode_number**: 1
**Time**: 09:30:00
**Characters**: agent ,Hella
**Place**: Restaurant
**Event**I come to a restaurant at 9:30 in the morning and mets a lady
who introduces herself as Hella who happens to be the owner of
all restaurants in the locality.
She suggests a good place where the agent can get Mexican food.
She informed me that I and my master have to register in the
locality and all the registrations were done in the football club

Figure 11: Memory chunk of an agent after extracting questions from master commands (e.g., 'Go and do the registration'). It includes the master command, extracted questions, contextual information, an action plan, and temporal dependencies between tasks.

effective algorithm to detect the correct path. Below is a representation of how stored modalities and retrieved goals aid the agent in finding the optimal path to its goal location.

- **Changing Hella's Position:** The agent adapts efficiently to different Hella positions, achieving high rewards, as shown in Figure 10(a).

- **Changing Wall Color:** The agent maintains stable trajectories and high efficiency despite wall color changes, as depicted in Figure 10(b).

- **Different Spawn Locations:** The agent navigates effectively from various starting points, although rewards decrease with greater distances from default positions (Figure 10(c)).

- **No Audio:** In the absence of audio, the agent relies solely on visual inputs, resulting in longer paths and lower rewards (Figure 10(d)), highlighting the importance of audio for improved navigation and goal localization.

Now, consider the agent interacting with four characters before reaching the football club. If asked How did I reach the football club?, the internal question generation module might generate questions like: Where was I before the football club?, Where did I go before the football club?, When did I reach the club?. The episodic retrieval module (as described in **??**) will answer these, retrieving relevant data to assist with goal planning based on past experiences. Overall, integrating audio data and graph memory significantly enhances pathfinding and goal planning. This underscores why audio data plays a crucial role in goal planning, making our model more effective in guiding agents towards their objectives in complex, multimodal environments.

