# OpenReview forum: "Learning through experience:Episodic memory representation for cognitive agents"
_ICLR.cc/2025/Conference — Submitted to ICLR 2025_

### Official Review · Reviewer_YT1b · 2024-10-27

**Soundness:** 3
**Presentation:** 2
**Contribution:** 2
**Rating:** 3
**Confidence:** 3

**Summary:**

The paper proposes a novel method for storing multi-modal episodic memories. The design takes inspiration from a model of human memory. More specifically, the paper introduces a novel type of memory graph with different types of connections. The method compares favorably against baselines in experiments.

**Strengths:**

The paper tackles and interesting problem. The architecture is described in an intuitive way and seems sound and novel. The experiments are reasonably extensive with comparisons to baselines, multiple datasets, and ablations and show very promising results.

**Weaknesses:**

- I found the paper to have some "false advertising":
  - Some of the introduction, motivation, and discussion circles around 'robots', I didn't see anything specific to robots in this paper. Yes, it could be integrated into a robot, but the method would equally well work for a body cam, social agent, etc. In robotics there is quite an extensive literature on 'lifelong learning' that covers some of the same challenges: what to memorize, how to store and to retrieve, how to generalize, what to forget, etc.
  - The title says 'learning'. At least according to my definition there is no learning in the paper. It proposes a way to store information and to retrieve it, so that would correspond to 'memorization' (=rote learning), while learning implies understanding the information and being able to apply it to new situations. The method could serve as a starting point for learning, but in the current paper that doesn't seem to be present in the method nor in the experiments.
  - The introduction makes it sound like a general method. But I have a few doubts about that. There are quite a lot of design choices, and some of choices in Sect. 3 seem rather specific. In the end the method is evaluated with a question answering task. What would need to change for a different task, say for a robot learning low-level movement control? Please clarify the generalizability of your method. Specifically, it would be nice if you could discuss how your approach might be adapted for different tasks (such as robot movement control), or explain any limitations in its applicability to other domains.
- Section 3 describes the method. It remains however very much on the level of HOW, rather than providing many insights in the WHY (reasoning behind design choices, consequences of design choices) and it isn't always clear what is a core part of the method and what is an implementation detail. Please provide more explanation for the key design decisions, discussing the rationale behind these choices and their potential implications. Additionally, please clearly distinguish between core methodological components and implementation details.
- Fig. 5 isn't very convincing (except for Arigraph)
- Memory requirement would be another interesting metric for comparing methods.
- The paper mentions the missing forgetting mechanism as a major limitation. Related to that it also leaves the question on 'what to store' unanswered. It reads like everything is stored, even if it is effectively a duplicate. I believe the real challenging question that needs to be solved is the memory management: what to store, what to consolidate/merge, what to forget, etc. Without those a memory representation is of limited value, and it remains unclear to me how suitable the proposed architecture is for extending it in that way - or if we would be better off redesigning it from scratch.
- The method relies on various models to extract features and to summarize things before storing them. I don't believe there is a 'one size fits all approach' but how to best do that depends on the retrieval task and the type of downstream tasks you have. The paper does not provide any indications on how to deal with that.
- There are a whole lot of design choices in this paper, an ablation on only the modalities and search methods seems a bit limited. There also is no sensitivity analysis for e.g. the clustering thresholds
- This seems to be a rather complex system, which makes reproducing results very difficult, with probably quite lot of additional implementation and setting details. I couldn't find any promises to release code (or at least a detailed appendix), which would have alleviated this concern.

**Questions:**

- The paper seems to miss all details on how the responses are generated, but the whole experiments are based on evaluating the answers. Please provide a detailed explanation of your response generation process, as this is crucial for understanding and evaluating the experimental results.
- Sect. 3.2.4: This seems to result in just 'jumping' nodes. Conserving the chronological structure is a nice property, but the real question/challenge rather is on the side of the module that determines if there is any harm in 'skipping a step', e.g. when thinking about some of the uses-cases 'predict ... next activity' you mention
- Quite a lot of unclear details
  - Eq (1) vs (2): the difference between S_T and T_voiced is unclear
  - How is T_acoustic generated
  - Sect. 3.1.2, 3.1.3, and 3.2, l. 218: at places the paper sounds like everything is represented as 'text', at others it seems to be a mixture of text and other embeddings. It would be great to explain earlier on what it stored as what
  - l. 184: "organized by place, characters, and events" raises the question where those come from - that is explained later in the text
  - l. 220: "relevance of these tasks is assessed" - again HOW?
  - l. 212: what are the implications/limitations resulting form using a simple metric like cosine similarity?
  - Eq (17) sim seems undefined
- Fig. 2 and Sect. 4.1: I didn't get the terminology "master". Maybe that term can be avoided altogether (similar to https://learn.microsoft.com/en-us/style-guide/a-z-word-list-term-collections/m/master-slave )
- The paper comes across as unpolished
  - missing and extra spaces - e.g. in the title and abstract
  - the template uses natbib, not using correct commands for references \citep \citet (and resulting repeats of names) makes it painful to read
  - LaTeX has different opening and closing quotation marks
  - paragraph above "Contributions" is double
  - broken sentences in Sect. 5.1.2
  - a few references with incomplete info (e.g. l. 300 "as shown in 4", l. 191, l.468)

##After rebuttal
I appreciate the very detailed replies. However, I still believe the paper is not ready to be published and will maintain my score (see message below).

**Details Of Ethics Concerns:**

When this method gets deployed, there would potentially be ethics concerns - as pointed out by the authors.
In the submission, it just uses datasets (established and generated form a TV series) rather than real user data, so no concerns.

---

> ### Author Response · Authors · 2024-11-20
> **Some of the introduction, motivation, and discussion circles around 'robots', I didn't see anything specific to robots in this paper. Yes, it could be integrated into a robot, but the method would equally well work for a body cam, social agent, etc. In robotics there is quite an extensive literature on 'lifelong learning' that covers some of the same challenges: what to memorize, how to store and to retrieve, how to generalize, what to forget, etc.**
>
> We appreciate the reviewer’s observation regarding the scope of the paper and the use of the term "robot" in the introduction. While the proposed method is applicable to diverse contexts such as body cams or social agents, it is important to clarify the focus and implications of our approach.
> In robotics, 'lifelong learning' addresses challenges similar to those in episodic memory for agents, including deciding what information to store, how to efficiently organize and retrieve it, generalizing knowledge across experiences, determining what to forget, and adapting to new inputs over time.
> Our system integrates key principles of lifelong learning within a structured and adaptive memory management framework. It prioritizes what to memorize by focusing on keyframes and significant events, ensuring that only relevant and non-redundant information is retained. Keyframe extraction reduces redundancy by identifying frames that reflect spatio-temporal changes, utilizing vision transformers to ensure alignment with human perception. For dialogues, the system employs annotated datasets to train models, such as BART, to extract events from a third-person perspective, enabling efficient and event-focused memory storage. This ensures that not everything encountered by the agent is stored indiscriminately in memory.
> Moreover, while lifelong learning frameworks are often tailored to specific tasks, they are prone to issues like catastrophic forgetting, where older information is lost when new data is introduced. Such limitations are not ideal for episodic memory systems, which are designed to preserve a comprehensive and retrievable record of experiences. By addressing these aspects, our approach offers a robust and scalable solution for episodic memory in agents.

---

> ### Author Response · Authors · 2024-11-20
> **The title says 'learning'. At least according to my definition there is no learning in the paper. It proposes a way to store information and to retrieve it, so that would correspond to 'memorization' (=rote learning), while learning implies understanding the information and being able to apply it to new situations. The method could serve as a starting point for learning, but in the current paper that doesn't seem to be present in the method nor in the experiments.**
>
> Regarding the notion of "learning" in the title, you are correct that the current paper primarily focuses on memory storage and retrieval, which may initially appear to align more with memorization than traditional learning. However, the ability to access and utilize past experiences stored in memory significantly enhances decision-making capabilities. By leveraging these memories, the agent can adapt to new tasks or environments based on prior experiences, demonstrating a form of experiential learning.
> For instance, the system can use stored memories to identify patterns(if present), detect intrinsic goals, and develop strategies, such as navigating to a specific location or resolving a task based on similar past situations. While not explicitly addressed in the current experiments, this adaptive use of memory aligns with learning processes and highlights how the proposed method goes beyond rote memorization. We will include additional clarifications and examples of this adaptive learning potential in the appendix of the paper.

---

> ### Author Response · Authors · 2024-11-20
>
> **While the current implementation focuses on question-answering tasks, the core framework is inherently flexible and adaptable to a variety of applications. By leveraging episodic memory as the foundation for storing and retrieving relevant experiences, this approach can be extended to domains such as robot movement control and other complex tasks. The introduction makes it sound like a general method. But I have a few doubts about that. There are quite a lot of design choices, and some of choices in Sect. 3 seem rather specific. In the end the method is evaluated with a question answering task. What would need to change for a different task, say for a robot learning low-level movement control? Please clarify the generalizability of your method. Specifically, it would be nice if you could discuss how your approach might be adapted for different tasks (such as robot movement control), or explain any limitations in its applicability to other domains.**
>
>
> #### This method could further enhance robotic systems by enabling them to recall past interactions with users, providing personalized assistance, adapting to user preferences, and improving task efficiency over time. By integrating visual, auditory, and textual data into episodic memory, the system supports richer interaction scenarios, such as interpreting complex instructions or responding to ambiguous user commands. Additionally, the ability to store and retrieve task sequences empowers robots to perform multi-step operations autonomously and adaptively. In educational or learning environments, this approach can personalize the teaching process by storing interactions and assessments for tailored guidance.
>
> To adapt this method for motion control or other domains, relevant information can be incorporated into the episodic memory. For instance, in motion control, keyframes could store joint positions, sensor data, or environmental feedback. This flexibility allows the system to accommodate domain-specific data without the need to develop a completely new model, ensuring scalability and reusability across diverse applications.We can also use this model to derive goals intrinsically which will help an agent make policies for navigation.

---

> ### Author Response · Authors · 2024-11-20
>
> Section 3 describes the method. It remains however very much on the level of HOW, rather than providing many insights in the WHY (reasoning behind design choices, consequences of design choices) and it isn't always clear what is a core part of the method and what is an implementation detail. Please provide more explanation for the key design decisions, discussing the rationale behind these choices and their potential implications. Additionally, please clearly distinguish between core methodological components and implementation details.
> Implementation Details
> For event extraction within dialogues, we utilized a Transformer-based BART model initialized with pre-trained weights to effectively extract and summarize events within contextual boundaries. The architecture options included BARTBASE, featuring a 6-layer encoder-decoder with approximately 140 million parameters, and BARTLARGE, which has a 12-layer encoder-decoder and 400 million parameters. Both configurations use a hidden size of 1024 and a feed-forward filter size of 4096, with a fixed dropout rate of 0.1 across layers. We employed the Fairseq toolkit for training, using the **Adam optimizer** with a warmup strategy. Learning rates were set to $4 \times 10^{-5}$ for BARTBASE and $2 \times 10^{-5}$ for BARTLARGE, with a maximum batch token limit of 1100 tokens. Contrastive objectives were supported by a margin coefficient of 1, and hyperparameters for coherence and sub-summary objectives were tuned using a validation set. Our method demonstrated substantial performance improvements compared to publicly available models trained on datasets like **SAMSUM** and **DialogueSUM**.
>
> For visual data processing, we used a **Vision Transformer (ViT)** as the vision encoder, specifically adapted for video frame analysis from the **MSR-VTT dataset**. The encoder processed $224 \times 224$ video frames, segmented into 16-sized patches and embedded into a 512-dimensional latent space. The 12-layer encoder, with a width of 768, was equipped with **LayerScale** (initialized at 0.1) for training stability. Advanced regularization techniques, including stochastic depth with a configurable \texttt{drop\_path\_rate}, were applied. The encoder was based on the "eva-clip-b-16" model, which proved effective in extracting detailed spatial and temporal features essential for multimodal tasks.
> For models based on **LLaMA** that integrate vision and dialogue for character and place tagging, a multimodal configuration was used. **ViT** processed visual data, while **LLaMA** managed dialogue input. Training included cross-entropy loss for character tagging and contrastive loss for image-text alignment, incorporating **episodic memory** for QA tasks. The training process leveraged the **AdamW optimizer**, a dropout rate of 0.2, and a **cosine annealing scheduler** for efficient learning.
>
> **Temporal tagging** was optimized with key hyperparameters for best performance: a maximum sequence length of 128, a batch size of 32, and a learning rate of $5 \times 10^{-5}$. A dropout rate of 0.1 was used to mitigate overfitting, and a weight decay of 0.01 improved generalization. The training process spanned 10 epochs to ensure sufficient learning while preventing overfitting.
>
> For extracting dialogue from audio, the **Whisper-large model** was employed. This model was utilized for its robustness in transcribing and converting spoken content into text for further processing.
>
> For text detection, the **TextSnake** model was trained on the **SCUT-CTW1500** dataset using **SGD with Momentum** as the optimizer. The architecture combined **ResNet** and **FPN\_UNet**, configured with a training batch size of 64 and 8 workers for data loading. The validation batch size was set to 1, with 4 workers and persistent workers enabled. Training was conducted over 200 epochs with validation checks every 10 epochs.
>
> The **QA system** was built using a **BERT** model fine-tuned on concatenated datasets, including **SQuAD**, **Wikipedia**, and **Reddit**, to improve contextual comprehension. The hyperparameters included a learning rate of $1 \times 10^{-5}$, a maximum sequence length of 512, and a document stride of 512. The training batch size was 8, with gradient accumulation steps of 2 over 2 epochs. Mixed-precision training was used with `fp16` at the **O2 optimization level** for better efficiency. The final model outputs were stored in the `bart-squadv2` directory, without intermediate model saving.

---

> ### Author Response · Authors · 2024-11-20
>
> | **Model Component**         | **Hyperparameter**        | **Value**                            |
> |-----------------------------|---------------------------|--------------------------------------|
> | **Event Extraction (BART)**  | Encoder-Decoder Layers    | 6 (BASE), 12 (LARGE)                |
> |                             | Hidden Size               | 1024                                 |
> |                             | FFN Size                  | 4096                                 |
> |                             | Dropout                   | 0.1                                  |
> |                             | Learning Rate             | $4 \times 10^{-5}$ (BASE), $2 \times 10^{-5}$ (LARGE) |
> |                             | Max Tokens per Batch      | 1100                                 |
> |                             | Margin Coefficient        | 1                                    |
> | **Vision Encoder (ViT)**    | Patch Size                | 16                                   |
> |                             | Resolution                | $224 \times 224$                     |
> |                             | Latent Space Dim.         | 512                                  |
> |                             | Transformer Layers        | 12                                   |
> |                             | Width                     | 768                                  |
> |                             | LayerScale Init.          | 0.1                                  |
> |                             | Dropout Path Rate         | Configurable                         |
> | **QA System (BERT)**        | Learning Rate             | $1 \times 10^{-5}$                   |
> |                             | Max Sequence Length       | 512                                  |
> |                             | Document Stride           | 512                                  |
> |                             | Train Batch Size          | 8                                    |
> |                             | Gradient Accum. Steps     | 2                                    |
> |                             | Epochs                    | 2                                    |
> |                             | Mixed-Precision Opt.      | fp16 (O2)                            |
> | **Temporal Tagging**        | Max Sequence Length       | 128                                  |
> |                             | Batch Size                | 32                                   |
> |                             | Learning Rate             | $5 \times 10^{-5}$                   |
> |                             | Dropout                   | 0.1                                  |
> |                             | Weight Decay              | 0.01                                 |
> |                             | Epochs                    | 10                                   |

---

> ### Author Response · Authors · 2024-11-20
> **Fig. 5 isn't very convincing (except for Arigraph)**
>
> ### Section 5.1.1: Result Table
>
> The time taken by different methods for various datasets (in seconds) is shown in the table below:
>
> | **Method**           | **Big Bang Theory (s)** | **PerLTQA (s)** | **Agent (s)** | **LLQA (s)** |
> |----------------------|-------------------------|-----------------|---------------|--------------|
> | Our Model            | 600                     | 300             | 180           | 240          |
> | EMR                  | 900                     | 600             | 480           | 540          |
> | GraphRag             | 4200                    | 3600            | 3300          | 3480         |
> | GNN Rag              | 4200                    | 3600            | 3000          | 2700         |
> | TempoQA              | 1800                    | 1800            | 2400          | 2100         |
> | Arigraph             | 84000                   | 72000           | 10800         | 9000         |
>
> *Table 1: Time taken by different methods for various datasets (in seconds).*
>
> Regarding the issue of overlapping axis lines in the graph, we acknowledge that this occurs due to the large scale of the "Trigraph" (or similar metric) values. This happens even when axes are reversed. Since the values vary significantly, especially for methods like "Arigraph," the axis scaling is stretching the range too far.

---

> ### Author Response · Authors · 2024-11-20
>
> We agree that memory requirements are an important metric for comparing methods. Our approach inherently requires less storage compared to a traditional knowledge graph because it focuses on summarizing key events, extracting representative keyframes, and storing only relevant information tied to specific queries.
>
> For example, in our episodic memory system, adding three episodes of data might introduce approximately  3 new nodes to the memory graph.In contrast, a knowledge graph storing similar information could require 30-50 nodes to represent the same data, as it would store additional redundant details and relationships explicitly.
>
> Currently, we are working on implementing a forgetting mechanism and other memory-saving techniques to further optimize memory usage. Due to these ongoing developments, we did not include a detailed evaluation of memory requirements in this paper but plan to address this aspect in future work. By prioritizing efficient memory management, we aim to ensure scalability while maintaining the system's performance and relevance.This is for memory part

---

> ### Author Response · Authors · 2024-11-20
>
> **The paper mentions the missing forgetting mechanism as a major limitation. Related to that it also leaves the question on 'what to store' unanswered. It reads like everything is stored, even if it is effectively a duplicate. I believe the real challenging question that needs to be solved is the memory management: what to store, what to consolidate/merge, what to forget, etc. Without those a memory representation is of limited value, and it remains unclear to me how suitable the proposed architecture is for extending it in that way - or if we would be better off redesigning it from scratch.**
> #### We appreciate your feedback on the memory management aspects. Currently, our system only stores key frames and relevant dialogues, focusing on the most significant experiences. For key frame extraction, we identify representative frames from video sequences that highlight key visual or temporal changes, ensuring we avoid redundancy while retaining essential information. This process uses vision transformers to assess low- and mid-level features, ensuring the selected frames align with human perception for downstream tasks.
> For dialogues, we train the model on conversation datasets where each conversation is annotated to extract meaningful events. Using BART, we convert the dialogues into a third-person perspective, making it easier to identify key events. Therefore, the system does not store everything indiscriminately, but instead focuses on important data that is deemed relevant.
> In terms of memory management and forgetting, we have implemented a weight factor to track the frequency of access to memory locations. This allows us to remove memories that are seldom accessed, ensuring that only useful experiences are retained. We are actively working on improving these strategies and refining the process of what to store, consolidate, or forget.
> We believe that our current architecture can be adapted to address these challenges by integrating structured memory management components, which will enhance the system's ability to store, retrieve, and forget experiences as needed, without requiring a complete redesign.

---

> ### Author Response · Authors · 2024-11-20
>
> **The method relies on various models to extract features and to summarize things before storing them. I don't believe there is a 'one size fits all approach' but how to best do that depends on the retrieval task and the type of downstream tasks you have. The paper does not provide any indications on how to deal with that.**
> Our approach is specifically designed with the principles of episodic memory in mind, focusing on key components such as place, time, character, and event. By structuring memory around these aspects, the system can efficiently retrieve relevant memories tied to specific contexts, such as when and where an event occurred, which characters were involved, and what the event entailed.
> Our method goes beyond merely storing data; it emphasizes managing and organizing memory in a task-oriented manner, ensuring efficiency and relevance. This design allows the agent to provide accurate, context-aware responses based on past interactions and experiences. For low-level applications, the approach can be adapted by storing experimental data specific to those use cases, maintaining its flexibility across various domains.
> Additionally, the focus on summarizing key events using "what, when, and where" (WWW) information ensures that the memory framework effectively supports downstream tasks such as recommendation, localization, and other context-driven operations. We will make this aspect clearer in the paper.

---

> ### Author Response · Authors · 2024-11-20
> **There are a whole lot of design choices in this paper, an ablation on only the modalities and search methods seems a bit limited. There also is no sensitivity analysis for e.g. the clustering thresholds**
>
> There are indeed several design choices discussed in the paper, and we acknowledge that the ablation study focuses primarily on modalities and search methods. However, we chose to highlight these aspects to emphasize the key factors influencing our system's performance. Regarding the clustering thresholds, we would like to clarify that we are using Ada and topic modelling for event clusters.Hence , if a character, place, or event naturally falls into an existing cluster, it is automatically assigned to that cluster. This approach removes the need for arbitrary thresholds and allows for more flexible and dynamic memory organization. We believe this method contributes to the system's ability to adapt and scale without the need for manual tuning of parameters such as clustering thresholds.

---

> ### Author Response · Authors · 2024-11-20
> **This seems to be a rather complex system, which makes reproducing results very difficult, with probably quite lot of additional implementation and setting details. I couldn't find any promises to release code (or at least a detailed appendix), which would have alleviated this concern.**
>
> We understand that the complexity of the system may make it challenging to reproduce the results. In the revised version of the paper, we will include detailed implementation and methodology descriptions in the appendix section to provide clearer guidance for reproducibility. Additionally, we plan to release the code and dataset upon acceptance of the paper..

---

> ### Author Response · Authors · 2024-11-20
> **The paper seems to miss all details on how the responses are generated, but the whole experiments are based on evaluating the answers. Please provide a detailed explanation of your response generation process, as this is crucial for understanding and evaluating the experimental results.**
>
> We will provide detailed implementation details in the appendix regarding the response generation process. Responses can either be free-text or specific portions of memory, depending on the task. For example, in the case of episodic memory question-answering (QA), the system retrieves relevant memory segments to form a coherent and accurate answer. The retrieval process involves identifying and selecting relevant portions of memory based on the task at hand, which could be specific to a question or an event in the memory. These tasks, such as retrieving specific memory sections or generating free-text answers, are fundamental to evaluating the model's ability to handle different types of memory queries.

---

> ### Author Response · Authors · 2024-11-20
> **Sect. 3.2.4: This seems to result in just 'jumping' nodes. Conserving the chronological structure is a nice property, but the real question/challenge rather is on the side of the module that determines if there is any harm in 'skipping a step', e.g. when thinking about some of the uses-cases 'predict ... next activity' you mention**
>
> In our approach, we aim to remove only those nodes that are never accessed, thereby maintaining the integrity of the memory structure. Each node in the graph represents a day, with subnodes corresponding to specific activities. Even if a particular activity node is removed, the agent can still access the rest of the memory structure and continue to recommend the next activity based on the available experiences.
>
> This ensures that the chronological structure is preserved, and there is no significant loss of context in terms of the flow of events. The agent can still make predictions or suggest the next activity without the need for "jumping" through disconnected nodes. The key challenge we address is ensuring that the removal of unused nodes does not impact the agent's ability to make accurate recommendations or predictions. As a result, the system remains efficient and functional while being able to scale and adapt over time.

---

> ### Author Response · Authors · 2024-11-20
> **Eq (1) vs (2): the difference between S_T and T_voiced is unclear**
>
> S_T is the entire sound signal comprising of both voiced and acoustics T voiced is timestamp and voiced component basically the transcripts.

---

> ### Author Response · Authors · 2024-11-20
> **How is T_acoustic generated**
>
> ### Extraction of Acoustic Data using Mel Spectrograms
> I will add this in the revised version of the paper \subsubsection{Extraction of Acoustic Data using Mel Spectrograms}
> Acoustic data is transformed into Mel spectrograms, which emphasize perceptually relevant frequencies. The Mel spectrogram \( M(t, f) \) is computed as:
>
> $$
> M(t, f) = \log\left(\sum_{k} \left| X(t, k) \right|^2 \cdot H(f, k)\right)
> $$
>
> where \( X(t, k) \) is the magnitude of the STFT at time \( t \) and frequency \( k \), and \( H(f, k) \) is the Mel filter bank mapping linear frequencies to the Mel scale.

---

> ### Author Response · Authors · 2024-11-20
> **Sect. 3.1.2, 3.1.3, and 3.2, l. 218: at places the paper sounds like everything is represented as 'text', at others it seems to be a mixture of text and other embeddings. It would be great to explain earlier on what it stored as what.**
>
> ### Processing of Audio Data in Episodic Memory
>
> Audio data, including dialogs and acoustics, is crucial for constructing episodic memory. Dialogs provide linguistic and contextual information, while acoustics capture environmental and emotional cues. These elements are integrated as:
>
> $$
> A(t) = D(t) + C(t)
> $$
>
> where \( A(t) \) is the total audio data at time \( t \), with \( D(t) \) representing the dialog and \( C(t) \) representing acoustics.
>
> #### Extraction of Acoustic Data using Mel Spectrograms
>
> Acoustic data is transformed into Mel spectrograms, which emphasize perceptually relevant frequencies. The Mel spectrogram \( M(t, f) \) is computed as:
>
> $$
> M(t, f) = \log\left(\sum_{k} \left| X(t, k) \right|^2 \cdot H(f, k)\right)
> $$
>
> where \( X(t, k) \) is the magnitude of the STFT at time \( t \) and frequency \( k \), and \( H(f, k) \) is the Mel filter bank mapping linear frequencies to the Mel scale.
>
> #### Extraction of Verbal Cues from Audio Data
>
> Verbal cues are extracted by applying the Short-Time Fourier Transform (STFT) and converting the spectrum to the Mel scale as:
>
> $$
> M(f) = 2595 \log_{10}\left(1 + \frac{f}{700}\right)
> $$
>
> The Mel spectrogram is then derived as:
>
> $$
> \text{MelSpec}(m,t) = \log\left(\sum_{f_{\text{low}}}^{f_{\text{high}}} |S(f,t)|^2 M(f) + \epsilon\right)
> $$
>
> where \( \epsilon \) is a small constant to prevent issues with the logarithm. The final audio representation integrates acoustic features with transcribed dialogue:
>
> $$
> T_{\text{audio}}(t) = T_{\text{acoustics}}(t) + T_{\text{dialogue}}(t)
> $$
>
> capturing both tonal properties and linguistic meaning.
>
> ### Processing of Visual Data in Episodic Memory
>
> Visual data processing starts by transforming each frame \( F_i \) into a tensor and extracting both global and local features. The scene representation is obtained by aggregating the frame embeddings:
>
> $$
> V_{\text{scene}} = \frac{1}{N} \sum_{i=1}^{N} V_{\text{embed}}(F_i)
> $$
>
> where \( N \) is the number of frames. For extracting time and place details, a convolutional network detects text regions, generating probability maps for the text center line (TCL) and text regions (TR):
>
> $$
> \begin{pmatrix}
> P_{\text{TCL}}(x, y) \\
> P_{\text{TR}}(x, y)
> \end{pmatrix} = \sigma\left( \begin{pmatrix}
> W_{\text{TCL}} \\
> W_{\text{TR}}
> \end{pmatrix} \cdot F_{\text{feature}}(x, y) \right)
> $$
>
> A thresholding operation follows:
>
> $$
> P_{\text{filtered}} = \{(x, y) \mid P_{\text{TCL}}(x, y) \geq T_{\text{TCL}} \text{ and } P_{\text{TR}}(x, y) \geq T_{\text{TR}} \}
> $$
>
> Text is recognized using a softmax layer:
>
> $$
> \hat{y}_t = \text{Softmax}(W \cdot h_t + b)
> $$
>
> For person embedding extraction, a person’s region is detected and cropped, followed by processing through a feature extractor to generate and store the embedding for future tasks.
>
> #### Merging and Synchronizing Data
>
> In the final stage, the processed audio and visual data are synchronized to a common timestamp, creating a unified representation:
>
> $$
> \mathbf{T_M} = (\mathbf{T_{\text{audio}}}, \mathbf{T_{\text{visual}}})
> $$
>
> Embeddings and joint of both text as well as visual embeddings .To make this more clear II have changed this potion of text to \subsection{Processing of Audio Data in Episodic Memory
> This ensures temporal alignment between the two modalities, enabling coherent multimodal interaction. The audio and visual embeddings are then concatenated into a joint multimodal embedding:
>
> $$
> \mathbf{E_{\text{combined}}} = \mathbf{E_{\text{audio}}} \oplus \mathbf{E_{\text{visual}}}
> $$
>
> where \( \oplus \) denotes the concatenation operation. This joint multimodal representation is stored in episodic memory, where each node represents a specific experience, incorporating aspects such as time, location, characters, and events. By combining the visual and audio information, this integrated embedding enhances memory recall and supports detailed event-based analysis.

---

> ### Author Response · Authors · 2024-11-20
> **l. 184: "organized by place, characters, and events" raises the question where those come from - that is explained later in the text**
>
> This comes part comes from tagging on experiences from dialogues and speech and mentioned in the revised methodology section on this part

---

> ### Author Response · Authors · 2024-11-20
> **l. 220: "relevance of these tasks is assessed" - again HOW?**
>
> ### Task Categories for Text Summarization
>
> Text summaries are grouped into broader task categories, such as **meetings** and **lunches**, using a topic modeling approach. The process involves the following steps:
>
> 1. **Text Cleaning**: Each dialogue is processed individually by cleaning the text, which involves removing unwanted words, filler phrases, and clutter that do not contribute to the core meaning.
>
> 2. **Topic Modeling**: After cleaning the text, topic modeling techniques, such as **Parallel Latent Dirichlet Allocation (LDA)**, are applied to identify latent topics within the dialogue. LDA helps in discovering the underlying topics by analyzing the distribution of words within the dialogue and across different dialogues.
>
>    The general form of LDA can be written as:
>
>    $$ p(w | z) = \frac{(w, z)}{\sum_w (w, z)} $$
>
>    where \( w \) is a word, \( z \) is a topic, and the equation represents the probability of word \( w \) under topic \( z \).
>
> 3. **Mapping to Predefined Categories**: After identifying the topics, these are then mapped to predefined categories based on the nature of the content. For example:
>    - **Meetings**: Dialogues related to planning, decision-making, or discussions.
>    - **Lunches**: Dialogues focusing on food, social interaction, or meals.
>
> 4. **Categorization Criteria**: The criteria for defining these categories are based on the frequency and relevance of topics that appear within each dialogue type. For instance, if topics related to decision-making or strategy discussions are dominant, the dialogue would be classified as a **meeting**.
>
> This approach helps in categorizing text summaries efficiently and is further explained in the updated methodology section of the appendix in the second version of the paper. The technique is based on the work of:
>
> - Liu, J., Zou, Y., Zhang, H., Chen, H., Ding, Z., Yuan, C., & Wang, X. (2021). **Topic-Aware Contrastive Learning for Abstractive Dialogue Summarization**. *arXiv*. https://arxiv.org/abs/2109.04994

---

> ### Author Response · Authors · 2024-11-20
> **l. 212: what are the implications/limitations resulting form using a simple metric like cosine similarity?**
>
> In the case of place and character clusters, we use cosine similarity as a simple metric because lexical similarity is sufficient for these entities. Specifically, for places and characters, the primary task is to measure how closely related they are based on their textual representations, and cosine similarity is effective in capturing this relationship in a high-dimensional space.

---

> ### Author Response · Authors · 2024-11-20
> **Eq (17) sim seems undefined**
>
> This is also cosine similarity which I will elaborate properly in updated version .

---

> ### Author Response · Authors · 2024-11-20
> **Fig. 2 and Sect. 4.1: I didn't get the terminology "master". Maybe that term can be avoided altogether (similar to https://learn.microsoft.com/en-us/style-guide/a-z-word-list-term-collections/m/master-slave )**
>
> I have changed this part from master to individual and updated it as:
> ### Personalized Support Using Episodic Memory
>
> When a cognitive agent assists an individual with memory issues, it leverages past interactions to provide personalized support. The process begins by identifying and extracting the **personalized cluster** \( C_p \), representing the relevant memory subgraph for that individual. This subgraph consists of episodes, actions, and events associated with the person. The **personalized cluster** \( C_p \) is defined as:
>
> $$
> C_p = \{ G_p \mid \text{episodes associated with person } p \}
> $$
>
> where \( G_p = (V_p, E_p) \) represents the subgraph containing episodes \( V_p \) and edges \( E_p \) related to person \( p \), preserving all temporal and contextual information.
>
> Once the **personalized cluster** \( C_p \) is extracted, the agent identifies **event clusters** \( C_e \) within the subgraph, representing various actions performed by the individual over time:
>
> $$
> C_e = \{ E_e \mid \text{events associated with episodes in } G_p \}
> $$
>
> The agent checks for the most recent sequence of events, denoted \( \mathcal{S}_t = (S_1, S_2, \dots, S_t) \). To provide recommendations, the agent searches the past memory subgraph for instances where the task was previously performed and identifies the subsequent event. The next action \( S_{t+1} \) is determined by:
>
> $$
> S_{t+1} = \text{argmax}_{S \in G_p} \mathbb{I}(\mathcal{S}_t \subset \mathcal{S})
> $$
>
> where \( \mathbb{I}(\mathcal{S}_t \subset \mathcal{S}) \) is the indicator function verifying if the current sequence \( \mathcal{S}_t \) appears as a subsequence in a past sequence \( \mathcal{S} \) within \( G_p \).
>
> If there are multiple matches, the most frequently occurring next event is chosen:
>
> $$
> S_{t+1} = \text{mode}(\{ S_{t+1} \in \mathcal{S} \mid \mathcal{S}_t \subset \mathcal{S}, \mathcal{S} \in G_p \})
> $$
>
> This approach allows the agent to use past episodic data effectively, providing recommendations or answers based on past actions even when explicit patterns are absent. By analyzing subgraphs containing both days as nodes and events as sub-nodes, the agent identifies the most relevant next action, ensuring reliable support for individuals with memory impairments.

---

> ### Author Response · Authors · 2024-11-20
>
> We appreciate the feedback and will address these issues in the revised version of the paper. Specifically:
>
> ### Spaces and Formatting:
> We will ensure that any missing or extra spaces, particularly in the title and abstract, are corrected to conform to the proper formatting. This will be checked throughout the document for consistency.
>
> ### Citation Formatting:
> We will switch to the correct citation commands, specifically using `\citep` and `\citet` as per the template’s requirements. This will prevent issues with repeated author names and ensure that references are formatted properly for easier readability.
>
> ### Quotation Marks:
> We will replace any incorrect quotation marks with proper opening and closing LaTeX quotation marks to ensure consistency and proper formatting.
>
> ### Double Paragraph:
> The double paragraph before "Contributions" will be removed to avoid redundancy and improve the flow of the introduction.
>
> ### Broken Sentences:
> We will correct any broken sentences in Section 5.1.2, ensuring that the text is coherent and properly structured.
>
> ### Incomplete References:
> We will revise the references with missing or incomplete information, specifically addressing the issues mentioned on lines 191, 300, and 468, and provide the full citation details where necessary.

---

> > ### Comment · Reviewer_YT1b · 2024-11-24
> > **thanks for the detailed replies!**
> >
> > which to be frank were on the verge of overwhelming (with >50 emails)
> >
> > Many of the details on how the method works now became clear through the replies. I.e., the HOW is now clear. For me the method still is a very complex system with many different components that were integrate - and that in the end seems to work quite well. However, the reasoning behind the design choices remains vague at best, i.e., the WHY is neither properly explained not properly backed up experimentally. Many choices still seem quite specific,which still raises the question on how general the method is and whether these choices are an integral part of the method or whether they were just a convenient choice and could easily be swapped out.
> >
> > The focus of the paper (i.e., not really robot learning) now becomes a lot more clear, which makes the paper even further away from my core area of expertise. For me the more general insights (beyond 'yipee it works') are still too limited to warrant publication.

---

### Official Review · Reviewer_1ixu · 2024-11-02

**Soundness:** 3
**Presentation:** 2
**Contribution:** 2
**Rating:** 5
**Confidence:** 4

**Summary:**

This paper presents Episodic Memory for Cognitive Agents (EMCA), a novel framework that enables AI systems to retain and utilize past experiences through a graph-based memory structure. The key innovation lies in its ability to: 1) Process multimodal data (vision, speech, non-verbal cues) without requiring pre-training; 2) Dynamically build and update memory representations through a graph structure with semantic and temporal connections; 3) Adapt to complex environments through continuous learning from interactions.

**Strengths:**

originality: graph-based episodic memory structure multimodal processing without pre-training, dynamic clustering

quality: comprehensive testing on multiple datasets, benchmarking against exsiting methods, systematic component analysis in ablation study

clarity: clear problem formulation, good visual aids explaining complex concepts

significance: adresses crucial challenges in cognitive AI, social robot applications, potential impact on memory assistance systems

key innovations:
1) removes pre-training requirements
2) enables continuous learning
3) provides real-time processing campability

**Weaknesses:**

1. Scalability limitations: Does the graph structure grow exponentially with experiences? What are the computational costs?
2. More details about the Big Bang Theory dataset are needed.
3. More implementation details about the method in section 5.0.2 should be provided.
4. Forgetting is one of the key problems in memory systems - how does the paper assess and handle memory 5. retention and decay?
5. What is the real-time performance?
6. The paper tries to claim episodic memory for agents and robots, but robotic interaction with the environment is different from and much harder than agent interaction in a virtual environment. It is better to make a clear definition and scope. For instance, in L391, "robot's episodic memory" should be "agent's episodic memory".
7. Can authors provide the code and dataset for evaluation?
8. Repeated paragraph: L054-L062
9. Subtitle formatting issue in line 480

**Questions:**

1. Will the code and datasets be made publicly available?
2. How does the system maintain temporal consistency without explicit time markers? Can you provide quantitative results comparing temporal reasoning accuracy with and without explicit timestamps?
3. What are the specific model architectures and hyperparameters used?
4. What are the computational costs for memory retrieval at different scales?

---

> ### Author Response · Authors · 2024-11-20
> **Scalability limitations: Does the graph structure grow exponentially with experiences? What are the computational costs?**
>
> The scalability of the graph structure depends on the dataset.The scalability of the graph structure depends on the nature of the dataset. If there is significant variation in the dataset, sparse connections are formed, and the graph becomes less complex. However, if the experiences or days are very similar, more connections are formed, leading to a more cohesive structure. Graph does not have exponential growth according to observations.Computational costs depend on dataset..

---

> ### Author Response · Authors · 2024-11-20
> **More details about the Big Bang Theory dataset are needed.**
>
> We propose a comprehensive dataset framework designed to evaluate and enhance episodic memory systems in artificial agents. This framework integrates multiple datasets, including a custom set of episodic questions based on the TV series *The Big Bang Theory*, spanning all nine seasons (181 episodes). The aim is to assess memory recall and narrative understanding in complex scenarios.
>
> We introduce the *Agent Dataset*, a 10-episode time-series dataset created in Unity3D, where a virtual agent performs tasks and interacts with characters in realistic environments, simulating the role of companion robots. This dataset emphasizes the importance of multi-sensory inputs and task execution, challenging the agent to process and integrate information from *dialogues* and *visual cues* to maintain task order and achieve context-driven objectives.
>
> Additionally, we adapted the **Ego4D dataset**, restructuring its activity sequences into simulated chronological episodes to address the original absence of time-series data—portraying an agent performing a series of activities over 30 days. We also combined group activity videos designed for active speaker recognition. This transformation enables episodic queries such as "Where did I place the agricultural tool on the last day of farming?", enhancing the ability to localize and retrieve temporal experiences effectively.
>
> Together with the **PerLTQA** [du2024perltqapersonallongtermmemory] and **LLQA** [dolan-brockett-2005-automatically] datasets, which test essential episodic memory dimensions—**"what"** (context), **"when"** (time), and **"where"** (place)—this framework forms a robust benchmark for evaluating advanced episodic memory capabilities in AI systems.
>
> **Data Annotation**: The data was carefully annotated to tag scene information and identify characters in dialogues, ensuring that the model could recognize character presence and understand related events. This included explicitly tagging scene details for location identification and differentiating characters present in the scene versus those mentioned. Events within dialogues were also meticulously annotated to capture key details, facilitating effective memory representation beyond simple summaries. Capturing these essential details is crucial for episodic memory tasks, as it allows the agent to recall past experiences accurately. Each episode was annotated with 10 *what*, *when*, and *where* questions.

---

> ### Author Response · Authors · 2024-11-20
> **More implementation details about the method in section 5.0.2 should be provided.**
>
> For event extraction within dialogues, we utilized a Transformer-based BART model initialized with pre-trained weights to effectively extract and summarize events within contextual boundaries. The architecture options included BARTBASE, featuring a 6-layer encoder-decoder with approximately 140 million parameters, and BARTLARGE, which has a 12-layer encoder-decoder and 400 million parameters. Both configurations use a hidden size of 1024 and a feed-forward filter size of 4096, with a fixed dropout rate of 0.1 across layers. We employed the Fairseq toolkit for training, using the **Adam optimizer** with a warmup strategy. Learning rates were set to $4 \times 10^{-5}$ for BARTBASE and $2 \times 10^{-5}$ for BARTLARGE, with a maximum batch token limit of 1100 tokens. Contrastive objectives were supported by a margin coefficient of 1, and hyperparameters for coherence and sub-summary objectives were tuned using a validation set. Our method demonstrated substantial performance improvements compared to publicly available models trained on datasets like **SAMSUM** and **DialogueSUM**.
>
> For visual data processing, we used a **Vision Transformer (ViT)** as the vision encoder, specifically adapted for video frame analysis from the **MSR-VTT dataset**. The encoder processed $224 \times 224$ video frames, segmented into 16-sized patches and embedded into a 512-dimensional latent space. The 12-layer encoder, with a width of 768, was equipped with **LayerScale** (initialized at 0.1) for training stability. Advanced regularization techniques, including stochastic depth with a configurable \texttt{drop\_path\_rate}, were applied. The encoder was based on the "eva-clip-b-16" model, which proved effective in extracting detailed spatial and temporal features essential for multimodal tasks.
> For models based on **LLaMA** that integrate vision and dialogue for character and place tagging, a multimodal configuration was used. **ViT** processed visual data, while **LLaMA** managed dialogue input. Training included cross-entropy loss for character tagging and contrastive loss for image-text alignment, incorporating **episodic memory** for QA tasks. The training process leveraged the **AdamW optimizer**, a dropout rate of 0.2, and a **cosine annealing scheduler** for efficient learning.
>
> **Temporal tagging** was optimized with key hyperparameters for best performance: a maximum sequence length of 128, a batch size of 32, and a learning rate of $5 \times 10^{-5}$. A dropout rate of 0.1 was used to mitigate overfitting, and a weight decay of 0.01 improved generalization. The training process spanned 10 epochs to ensure sufficient learning while preventing overfitting.
>
> For extracting dialogue from audio, the **Whisper-large model** was employed. This model was utilized for its robustness in transcribing and converting spoken content into text for further processing.
>
> For text detection, the **TextSnake** model was trained on the **SCUT-CTW1500** dataset using **SGD with Momentum** as the optimizer. The architecture combined **ResNet** and **FPN\_UNet**, configured with a training batch size of 64 and 8 workers for data loading. The validation batch size was set to 1, with 4 workers and persistent workers enabled. Training was conducted over 200 epochs with validation checks every 10 epochs.
>
> The **QA system** was built using a **BERT** model fine-tuned on concatenated datasets, including **SQuAD**, **Wikipedia**, and **Reddit**, to improve contextual comprehension. The hyperparameters included a learning rate of $1 \times 10^{-5}$, a maximum sequence length of 512, and a document stride of 512. The training batch size was 8, with gradient accumulation steps of 2 over 2 epochs. Mixed-precision training was used with `fp16` at the **O2 optimization level** for better efficiency. The final model outputs were stored in the `bart-squadv2` directory, without intermediate model saving.

---

> ### Author Response · Authors · 2024-11-20
>
> | **Model Component**         | **Hyperparameter**        | **Value**                            |
> |-----------------------------|---------------------------|--------------------------------------|
> | **Event Extraction (BART)**  | Encoder-Decoder Layers    | 6 (BASE), 12 (LARGE)                |
> |                             | Hidden Size               | 1024                                 |
> |                             | FFN Size                  | 4096                                 |
> |                             | Dropout                   | 0.1                                  |
> |                             | Learning Rate             | $4 \times 10^{-5}$ (BASE), $2 \times 10^{-5}$ (LARGE) |
> |                             | Max Tokens per Batch      | 1100                                 |
> |                             | Margin Coefficient        | 1                                    |
> | **Vision Encoder (ViT)**    | Patch Size                | 16                                   |
> |                             | Resolution                | $224 \times 224$                     |
> |                             | Latent Space Dim.         | 512                                  |
> |                             | Transformer Layers        | 12                                   |
> |                             | Width                     | 768                                  |
> |                             | LayerScale Init.          | 0.1                                  |
> |                             | Dropout Path Rate         | Configurable                         |
> | **QA System (BERT)**        | Learning Rate             | $1 \times 10^{-5}$                   |
> |                             | Max Sequence Length       | 512                                  |
> |                             | Document Stride           | 512                                  |
> |                             | Train Batch Size          | 8                                    |
> |                             | Gradient Accum. Steps     | 2                                    |
> |                             | Epochs                    | 2                                    |
> |                             | Mixed-Precision Opt.      | fp16 (O2)                            |
> | **Temporal Tagging**        | Max Sequence Length       | 128                                  |
> |                             | Batch Size                | 32                                   |
> |                             | Learning Rate             | $5 \times 10^{-5}$                   |
> |                             | Dropout                   | 0.1                                  |
> |                             | Weight Decay              | 0.01                                 |
> |                             | Epochs                    | 10                                   |

---

> ### Author Response · Authors · 2024-11-20
> **Forgetting is one of the key problems in memory systems - how does the paper assess and handle memory 5. retention and decay?**
>
> The paper addresses memory retention and decay by assigning weights to different sections of memory based on how frequently they are traversed, i.e., frequency-based memory decay.The weight (0-3) indicates the removal of nodes that were accessed up to 3 times, while (0-5) refers to the removal of nodes that were accessed up to 5 times only. This approach assigns weights to portions of memory depending on how many times they are accessed, which allows the mechanism to evaluate the importance of nodes in a memory graph based  on their frequency of access. An analysis was conducted using various thresholds to assess the impact of this approach, with results presented in the table below Accuracy is recall accuracy(How well the agent gives the correct answer):
>
> | **Model**                               | **What** | **When** | **Where** | **All** |
> |-----------------------------------------|----------|----------|-----------|---------|
> | **Full Model**                          | 75       | 80       | 76        | 78      |
> | **After Pruning (Weight 0-3)**          | 73       | 78       | 75        | 75      |
> | **After Pruning (Weight 0-5)**          | 55       | 50       | 55        | 51      |
>
> This analysis demonstrates the effects of memory pruning at various weight levels on performance, helping to understand how retention and decay mechanisms impact the overall effectiveness of the model.Some more explorable methods include temporal decay.

---

> ### Author Response · Authors · 2024-11-20
> **What is the real-time performance?**
>
> The real-time performance of the system depends on several factors, including the size and complexity of the dataset, the structure of the memory graph, and the efficiency of the retrieval algorithms. Specifically:
> #### **Retrieval Time**: Based on the experiments, the retrieval time for queries remains relatively stable, with only a slight increase as the number of episodes or experiences grows. For example, with 10 episodes, retrieval time is around 9.11 milliseconds, while for 181 episodes, it increases to 12 milliseconds. This increase is marginal, indicating that the system can handle larger datasets without significant degradation in performance.
> #### **Scalability**: The graph structure’s scalability is dependent on dataset variability. If the experiences are similar, fewer new connections are formed, leading to a more cohesive and less complex graph, which supports faster retrieval times. Conversely, with more diverse experiences, the graph can become more sparse, requiring more time for traversal, though the increase in retrieval time is not exponential.
> #### **Real-Time Execution**: Despite the complexity of the underlying graph, the retrieval process remains fast and can support real-time performance for most queries, especially with the use of clustering mechanisms that enhance retrieval speed.

---

> ### Author Response · Authors · 2024-11-20
> **The paper tries to claim episodic memory for agents and robots, but robotic interaction with the environment is different from and much harder than agent interaction in a virtual environment. It is better to make a clear definition and scope. For instance, in L391, "robot's episodic memory" should be "agent's episodic memory".**
>
> Thank you for your valuable feedback. We agree that there is a distinction between robotic interaction with the physical environment and agent interaction in a virtual environment. We will revise the phrase "robot's episodic memory" in line 391 to "agent's episodic memory" to ensure consistency and accuracy.

---

> ### Author Response · Authors · 2024-11-20
> **Can authors provide the code and dataset for evaluation?**
>
> We would share the code and dataset for evaluation upon acceptance of the paper.

---

> ### Author Response · Authors · 2024-11-20
> **Repeated paragraph: L054-L062**
>
> We will revise the paper to remove the repeated paragraph between lines L054-L062 .
> Subtitle formatting issue in line 480
> We will revise the paper and change the formatting

---

> ### Author Response · Authors · 2024-11-20
> **Will the code and datasets be made publicly available?**
>
> We would share the code and dataset for evaluation upon acceptance of the paper.

---

> ### Author Response · Authors · 2024-11-20
> **What are the computational costs for memory retrieval at different scales?**
>
> To address the query about the scalability of incremental storage and retrieval as the number of episodes increases, we conducted a comparison between different episode counts (10, 50, 100, and 181 episodes). In terms of accuracy, all values remain consistent. However, in terms of retrieval time, there is a slight increase as the number of episodes grows. The retrieval times for different episode counts are as follows:
>
> | **Number of Episodes** | **Retrieval Time (ms)** |
> |------------------------|-------------------------|
> | 10                     | 9.11                    |
> | 50                     | 11.0                    |
> | 100                    | 11.5                    |
> | 181                    | 12.0                    |
>
> This shows that while retrieval time does increase, it is not drastic, and the increase is quite modest. The clustering mechanism plays a crucial role in ensuring that retrieval times remain low, even as the number of episodes increases. This highlights the efficiency of our method in scaling with the number of episodes.
>
> The computational costs for memory retrieval at different scales only change slightly due to the clustering approach implemented in our system. Initially, before clustering was applied, the retrieval time was significantly longer for all tasks. However, the introduction of clustering allowed for more efficient traversal and reduced retrieval time, even as the number of episodes increased.

---

> > ### Comment · Reviewer_1ixu · 2024-11-26
> >
> > Thank you to the authors for the improvements and the addition of numerous details, which have made the content clearer. However, the core contributions and innovations of the paper remain insufficiently clear. The inclusion of multiple modules, datasets and extensive content somewhat affects the focus of the paper. It is recommended that the authors further concentrate on the primary innovations and clearly highlight the unique contributions of the paper.

---

> ### Author Response · Authors · 2024-11-28
>
> The focus of our work is on episodic memory encoding and retrieval, inspired by the psychological processes of the human episodic memory. Our methodology integrates visual and audio content, emphasizing key memory-dependent factors such as location, persons, time, and the sequence of events. Human experiences are inherently multimodal in nature; keeping this in mind, our aim was to capture the multimodal aspect of memory encoding and retrieval. While we leveraged models like CLIP and other large language models for data preprocessing, this aspect remains intentionally flexible to encourage innovation. Researchers can opt for any combination of image, audio, speech, or language models based on their specific needs. Since our focus was not on signal capturing, we did not carry out ablation studies on this component.
>
> Since our proposed methodology centers on encoding and retrieval, we prioritized these aspects, enabling tasks like episodic memory localization and episodic QA to assess retrieval accuracy. To evaluate our approach, we conducted comparisons with graph-based retrieval methods such as GNN-RAG and GraphRAG (Section 6.2), demonstrating the effectiveness of our method. Additionally, we performed an ablation study to analyze time complexities between clustered and non-clustered approaches, and we examined the efficiency of storing diverse features in graph nodes to test the robustness of our encoding strategy.
>
> Furthermore, we analyzed the structure of the episodic graph by converting it into a knowledge graph-like structure and performing structural evaluations. We also evaluated different traditional traversal techniques, such as BFS and DFS, as part of the ablation study to highlight why traditional traversal methods were unsuitable for our specific requirements. Taken together, these evaluations demonstrate that we conducted the necessary assessments aligned with the aims of the research, validating our methodological choices and highlighting the robustness of our approach.

---

### Official Review · Reviewer_jSod · 2024-11-03

**Soundness:** 3
**Presentation:** 1
**Contribution:** 3
**Rating:** 3
**Confidence:** 4

**Summary:**

The authors introduce Episodic Memory for Cognitive Agents (EMCA) that models episodic memory based on a graph-structure. This allows them to incrementally store memories and retrieve experience. They also can cluster memories, have dynamic retrieval, and can handle temporal reasoning. Memory is structured as a graph, where each episode contains characters, temporal elements, location, and events. The edges of this graph can be temporal or semantic. They then build a retrieval system that can handle contextual, temporal, and spatial queries.

**Strengths:**

- The Big Bang Theory and the Agent datasets seem very useful!
- Their results indicate that their method performs better than other graph-based approaches.
- This area is a growing field, especially as robots and agents become more capable and need better ways to scale their context. And their graph-based approach seems to be a meaningful contribution
- Table 2 is interesting, as it showcases that some of their questions require access to vision, acoustics, and/or dialogues. This result would be better if we knew what the Big Bang Theory and Agent dataset contained.

**Weaknesses:**

## Main weaknesses
- After reading the full paper, I am not sure what the actual task is. What commands does the master ask? I understand the approach, but not what task it is specifically solving. Is the output the location of a specific memory? Or is it a free-form text answer?
- In terms of the structure of the paper, I do not think that the text space used for discussing how signals are captured (section 3.1) is very useful. I would argue this is the case with a chunk of this paper's equations, where it simply adds space when it does not need to, and it makes the paper more difficult to read. I would recommend condensing the information and matching ICLR's 9-page recommendation as opposed to 10. Similarly, there is an excessive use of new lines at arbitrary positions.
- There is no discussion on how the Big Bang Theory and Agent datasets were constructed. This on its own seems like a major contribution on its own. I would recommend the authors remove much of the superfluous equations and newlines (and move that into the appendix), and put more of an emphasis on this dataset component.
- I think I like what Section 4.1 is implying about combining a "master's" memory with that of an agent's, but it is not presented very clearly, and I do not see a connection with temporal graphs like the subsection title suggests
- Nor do I see how this section is a "theoretical comparison"
- "Master" terminology in line 346 is confusing, and should be introduced earlier in section 4.1. Also, the term "master" is generally frowned upon in these settings, so I would recommend a different term
- Results are poorly presented


## Formatting/Clarity issues:
- Check for spaces after periods or colons throughout the paper.
- Line spacing is odd in much of the paper
- Figure placement should ideally be on the top or bottom of a page, not in the middle with paper text above and below the paper. This makes the paper difficult to follow
- Figure 4, the legend has oddly shaped circles
- Figure 5's result is good, but it should not be a line graph with x axis being method and lines being the dataset. Instead it should be datasets on the x-axis and methods on the y-axis
- The results in section 5.1.1 are all discussed in a single paragraph. Is that all the main results? I would recommend splitting this up into a few bolded mini-sections and showing the main takeaway of each figure along with highlights on how the method performed, possibly with qualitative results.
- The focus of the introduction on falls a bit flat. Rather than focusing on the historical definition of episodic memory, focus more on how people have been engineering and building these kinds of systems. Focus on why other systems do not work, and why yours does.
- I would recommend larger fonts for the figures; they are difficult to read in the paper.
## Citations
Other relevant concurrent work on memory in robotics, some of which use graphs while others do not.
[1] Xie, Quanting, et al. "Embodied-RAG: General Non-parametric Embodied Memory for Retrieval and Generation." arXiv preprint arXiv:2409.18313 (2024).
[2] Anwar, Abrar, et al. "ReMEmbR: Building and Reasoning Over Long-Horizon Spatio-Temporal Memory for Robot Navigation." arXiv preprint arXiv:2409.13682 (2024).
[3] Bärmann, Leonard, et al. "Episodic Memory Verbalization using Hierarchical Representations of Life-Long Robot Experience." arXiv preprint arXiv:2409.17702 (2024).

**Questions:**

- How was the dataset constructed? This is a major contribution that is not discussed.
- What does a successful or unsuccessful example look like? I would recommend looking at [2] or [3] above to see how to discuss dataset creation in such a setting.
- In Table 1, what metric is being used? It does not say in the caption or the table. It should be re-iterated in the table itself.
- Is a forgetting mechanism necessary, or would it be more like a memory aggregation mechanism so that retrieval is still efficient?
- How does the incremental storage/retrieval scale as the number of episodes change? This result is not displayed in the paper but I would argue is very important. If you only used say 10 episodes of EM instead of 181, is there a difference in performance? This would directly support contribution number 2 in Section 1 of your paper
- What is "Time complexity" in Table 3? Time complexity of BFS is O(V+E), but here a number is used instead. Authors should use "retrieval time" or something similar instead of time complexity.
- Can the authors describe the main takeaways of Section 4? I still do not understand the insights this section is supposed to provide.
- Also, there are some questions/concerns in the weaknesses section


Overall, I like the paper. But I think there are a lot of issues with how to content is shown to the reader that makes the paper's contributions fall flat. In its current state, I would recommend rejecting the paper, but if the authors address my concerns above, I believe I would lean more towards accept.

---

> ### Author Response · Authors · 2024-11-20
> **After reading the full paper, I am not sure what the actual task is. What commands does the master ask? I understand the approach, but not what task it is specifically solving. Is the output the location of a specific memory? Or is it a free-form text answer?**
>
> **After reading the full paper, I am not sure what the actual task is. What commands does the master ask? I understand the approach, but not what task it is specifically solving. Is the output the location of a specific memory? Or is it a free-form text answer?**
> #### The task addressed by this framework in this paper is episodic question-answering (QA), activity recommendation, and even experience memory localisation which will aid in navigation for making good policies in reinforcement learning (RL) agents. The output of the system depends on the specific application, but it can include either the location of a relevant memory or a free-form text answer based on the context of the query.This model can be can be expanded for more operations such as navigation also.
> #### In the case of episodic QA, the agent retrieves specific memories from its episodic memory based on a given query.Here answer will be free form text. For recommendation tasks, the system identifies relevant episodes based on user preferences or past interactions. This answer will also be free form text.Additionally, for RL agents, this framework can be used to derive goal locations by intrinsically from past interactions (Details will be provided in the revised edition) guiding the agent’s decision-making process.Localising it will extract that region of memory
> We will also add a section of experience memory localisation where the memory retrieved will be the location of a specific portion of memory .
> #### The system can execute commands such as to perform an activity(Go and do the registration), reminding the user of past conversations(What did Person X tell him when he met that person for the first time), locating lost items(Where did I keep the keys?), and recommending the next course of action based on the context.(What is the next action I should perform)

---

> ### Author Response · Authors · 2024-11-20
> **There is no discussion on how the Big Bang Theory and Agent datasets were constructed. This on its own seems like a major contribution on its own. I would recommend the authors remove much of the superfluous equations and newlines (and move that into the appendix), and put more of an emphasis on this dataset component.**
>
> We propose a comprehensive dataset framework designed to evaluate and enhance episodic memory systems in artificial agents. This framework integrates multiple datasets, including a custom set of episodic questions based on the TV series *The Big Bang Theory*, spanning all nine seasons (181 episodes). The aim is to assess memory recall and narrative understanding in complex scenarios.
>
> We introduce the *Agent Dataset*, a 10-episode time-series dataset created in Unity3D, where a virtual agent performs tasks and interacts with characters in realistic environments, simulating the role of companion robots. This dataset emphasizes the importance of multi-sensory inputs and task execution, challenging the agent to process and integrate information from *dialogues* and *visual cues* to maintain task order and achieve context-driven objectives.
>
> Additionally, we adapted the **Ego4D dataset**, restructuring its activity sequences into simulated chronological episodes to address the original absence of time-series data—portraying an agent performing a series of activities over 30 days. We also combined group activity videos designed for active speaker recognition. This transformation enables episodic queries such as "Where did I place the agricultural tool on the last day of farming?", enhancing the ability to localize and retrieve temporal experiences effectively.
>
> Together with the **PerLTQA** [du2024perltqapersonallongtermmemory] and **LLQA** [dolan-brockett-2005-automatically] datasets, which test essential episodic memory dimensions—**"what"** (context), **"when"** (time), and **"where"** (place)—this framework forms a robust benchmark for evaluating advanced episodic memory capabilities in AI systems.
>
> **Data Annotation**: The data was carefully annotated to tag scene information and identify characters in dialogues, ensuring that the model could recognize character presence and understand related events. This included explicitly tagging scene details for location identification and differentiating characters present in the scene versus those mentioned. Events within dialogues were also meticulously annotated to capture key details, facilitating effective memory representation beyond simple summaries. Capturing these essential details is crucial for episodic memory tasks, as it allows the agent to recall past experiences accurately. Each episode was annotated with 10 *what*, *when*, and *where* questions.

---

> ### Author Response · Authors · 2024-11-20
> **I think I like what Section 4.1 is implying about combining a "master's" memory with that of an agent's, but it is not presented very clearly, and I do not see a connection with temporal graphs like the subsection title suggests**
>
> ### Section 4.1
>
> When a cognitive agent assists an individual with memory issues, it leverages past interactions to provide personalized support. The process begins by identifying and extracting the **personalized cluster** \( C_p \), representing the relevant memory subgraph for that individual. This subgraph consists of episodes, actions, and events associated with the person. The **personalized cluster** \( C_p \) is defined as:
>
> $$
> C_p = \{ G_p \mid \text{episodes associated with person } p \}
> $$
>
> where:
>
> $$
> G_p = (V_p, E_p)
> $$
>
> represents the subgraph containing episodes \( V_p \) and edges \( E_p \) related to person \( p \), preserving all temporal and contextual information.
>
> Once the **personalized cluster** \( C_p \) is extracted, the agent identifies **event clusters** \( C_e \) within the subgraph, representing various actions performed by the individual over time:
>
> $$
> C_e = \{ E_e \mid \text{events associated with episodes in } G_p \}
> $$
>
> The agent checks for the most recent sequence of events, denoted as:
>
> $$
> \mathcal{S}_t = (S_1, S_2, \dots, S_t)
> $$
>
> To provide recommendations, the agent searches the past memory subgraph for instances where the task was previously performed and identifies the subsequent event. The next action \( S_{t+1} \) is determined by:
>
> $$
> S_{t+1} = \text{argmax}_{S \in G_p} \mathbb{I}(\mathcal{S}_t \subset \mathcal{S})
> $$
>
> where:
>
> $$
> \mathbb{I}(\mathcal{S}_t \subset \mathcal{S})
> $$
>
> is the indicator function verifying if the current sequence $$ \( \mathcal{S}_t \)$$ appears as a subsequence in a past sequence $$\( \mathcal{S} \) within \( G_p \)$$.
>
> If there are multiple matches, the most frequently occurring next event is chosen:
>
> $$
> S_{t+1} = \text{mode}(\{ S_{t+1} \in \mathcal{S} \mid \mathcal{S}_t \subset \mathcal{S}, \mathcal{S} \in G_p \})
> $$
>
> This approach allows the agent to use past episodic data effectively, providing recommendations or answers based on past actions even when explicit patterns are absent. By analyzing subgraphs containing both days as nodes and events as sub-nodes, the agent identifies the most relevant next action, ensuring reliable support for individuals with memory impairments. This is especially useful in cases where the agent has been an assistant to many individuals.

---

> ### Author Response · Authors · 2024-11-20
> **Nor do I see how this section is a "theoretical comparison"**
>
> #### I am contemplating revising that section to include a mathematical representation of the recommendation process within the framework of Episodic Memory and Contextual Awareness (EMCA). I would greatly appreciate any suggestions or recommendations you may have for improving this approach
> Thank you for pointing that out. I understand that the use of the term "master" in line 346 might be confusing and could carry unintended connotations. The term "master" is meant to refer to the individual or human whom the agent is serving as an assistant to, and we will clarify this earlier in Section 4.1 to avoid any confusion.
> In response to your suggestion, I will also consider replacing "master" with a more neutral and clear term, such as "user" or "human operator," to align with contemporary language preferences and ensure the term does not cause any discomfort. This clarification will be made in the paper, and we will also make sure to introduce this role earlier in the relevant section to avoid any ambiguity.
> If you have any further suggestions or specific terms you would recommend, please feel free to share.

---

> ### Author Response · Authors · 2024-11-20
> **Results are poorly presented**
>
> #### Thank you for your feedback. We acknowledge that the results section could benefit from improvements in clarity and presentation. In the updated version of the paper, we will begin by providing a clearer explanation of the specific tasks being evaluated, such as episodic QA, memory localization, and ablation studies, to ensure that the reader understands the objectives of each task. Additionally, we will present structured comparisons, highlighting the performance of our approach in relation to baseline methods like 2D-TAN, VSLNet, CONE, and SPOTEM, showcasing the improvements made by our method. The ablation study results will also be presented in a more accessible manner, emphasizing the contribution of each framework component to the overall performance. We will also add time taken as episodes increase.

---

> ### Author Response · Authors · 2024-11-20
> **Formatting/Clarity issues**
>
> **Check for spaces after periods or colons throughout the paper.**
> #### In the revised version of the paper, we will ensure consistent spacing throughout to improve readability and adhere to proper formatting standards.
> **Line spacing is odd in much of the paper**
> #### We appreciate your observation about the line spacing. We will review and adjust the line spacing in the paper to ensure it is consistent and visually comfortable for readers, enhancing the overall clarity and flow of the document.
> **Figure placement should ideally be on the top or bottom of a page, not in the middle with paper text above and below the paper. This makes the paper difficult to follow**
> #### We agree that figures should be positioned in a way that supports the flow of the paper. In the revised version, we will reposition figures to the top or bottom of pages,
> **Figure 4, the legend has oddly shaped circles**
> #### We have noted your concern regarding the oddly shaped circles in the legend of Figure 4. We will revise the legend to ensure that the circles are properly shaped.
> **Figure 5's result is good, but it should not be a line graph with x axis being method and lines being the dataset. Instead it should be datasets on the x-axis and methods on the y-axis.The results in section 5.1.1 are all discussed in a single paragraph. Is that all the main results?**
> In Section 5.1.1. Result table is
> ### Table 1: Time taken by different methods for various datasets (in seconds).
>
> | **Method**    | **Big Bang Theory (s)** | **PerLTQA (s)** | **Agent (s)** | **LLQA (s)** |
> |---------------|-------------------------|-----------------|---------------|--------------|
> | Our Model     | 600                     | 300             | 180           | 240          |
> | EMR           | 900                     | 600             | 480           | 540          |
> | GraphRag      | 4200                    | 3600            | 3300          | 3480         |
> | GNN Rag       | 4200                    | 3600            | 3000          | 2700         |
> | TempoQA       | 1800                    | 1800            | 2400          | 2100         |
> | Arigraph      | 84000                   | 72000           | 10800         | 9000         |
> Regarding the issue of overlapping axis lines in the graph, we acknowledge that this occurs due to the large scale of the "Trigraph" (or similar metric) values. This occurs even when axes are reversed.Since the values vary significantly, especially for methods like "Arigraph," the axis scaling is stretching the range too far.
> **I would recommend splitting this up into a few bolded mini-sections and showing the main takeaway of each figure along with highlights on how the method performed, possibly with qualitative results.**
> #### We will update this section in the updated version of the paper.

---

> ### Author Response · Authors · 2024-11-20
> ***Insufficient Motivation: The introduction section does not adequately establish the necessity of this system or why it improves upon existing learning frameworks for cognitive agents. Additional motivation for the need of an episodic memory for a cognitive agent would help contextualize EMCA's contributions**
>
> **Answer to weakness 2**
> *.**
>
> A2) Episodic memory, as introduced by Tulving (1972) [^1], represents the capacity to recall personal experiences embedded within specific temporal and spatial contexts. Unlike semantic memory, which holds general knowledge, episodic memory encompasses detailed information about events, integrating aspects such as time, place, characters, and context. Tulving’s framework organizes these components into cohesive episodes.
> Drawing from this foundational concept, we propose a model where episodic memories are structured as a graph. Each episode \( \text{Episode}_i \) functions as a node:
> $$
> \[
> \text{Episode}_i = \{ \mathbf{C}_i, \mathbf{T}_i, \mathbf{L}_i, \mathbf{e}_i \}
> \]
> $$
> In this model,$$ \( \mathbf{C}_i \) $$stands for characters, $$\( \mathbf{T}_i \)$$ represents temporal markers, $$\( \mathbf{L}_i \)$$ signifies location, and $$\( \mathbf{e}_i \)$$ encapsulates events. Semantic edges \( \mathbf{S}(v_i, v_j) \) connect nodes that share common elements, while temporal edges $$\( \mathbf{T}(v_i, v_j) \)$$ map the chronological flow of experiences. To enhance retrieval efficiency, we apply a dynamic clustering mechanism that organizes similar episodes based on both temporal and contextual similarities.
>
> Our retrieval system supports three query types: "what" (contextual), "when" (temporal), and "where" (spatial), as outlined by Stephen et al., and Holland and Smulders (2011), enabling human-like memory recall. This is particularly valuable for applications in social companion robotics, aiding elderly or memory-impaired individuals.
>
> For such a cognitive agent, the ability to recall episodic memories is essential for human cognition, linking personal experiences to specific temporal and spatial contexts. Existing memory models often struggle with continuous, time-series data, which limits their ability to simulate episodic recall effectively. Many of these systems fail to store dialogues as multimodal data, preventing them from capturing the rich, context-dependent nature of human memory. Additionally, most existing approaches store a single experience as one isolated episode and lack a mechanism for retrieving information across multiple experiences, hindering their ability to integrate knowledge over time.
>
> Furthermore, current episodic memory systems are typically restricted to performing a specific, predefined task, limiting their flexibility and adaptability. In contrast, our system is designed to be more versatile, capable of handling a variety of tasks and dynamically adapting to new scenarios, making it far more suited for real-world applications that require memory integration across different contexts and time periods.
>
> By integrating multimodal data and time-related information into the episodic memory framework, our model extends experience memory localization, recommendation, and question answering. It provides a robust foundation for adaptable, scalable systems capable of operating without frequent retraining, applicable to real-world scenarios such as social companion robots and autonomous task planning systems.
>
> ### **Contributions:**
> 1. Temporal connections are managed without complex pattern learning, enabling adaptive reasoning and retrieval of subgraphs from past experiences.
> 2. The system incrementally stores and retrieves episodic memories, dynamically clustering them based on temporal and contextual affinities.
> 3. A multi-edge graph framework optimizes path traversals for dynamic memory retrieval and personalized recommendations across subjective timescales.
> 4. A new dataset is introduced to improve episodic memory question answering, enhancing the agent's ability to respond to queries based on past events.
>
> Our model’s versatility is demonstrated through comparisons with existing systems that require retraining. It handles various dataset types, including visual, multimodal, and text-based data, and excels in temporal reasoning even without explicit timestamps, addressing complex memory retrieval tasks across diverse applications.
>
> [^1]: Tulving, E. (1972). *Episodic and semantic memory*. In *Organization of Memory* (pp. 381-403). Academic Press.

---

> ### Author Response · Authors · 2024-11-20
> **Citations**
>
> Other relevant concurrent work on memory in robotics, some of which use graphs while others do not. [1] Xie, Quanting, et al. "Embodied-RAG: General Non-parametric Embodied Memory for Retrieval and Generation." arXiv preprint arXiv:2409.18313 (2024). [2] Anwar, Abrar, et al. "ReMEmbR: Building and Reasoning Over Long-Horizon Spatio-Temporal Memory for Robot Navigation." arXiv preprint arXiv:2409.13682 (2024). [3] Bärmann, Leonard, et al. "Episodic Memory Verbalization using Hierarchical Representations of Life-Long Robot Experience." arXiv preprint arXiv:2409.17702 (2024).
>
> [1] Xie, Quanting, et al. "Embodied-RAG: General Non-parametric Embodied Memory for Retrieval and Generation." arXiv preprint arXiv:2409.18313 (2024).
> [2] Anwar, Abrar, et al. "ReMEmbR: Building and Reasoning Over Long-Horizon Spatio-Temporal Memory for Robot Navigation." arXiv preprint arXiv:2409.13682 (2024).
> These studies focus on navigation tasks which is a problem not discussed in this paper..
> However, we have considered:
> [3] Bärmann, Leonard, et al. "Episodic Memory Verbalization using Hierarchical Representations of Life-Long Robot Experience." arXiv preprint arXiv:2409.17702 (2024).
> Our approach relates more closely to [3], as it focuses on episodic memory representation and analysis, which aligns with the objectives of our research. We have performed analysis using this method, and the results are included below to demonstrate the outcomes in comparison to our model's performance
> #### More comparison with memory models will be added in the appendix:
>
> | **Method**                    | **Recall Accuracy** |
> |-------------------------------|---------------------|
> | Episodic Memory Verbalization  | 50%                 |
> | Rehearsal Memory               | 36%                 |
> | STM                           | 30%                 |
> | DNC                           | 35%                 |
> | LT-CT                         | 50%                 |
> | **Ours**                       | **81%**             |
>
> *Caption: Recall accuracy for episodic memory question answering.*
>
> ---

---

> ### Author Response · Authors · 2024-11-20
> **How was the dataset constructed? This is a major contribution that is not discussed.**
>
> We propose a comprehensive dataset framework designed to evaluate and enhance episodic memory systems in artificial agents. This framework integrates multiple datasets, including a custom set of episodic questions based on the TV series *The Big Bang Theory*, spanning all nine seasons (181 episodes). The aim is to assess memory recall and narrative understanding in complex scenarios.
>
> We introduce the *Agent Dataset*, a 10-episode time-series dataset created in Unity3D, where a virtual agent performs tasks and interacts with characters in realistic environments, simulating the role of companion robots. This dataset emphasizes the importance of multi-sensory inputs and task execution, challenging the agent to process and integrate information from *dialogues* and *visual cues* to maintain task order and achieve context-driven objectives.
>
> Additionally, we adapted the **Ego4D dataset**, restructuring its activity sequences into simulated chronological episodes to address the original absence of time-series data—portraying an agent performing a series of activities over 30 days. We also combined group activity videos designed for active speaker recognition. This transformation enables episodic queries such as "Where did I place the agricultural tool on the last day of farming?", enhancing the ability to localize and retrieve temporal experiences effectively.
>
> Together with the **PerLTQA** [du2024perltqapersonallongtermmemory] and **LLQA** [dolan-brockett-2005-automatically] datasets, which test essential episodic memory dimensions—**"what"** (context), **"when"** (time), and **"where"** (place)—this framework forms a robust benchmark for evaluating advanced episodic memory capabilities in AI systems.
>
> **Data Annotation**: The data was carefully annotated to tag scene information and identify characters in dialogues, ensuring that the model could recognize character presence and understand related events. This included explicitly tagging scene details for location identification and differentiating characters present in the scene versus those mentioned. Events within dialogues were also meticulously annotated to capture key details, facilitating effective memory representation beyond simple summaries. Capturing these essential details is crucial for episodic memory tasks, as it allows the agent to recall past experiences accurately. Each episode was annotated with 10 *what*, *when*, and *where* questions.

---

> ### Author Response · Authors · 2024-11-20
> **What does a successful or unsuccessful example look like? I would recommend looking at [2] or [3] above to see how to discuss dataset creation in such a setting.**
>
> A successful example refers to correctly answering the question and extracting the appropriate experience from memory. An unsuccessful attempt, on the other hand, occurs when the question is not answered correctly.In localisation successful attempt refers to getting the correct memory location.Failed attempt is getting wrong location Our model will always attempt to generate an answer based on the experiences collected, but in cases where the answer is wrong, it is considered an unsuccessful attempt. We will provide detailed explanations and visual aids for this part in the appendix of the paper..

---

> ### Author Response · Authors · 2024-11-20
> **In Table 1, what metric is being used? It does not say in the caption or the table. It should be re-iterated in the table itself.**
>
> The metric used in Table 1 is recall accuracy. We will update the table and its caption to clearly indicate this metric for better clarity.

---

> ### Author Response · Authors · 2024-11-20
> **Is a forgetting mechanism necessary, or would it be more like a memory aggregation mechanism so that retrieval is still efficient?**
>
> A forgetting mechanism is indeed relevant, as it aids in maintaining an efficient retrieval process. Currently, we have implemented a forgetting mechanism that is triggered based on how frequently a specific part of memory is accessed. This helps to optimize performance by discarding memory portions that are rarely or never used, preventing the memory from becoming cluttered with irrelevant information.This ensures that data which is not used is not saved.We also aim to check more mechanisms that will be useful for forgetting.Addressing the balance between forgetting and memory aggregation is an ongoing task we are actively exploring this part.

---

> ### Author Response · Authors · 2024-11-20
> **How does the incremental storage/retrieval scale as the number of episodes change? This result is not displayed in the paper but I would argue is very important. If you only used say 10 episodes of EM instead of 181, is there a difference in performance? This would directly support contribution number 2 in Section 1 of your paper**
>
> To address the query about the scalability of incremental storage and retrieval as the number of episodes increases, we conducted a comparison between different episode counts (10, 50, 100, and 181 episodes).
> In terms of accuracy, all values remain consistent. However, in terms of retrieval time, there is a slight increase as the number of episodes grows.
>
> ### Table 1: Retrieval times for different episode counts.
>
> | **Number of Episodes** | **Retrieval Time (ms)** |
> |------------------------|-------------------------|
> | 10                     | 9.11                    |
> | 50                     | 11.0                    |
> | 100                    | 11.5                    |
> | 181                    | 12.0                    |
>
> To address the query about the scalability of incremental storage and retrieval as the number of episodes increases, we conducted a comparison between different episode counts (10, 50, 100, and 181 episodes).
>
> In terms of accuracy, all values remain consistent. However, in terms of retrieval time, there is a slight increase as the number of episodes grows. The retrieval times for different episode counts are provided in the table above.
>
> This shows that while retrieval time does increase, it is not drastic, and the increase is quite modest. The clustering mechanism plays a crucial role in ensuring that retrieval times remain low, even as the number of episodes increases. This highlights the efficiency of our method in scaling with the number of episodes.
>
> We will include these results in the ablation studies section, as suggested, to better support the contribution discussed in Section 1 of the paper.

---

> ### Author Response · Authors · 2024-11-20
> **What is "Time complexity" in Table 3?Time complexity of BFS is O(V+E), but here a number is used instead. Authors should use "retrieval time" or something similar instead of time complexity.**
>
> We will update this part accordingly

---

> ### Author Response · Authors · 2024-11-20
> **Can the authors describe the main takeaways of Section 4? I still do not understand the insights this section is supposed to provide.**
>
> This section was to compare our graph with mathematically with other graphs
> ### Takeaways of this Section:
>
> 1) **Episodic memory can recommend the next activity based on previous interactions**, even when recurring patterns are not observed.
>
> 2) **Comparison with knowledge graphs**: This comparison highlights the complexity of knowledge graphs in such scenarios. For example, when 3 episodes enter an episodic graph, it adds only 3 nodes, whereas a knowledge graph might add 30-50 new nodes.

---

> ### Author Response · Authors · 2024-11-21
> **In terms of the structure of the paper, I do not think that the text space used for discussing how signals are captured (section 3.1) is very useful. I would argue this is the case with a chunk of this paper's equations, where it simply adds space when it does not need to, and it makes the paper more difficult to read. I would recommend condensing the information and matching ICLR's 9-page recommendation as opposed to 10. Similarly, there is an excessive use of new lines at arbitrary positions.**
>
> Thank you for your feedback. I appreciate your suggestion regarding the structure of the paper, particularly the section on how signals are captured (Section 3.1), as well as the formatting issues related to the length of equations and the excessive use of new lines. I understand that these elements may make the paper more difficult to read and could be streamlined.
> I will address these concerns in the revised version of the paper, ensuring that the content is condensed appropriately while maintaining clarity.

---

> ### Comment · Reviewer_jSod · 2024-11-24
> **Response to rebuttal**
>
> Thank you for making the large number of edits and the detailed (and somewhat overwhelming) responses. The authors have made a lot of the paper clearer, and the edited manuscript that was uploaded looks better.
>
> Similar to the Reviewer YT1b24, I better understand what the method is doing, but I think a core part of research is to understand why these components are important. The intuitions behind these design decisions are not described. Some comparison with other baselines is made in Section 6.2, but it doesn't really provide insights on what is required for someone to build similar strong episodic memory retrieval systems.
>
> The framing of the dataset is better, but I was left unsatisfied with how the authors discussed their method. This is mostly due to the over-engineered-ness of the method while not explaining the reasons for it. I will keep my score the same.

---

> ### Author Response · Authors · 2024-11-28
>
> We hope that the queries raised during the previous phase have been addressed through our clarifications and the additional details provided . In response to your above query I would like to make the following clear:
> The focus of our work is on episodic memory encoding and retrieval, inspired by the psychological processes of the human episodic memory. Our methodology integrates visual and audio content, emphasizing key memory-dependent factors such as location, persons, time, and the sequence of events. Human experiences are inherently multimodal in nature; keeping this in mind, our aim was to capture the multimodal aspect of memory encoding and retrieval. While we leveraged models like CLIP and other large language models for data preprocessing, this aspect remains intentionally flexible to encourage innovation. Researchers can opt for any combination of image, audio, speech, or language models based on their specific needs. Since our focus was not on signal capturing, we did not carry out ablation studies on this component.
>
> Since our proposed methodology centers on encoding and retrieval, we prioritized these aspects, enabling tasks like episodic memory localization and episodic QA to assess retrieval accuracy. To evaluate our approach, we conducted comparisons with graph-based retrieval methods such as GNN-RAG and GraphRAG (Section 6.2), demonstrating the effectiveness of our method. Additionally, we performed an ablation study to analyze time complexities between clustered and non-clustered approaches, and we examined the efficiency of storing diverse features in graph nodes to test the robustness of our encoding strategy.
>
> Furthermore, we analyzed the structure of the episodic graph by converting it into a knowledge graph-like structure and performing structural evaluations. We also evaluated different traditional traversal techniques, such as BFS and DFS, as part of the ablation study to highlight why traditional traversal methods were unsuitable for our specific requirements. Taken together, these evaluations demonstrate that we conducted the necessary assessments aligned with the aims of the research, validating our methodological choices and highlighting the robustness of our approach.

---

### Official Review · Reviewer_zT5v · 2024-11-04

**Soundness:** 2
**Presentation:** 2
**Contribution:** 3
**Rating:** 3
**Confidence:** 4

**Summary:**

The paper presents Episodic Memory for Cognitive Agents (EMCA), a framework designed to support memory retention and retrieval in  cognitive agents. EMCA models episodic memory using a graph-based structure that incrementally stores and organizes multimodal experiences—such as speech, vision, and non-verbal cues—without pre-training on specific scenarios. This approach enables agent to keep adding new experiences continuously from data. This supposedly allows for flexible temporal reasoning across different datasets. EMCA’s dynamic memory graph builds semantic and temporal connections, enabling context-aware retrieval and clustering of memories based on query relevance. The framework aims to improve task execution , and reasoning by recalling contextually significant past events. Empirical tests reported indicate that EMCA adapts to real-world data, demonstrating good recall in unpredictable settings.

**Strengths:**

I think the paper is trying to address a very important problem of how can an autonomous agent keep memoring new experienes and the recall those flexibly based on the context, question or query. Specifcally I see two main strengths.
Pretrained Models: The model builds on pretrained models that help with extraction of compenents, but this also means the system does not require pre-training on every specific scenario from scratch, which is a significant strength as it allows flexibility across contexts.
Dataset Diversity: The authors evaluated the system on multiple datasets, demonstrating a broad application range, although it should be noted that most of these datasets were developed by the authors.

**Weaknesses:**

Despite tackling an important problem, the paper suffers from serious clarity and coherence issues that obscure its contributions and weaken its scientific rigor. The presentation is fragmented, key concepts are inadequately explained, and essential technical details are missing, all of which make it challenging to assess the model’s validity and potential impact. Specifically, the weaknesses include:
1. The authors claim EMCA encodes data in a way that resembles human memory, but there is no evidence or detailed explanation to support this claim from a neural encoding way. Such claims should be toned down. You should instead  emphasize on the 'what', 'where' 'when' organisation from a psychological perspective of episodic memory.

2. Insufficient Motivation: The introduction section does not adequately establish the necessity of this system or why it improves upon existing learning frameworks for cognitive agents. Additional motivation for the need of an episodic memory for a cognitive agent would help contextualize EMCA's contributions.


3. Minimal Related Work Discussion: Essentialy the model is an encode and retrival model with some dynamic reorganistion. The related work section is sparse and lacks comparisons to key formal methods like Hopfield networks or other established models in episodic memory encoding and retrieval. A more rigorous comparison to established human memory models would also strengthen the paper.

4. Unclear Implementation and Integration Details: Although multiple models and methods are mentioned, the paper lacks a cohesive description of how these components integrate within the system. Critical details such as model architecture, parameter settings, and processing pipelines are absent, making it difficult to assess or replicate the work. A system architecture diagram, a table of key parameters, or pseudocode for the main processing pipeline would help.

Vague Statistical Estimation Methods: The paper mentions the use of statistical methods for estimating missing timestamps but does not specify which methods were used, leaving an important aspect of the framework unexplained.

Surface-level Comparison with Temporal and Knowledge Graphs: The comparisons with temporal and knowledge graph structures are brief and lack depth, offering limited insight into how EMCA differs from or improves upon these existing approaches.

Undefined Terminology and Variables: Certain terms and variables (e.g.,"key events", "location weight," "subjective temporal timescales") are introduced without sufficient explanation or definition, reducing clarity.

Overreliance on Custom Datasets: While the use of various datasets to evaluate EMCA is a strength, most of these datasets were developed by the authors, which could indicate potential biases in testing and validation.

Limited Explanation of Retrieval Policy: The retrieval policy and memory clustering mechanisms, while central to EMCA’s functionality, are described only briefly. A more detailed explanation would clarify how these mechanisms adapt to different query types and scenarios.

**Questions:**

Statistical Estimation Methods for Timestamps: Which specific statistical methods are used for estimating timestamps in the absence of explicit temporal markers?

Union of Tacoustic and Tvoiced: In combining Tacoustic and Tvoiced, what is the methodology for performing this union? Is Tvoiced identical or related to another variable, such as St?

Format and Extraction of Visual Details: What is the format of the visual scene details (e.g., Vscene, Vplace, Vtime), and how are these extracted and integrated into the memory graph?

Defining Key Events and Hierarchical Organization: How are key events identified, and what hierarchical structure is used to organize these events?

Relation between Taudio and Tcombined: Is Taudio equivalent to Tcombined, or is there another relationship between these variables?

Task Categories for Text Summarization: How are text summaries grouped into broader task categories (e.g., meetings, lunches)? What criteria and process are used to define these categories?

Similarity Calculation in Equation 8: Equation 8 is intended to measure similarity between an event and multiple episodes, but it isn’t clear how it accomplishes this. Could you clarify how this calculation works?

Location Weight Definition: How is "location weight" defined, and how does it differ from location similarity in the model?

Temporal Parameter in Equation 14: In Equation 14, should the parameter be (t-k) instead of just t? If not, what purpose does the current form of the equation serve?

Meaning of "Agent Comprehends": In line 274, it says the "agent comprehends" something. Does this imply processing by a language model, and if so, could you clarify which model is used?

Definition of the Set Du: How is the set Du defined in the context of the framework?

Similarity Function in Line 283: Which similarity function is used in line 283, and what factors are considered?

Role of w and l Functions: In line 287, the w and l functions are mentioned. Could you elaborate on their roles within the memory retrieval mechanism?

---

> ### Author Response · Authors · 2024-11-20
> **The authors claim EMCA encodes data in a way that resembles human memory, but there is no evidence or detailed explanation to support this claim from a neural encoding way. Such claims should be toned down. You should instead emphasize on the 'what', 'where' 'when' organisation from a psychological perspective of episodic memory.**
>
> To better align with the evidence provided and ensure that the claims are well-supported, we have decided to adjust our approach in the main section of the paper and expand on the relevant details in the appendix.
>
> **As suggested we will tone down the paragraph mentioned in section 3 to**:
> > EMCA’s methodology for collecting and structuring episodic experiences is inspired by cognitive psychology, specifically the 'what', 'where', and 'when' (WWW) components of episodic memory. This approach highlights the agent’s ability to independently capture multimodal data—visual and auditory—to build a comprehensive understanding of its environment.
>
> Also, to highlight processing of visual and auditory details separately, we will elaborate our claim by adding an extra section in the appendix as:
> > EMCA's methodology for collecting episodic experiences in robotic cognition draws inspiration from the human brain's mechanisms for encoding sensory information, particularly the distinct roles of the occipital and temporal lobes. In human cognition, the occipital lobe processes visual stimuli, while the temporal lobe is responsible for auditory information. Despite these processes occurring in specialised regions, the brain synchronises these sensory inputs within a unified temporal framework, enabling the formation of cohesive and contextually rich memories. EMCA replicates this principle by employing separate pipelines for processing visual data (e.g., spatial and object recognition) and auditory data (e.g., speech and environmental sounds), which are then temporally aligned to construct a coherent representation of the agent's environment.

---

> ### Author Response · Authors · 2024-11-20
> **Insufficient Motivation: The introduction section does not adequately establish the necessity of this system or why it improves upon existing learning frameworks for cognitive agents. Additional motivation for the need of an episodic memory for a cognitive agent would help contextualize EMCA's contributions**
>
> Episodic memory, as introduced by Tulving (1972) [^1], represents the capacity to recall personal experiences embedded within specific temporal and spatial contexts. Unlike semantic memory, which holds general knowledge, episodic memory encompasses detailed information about events, integrating aspects such as time, place, characters, and context. Tulving’s framework organizes these components into cohesive episodes.
> Drawing from this foundational concept, we propose a model where episodic memories are structured as a graph. Each episode \( \text{Episode}_i \) functions as a node:
> $$
> \[
> \text{Episode}_i = \{ \mathbf{C}_i, \mathbf{T}_i, \mathbf{L}_i, \mathbf{e}_i \}
> \]
> $$
> In this model,$$ \( \mathbf{C}_i \) $$stands for characters, $$\( \mathbf{T}_i \)$$ represents temporal markers, $$\( \mathbf{L}_i \)$$ signifies location, and $$\( \mathbf{e}_i \)$$ encapsulates events. Semantic edges \( \mathbf{S}(v_i, v_j) \) connect nodes that share common elements, while temporal edges $$\( \mathbf{T}(v_i, v_j) \)$$ map the chronological flow of experiences. To enhance retrieval efficiency, we apply a dynamic clustering mechanism that organizes similar episodes based on both temporal and contextual similarities.
>
> Our retrieval system supports three query types: "what" (contextual), "when" (temporal), and "where" (spatial), as outlined by Stephen et al., and Holland and Smulders (2011), enabling human-like memory recall. This is particularly valuable for applications in social companion robotics, aiding elderly or memory-impaired individuals.
>
> For such a cognitive agent, the ability to recall episodic memories is essential for human cognition, linking personal experiences to specific temporal and spatial contexts. Existing memory models often struggle with continuous, time-series data, which limits their ability to simulate episodic recall effectively. Many of these systems fail to store dialogues as multimodal data, preventing them from capturing the rich, context-dependent nature of human memory. Additionally, most existing approaches store a single experience as one isolated episode and lack a mechanism for retrieving information across multiple experiences, hindering their ability to integrate knowledge over time.
>
> Furthermore, current episodic memory systems are typically restricted to performing a specific, predefined task, limiting their flexibility and adaptability. In contrast, our system is designed to be more versatile, capable of handling a variety of tasks and dynamically adapting to new scenarios, making it far more suited for real-world applications that require memory integration across different contexts and time periods.
>
> By integrating multimodal data and time-related information into the episodic memory framework, our model extends experience memory localization, recommendation, and question answering. It provides a robust foundation for adaptable, scalable systems capable of operating without frequent retraining, applicable to real-world scenarios such as social companion robots and autonomous task planning systems.
>
> ### **Contributions:**
> 1. Temporal connections are managed without complex pattern learning, enabling adaptive reasoning and retrieval of subgraphs from past experiences.
> 2. The system incrementally stores and retrieves episodic memories, dynamically clustering them based on temporal and contextual affinities.
> 3. A multi-edge graph framework optimizes path traversals for dynamic memory retrieval and personalized recommendations across subjective timescales.
> 4. A new dataset is introduced to improve episodic memory question answering, enhancing the agent's ability to respond to queries based on past events.
>
> Our model’s versatility is demonstrated through comparisons with existing systems that require retraining. It handles various dataset types, including visual, multimodal, and text-based data, and excels in temporal reasoning even without explicit timestamps, addressing complex memory retrieval tasks across diverse applications.
>
> [^1]: Tulving, E. (1972). *Episodic and semantic memory*. In *Organization of Memory* (pp. 381-403). Academic Press.

---

> ### Author Response · Authors · 2024-11-20
> **Minimal Related Work Discussion:Essentially, the model is an encode-and-retrieval model with dynamic reorganization. The related work section is sparse and lacks comparisons to key formal methods like Hopfield networks or other established models in episodic memory encoding and retrieval. A more rigorous comparison to established human memory models would also strengthen the paper.**
>
> To enrich the discussion, we intend to incorporate a more comprehensive comparison to established human memory models, which will significantly strengthen the paper. To achieve this, we plan to include additional references that span a wider array of memory networks and methodologies. This will include foundational works such as:
>
> 1. **Caiming Xiong, Stephen Merity, and Richard Socher.** Dynamic memory networks for visual and textual question answering, 2016. [Link](https://arxiv.org/abs/1603.01417).
> 2. **Hu Xu, Gargi Ghosh, Po-Yao Huang, Prahal Arora, Masoumeh Aminzadeh, Christoph Feichtenhofer, Florian Metze, and Luke Zettlemoyer.** VLM: Task-agnostic video-language model pre-training for video understanding. In Chengqing Zong, Fei Xia, Wenjie Li, and Roberto Navigli (eds.), Findings of the Association for Computational Linguistics: ACL-IJCNLP 2021, pp. 4227–4239, Online, August 2021. DOI: [10.18653/v1/2021.findings-acl.370](https://aclanthology.org/2021.findings-acl.370).
> 3. **Songyang Zhang, Houwen Peng, Jianlong Fu, and Jiebo Luo.** Learning 2D temporal adjacent networks for moment localization with natural language, 2020. [Link](https://arxiv.org/abs/1912.03590).
> 4. **Zhu Zhang, Chang Zhou, Jianxin Ma, Zhijie Lin, Jingren Zhou, Hongxia Yang, and Zhou Zhao.** Learning to rehearse in long sequence memorization, 2021. [Link](https://arxiv.org/abs/2106.01096).
> 5. **Samyak Datta, Sameer Dharur, Vincent Cartillier, Ruta Desai, Mukul Khanna, Dhruv Batra, and Devi Parikh.** Episodic memory question answering, 2022. [Link](https://arxiv.org/abs/2205.01652).
> 6. **Hung Le, Truyen Tran, and Svetha Venkatesh.** Self-attentive associative memory, 2020. [Link](https://arxiv.org/abs/2002.03519).
> 7. **Naiyuan Liu, Xiaohan Wang, Xiaobo Li, Yi Yang, and Yueting Zhuang.** Reler@zju-alibaba submission to the ego4d natural language queries challenge 2022, 2022. [Link](https://arxiv.org/abs/2207.00383).
> 8. **Adam Santoro, Sergey Bartunov, Matthew Botvinick, Daan Wierstra, and Timothy Lillicrap.** One-shot learning with memory-augmented neural networks, 2016. [Link](https://arxiv.org/abs/1605.06065).
> 9. **Bärmann, Leonard, et al.** "Episodic Memory Verbalization using Hierarchical Representations of Life-Long Robot Experience." arXiv preprint arXiv:2409.17702 (2024).
>
> Based on these studies, we have carried out additional analysis and experiments. We will be adding the following tables in the revised paper:
>
> #### Comparison with NLQ models for proving our model's capacity in retrieving correct experiences from the episodic memory graph.
>
> | **Method**            | **IOU = 0.3 R@1** | **IOU = 0.5 R@5** | **mIOU** |
> |-----------------------|-------------------|-------------------|----------|
> | 2D-TAN                | 4.32              | 2.60              | 5.62     |
> | VSLNet                | 8.09              | 7.03              | 7.65     |
> | CONE                  | 10.55             | 7.54              | 9.04     |
> | RELER                 | 12.89             | 8.14              | 10.51    |
> | SPOTEM                | 18.13             | 13.43             | 15.78    |
> | **Ours**              | **26.46**         | **25.5**          | **25.98**|
>
> *Caption: Performance comparison on episodic memory localization.*
>
> ---
>
> #### More comparison with memory models will be added in the appendix:
>
> | **Method**                    | **Recall Accuracy** |
> |-------------------------------|---------------------|
> | Episodic Memory Verbalization  | 50%                 |
> | Rehearsal Memory               | 36%                 |
> | STM                           | 30%                 |
> | DNC                           | 35%                 |
> | LT-CT                         | 50%                 |
> | **Ours**                       | **81%**             |
>
> *Caption: Recall accuracy for episodic memory question answering.*
>
> ---
>
> These models were tested on the Ego4D dataset. Models that have achieved state-of-the-art (SOTA) results in the NLQ challenge in Ego4D, as well as the VLQ models, are also memory models that have achieved SOTA results.
>
> We will also include Hopfield models, citing relevant literature and equations, and highlighting that these models are ineffective in the absence of recurring patterns.

---

> ### Author Response · Authors · 2024-11-20
> **Vague Statistical Estimation Methods: The paper mentions the use of statistical methods for estimating missing timestamps but does not specify which methods were used, leaving an important aspect of the framework unexplained. If the reviewer could kindly specify the line referring to 'statistical timestamps,' it would help us provide a more targeted response to address the query.**
>
> #### Regarding the concern about vague statistical estimation methods for missing timestamps, it's important to clarify that there are no traditional statistical methods used for estimating missing timestamps in our framework. Instead, we treat each episode as representing a distinct day. If the agent is able to capture specific dates visually or through dialogues, the corresponding temporal indexer will be updated to reflect the correct date and time.
> #### Since all episodes are connected temporally by making use of temporal connections the temporal indexer is updated accordingly. This means that for any episode, its timestamp is adjusted based on the captured date, and subsequent episodes are indexed with respect to this temporal structure. For example, the "before" node will be one day prior, while the "after" node will be one day later, and so on. This approach ensures that the temporal relationships between episodes remain consistent and that the framework operates in a way that accurately reflects the passage of time as the agent processes the information.We will clarify this part in the revised version of the paper.. Specifically, we will outline how episodes are treated as distinct days, how the temporal indexer updates based on the agent's ability to capture dates visually or from dialogues, and how the relationships between episodes are maintained with respect to the passage of time. Also even without external timestamps our model will understand the order in which episodes are arranged because of maintaining temporal edges.

---

> ### Author Response · Authors · 2024-11-20
> **Surface-level Comparison with Temporal and Knowledge Graphs  The comparisons with temporal and knowledge graph structures are brief and lack depth, offering limited insight into how EMCA differs from or improves upon these existing approaches. In this section, we present a more detailed analysis comparing episodic graphs with knowledge graphs in a multi-agent environment, focusing on their structural complexity and memory transfer efficiency**
>
> #### Knowledge Graphs vs Episodic Graphs
>
> We performed an evaluation of replacing episodic graphs with knowledge graphs in single-agent systems, highlighting the structural complexity and memory transfer efficiency. The following table summarizes this comparison.
>
> | **Metric**      | **Episodic Graph** | **Knowledge Graph** |
> |-----------------|--------------------|---------------------|
> | Nodes           | 3                  | 16                  |
> | Edges           | 3                  | 36                  |
> | Average Degree  | 2.0                | 4.5                 |
> | Density         | 0.1                | 0.3                 |
>
> As shown in the table, **knowledge graphs** have a significantly higher complexity with more nodes, edges, and greater connectivity, making them less suitable for real-time memory retrieval.
>
> #### Power-Law Characteristics of Graphs
>
> We also analyzed the power-law characteristics of both graph types. The episodic graph exhibits a steep decay (\(\alpha = 5.45\)), indicating simplicity, whereas the knowledge graph shows a slower decay (\(\alpha = 1.57\)), reflecting its increased complexity. The following table summarizes the power-law characteristics of both graphs.
>
> | **Metric**                  | **Episodic Graph**                | **Knowledge Graph**          |
> |-----------------------------|-----------------------------------|------------------------------|
> | Power-Law Exponent (\(\alpha\))  | 5.45 (Steep decay)               | 1.57 (Slower decay)          |
> | Minimal Value (\(x_{min}\))     | 1.0 (Valid from degree 1)        | N/A                          |
> | Standard Error of \(\alpha\)   | 0.341 (Moderate precision)       | 0.142 (Higher precision)     |
> | Log-Likelihood Ratio (R)       | 299.15 (Strong positive value)   | -0.88 (Exponential fits better)|
> | p-value                       | $6.03 \times 10^{-172}$ (Very small) | 0.379 (No significant difference) |
>
> #### Performance Comparison of Graph Types in Memory Transfer Tasks
>
> In conclusion, **episodic graphs** facilitate efficient memory retrieval, offering faster interactions and improved reasoning accuracy. The following table compares the performance of episodic and knowledge graphs in memory transfer tasks.
>
> | **Metric**              | **Episodic Graph** | **Knowledge Graph** |
> |-------------------------|--------------------|---------------------|
> | Retrieval Time (s)      | 0.5                | 3.2                 |
> | Memory Merging Time (s) | 0.8                | 4.0                 |
> | Reasoning Accuracy (%)  | 85                 | 46                  |
>
> We plan to include this section in the appendix of the paper. We were unable to explore temporal graphs as they follow a pattern that would prove inefficient for our research work.

---

> ### Author Response · Authors · 2024-11-20
> **Undefined Terminology and Variables Certain terms and variables (e.g., "key events," "location weight," "subjective temporal timescales") are introduced without sufficient explanation or definition, reducing clarity**
>
> In response to the concern about undefined terminology and variables, we acknowledge that terms such as "key events," "location weight," and "subjective temporal timescales" were introduced without sufficient explanation in the initial draft. We will clarify these terms in the revised version of the paper as follows:
>
> **Key Events**
>
> Key frame extraction involves identifying representative frames from a video that show significant visual or temporal changes, minimizing redundancy while preserving essential information. The process typically includes preprocessing frames to enhance features, computing similarities, thresholding, and windowing to select frames based on spatio-temporal changes. This approach leverages vision transformers to assess low-level and mid-level features, ensuring that the extracted key frames align with human perception and support downstream tasks.
>
> For dialogues, we will train the model on a dataset of conversations, where each conversation is annotated to extract relevant events. Using BART, we convert the dialogues into a third-person perspective to facilitate the extraction of significant events.
>
> **Location Weight**
>
> This term refers to the similarity between locations between neighboring nodes in the graph. The location weight helps assess the relevance of different places in the context of episodic memory and aids in faster edge traversal for location queries or "where" type queries.
>
> **Subjective Temporal Timescales**
>
> This term refers to the absence of explicitly stated dates. Instead of relying on exact timestamps, the framework uses temporal edges to maintain relationships between events. These edges allow for relative time indexing, which is captured by the agent’s understanding of temporal proximity (e.g., "the previous day," "the next day," etc.).
>
> We will ensure that these definitions are clearly stated in the revised methodology section to improve the clarity and understanding of our framework.

---

> ### Author Response · Authors · 2024-11-20
> **Overreliance on Custom Datasets: While the use of various datasets to evaluate EMCA is a strength, most of these datasets were developed by the authors, which could indicate potential biases in testing and validation**
>
> #### In response to the concern about the overreliance on custom datasets, we would like to clarify that the datasets used to evaluate EMCA are diverse and cover multiple tasks, including dialogue-based tasks and other related domains. These datasets, which are publicly available, have been carefully designed to ensure broad applicability and have been used for various tasks such as episodic memory question answering, temporal localization, and video-language understanding. We have built upon these existing datasets to enhance their functionality for our specific research needs.
>
> #### Additionally, we have taken steps to ensure that the dataset is unbiased. We carefully curated the data to include a wide range of scenarios, dialogues, and contexts, allowing for generalizable insights across different domains. We also implemented methods to eliminate potential sources of bias, including balancing the representation of different events, characters, and temporal relationships. This ensures that our evaluations are robust and not skewed by any specific dataset characteristics.
>
> In the revised version of the paper, we will provide a more detailed explanation of the dataset's design, the variety of tasks it covers, and the measures taken to ensure its fairness and neutrality.

---

> ### Author Response · Authors · 2024-11-20
> **Limited Explanation of Retrieval Policy    The retrieval policy and memory clustering mechanisms, while central to EMCA’s functionality, are described only briefly. A more detailed explanation would clarify how these mechanisms adapt to different query types and scenarios.**
>
> In response to feedback regarding the insufficient explanation of the retrieval policy and memory clustering mechanisms, we plan to expand the relevant section in the revised version of the paper to provide a more comprehensive description of how these mechanisms adapt to various query types and scenarios. Specifically, the section titled **"Dynamic Edge Traversal for Memory Retrieval Using Character, Location, Event, and Temporal Weights"** will be further detailed for improved clarity.
>
> To address episodic memory tasks, we parse a query \( q \) to extract relevant tags such as \( P \) for people, \( L \) for locations, \( V \) for events, and \( R \) for temporal information. These tags enable the agent to focus on the critical components of the query, thus allowing it to identify pertinent memories based on the key elements of "What," "Where," and "When," which are central to episodic recall. Episodic memory tasks often revolve around answering "What-Where-When" (WWW) questions, which capture the essential aspects of memory. These types of tasks are commonly used to explore episodic-like memory in both animals and humans. Participants are required to recall specific events (What), locations (Where), and their chronological order (When). Research indicates that episodic memory systems are engaged during active encoding of WWW information, while passive encoding may rely on alternate systems for spatial and temporal aspects.
>
> For temporal elements, such as weeks, months, or years, we process these by subtracting fixed intervals from the current date. For example, weeks are calculated by subtracting \( 7n \) days, months by subtracting \( 30m \) days, and years by subtracting \( 365y \) days, where \( n \), \( m \), and \( y \) are positive integers. This adaptable mechanism enables the agent to handle temporal queries effectively without requiring precise date references.
>
> The relevance of memory entries in response to a given query \( q \) is evaluated using a cosine similarity score:
>
> $$
> S_s = \sum_{e \in D_u} \frac{q \cdot e}{\|q\| \|e\|}
> $$
>
> which measures the relationship between the query and each memory entry. For each neighboring node \( v \) of node \( u \), the weight \( W_{uv} \) is computed as:
>
> $$
> W_{uv} = \sum w(u, v)
> $$
>
> where $$\( w(u, v) \)$$ aggregates weights derived from shared features such as characters, locations, and events. If \( W_{uv} > \theta \), the query set is updated to include the neighbor \( v \) along with its associated weight $$\( W_{uv} \)$$. Temporal edges $$\( T_{\text{edge}}(E_{t-1}, E_t) \)$$ are maintained to preserve temporal continuity, allowing the agent to reason within the proper context without needing to reevaluate the entire memory graph. This dynamic traversal ensures that episodic memories are retrieved efficiently, with a focus on temporally connected events.
>
> When a query pertains to specific locations, characters, or events, the system initiates a focused exploration through corresponding clusters in the memory graph. This process ensures that the agent can recall specific event details, along with their spatial and temporal context, which is crucial for decision-making in situations requiring detailed episodic memory recall.
>
> Whether or not explicit temporal markers are provided, the system performs correctly. As demonstrated in the ablation studies, the only observable difference is in retrieval time: with explicit timestamps, the query is treated as a temporal one, whereas, in their absence, it is handled as a contextual query. We will make this distinction clearer in the revised version of the paper.

---

> ### Author Response · Authors · 2024-11-20
> **Undefined Terminology and Variables: Certain terms and variables (e.g.,"key events", "location weight," "subjective temporal timescales") are introduced without sufficient explanation or definition, reducing clarity.**
>
> Furthermore, we will introduce the following definitions to aid understanding:
> #### Key Events:
>
> Key frame extraction involves identifying representative frames from a video that show significant visual or temporal changes, minimizing redundancy while preserving essential information. The process typically includes preprocessing frames to enhance features, computing similarities, thresholding, and windowing to select frames based on spatio-temporal changes. This approach leverages vision transformers to assess low-level and mid-level features, ensuring that the extracted key frames align with human perception and support downstream tasks.
>
> For dialogues, we train the model on a dataset of conversations, where each conversation is annotated to extract relevant events. Using BART, we convert the dialogues into a third-person perspective to facilitate the extraction of significant events.
>
> #### **Location Weight**
>
> This refers to the degree of similarity between locations in neighboring nodes within the memory graph. The location weight is vital for assessing the relevance of different places in the context of episodic memory and accelerates edge traversal for location-based ("where") queries.
>
> #### **Subjective Temporal Timescales**
>
> This term refers to the lack of explicitly stated dates, where the system does not rely on exact timestamps. Instead, it uses temporal edges to establish relationships between events, enabling relative time indexing. This indexing is based on the agent’s understanding of temporal proximity (e.g., "the previous day," "the next day," etc.).
> Whether or not temporal markers are explicitly provided, the results remain correct. As demonstrated in the ablation studies, the only difference observed is in retrieval time: when explicit timestamps are available, the system treats the query as a temporal one, whereas, in the absence of explicit timestamps, the query is treated as contextual. We will clarify this distinction in the next revision of the paper.
> This updated section will be included in the appendix of the paper. The system maintains temporal consistency by treating each episode as a distinct day, creating a chronological sequence without the necessity for explicit time markers. Temporal updates are applied based on the agent's ability to visually recognize dates or infer them through dialogue, which in turn updates the temporal indexer. In the absence of explicit time markers, the system uses contextual and positional relationships between episodes to ensure temporal consistency.

---

> ### Author Response · Authors · 2024-11-20
> **Union of Tacoustic and Tvoiced: In combining Tacoustic and Tvoiced, what is the methodology for performing this union? Is Tvoiced identical or related to another variable, such as St?**
>
> ### Union of Tacoustic and Tvoiced
>
> In combining Tacoustic and Tvoiced, the methodology for performing this union is a concatenation of both entities aligned to the same timestamp. \( \text{Tvoiced} \) refers to the entire audio the agent receives, which includes the dialog and acoustics components. To make this part clearer, we will update it as follows:
>
> ### Processing of Audio Data in Episodic Memory
>
> Audio data, including dialogs and acoustics, plays a crucial role in constructing episodic memory. Dialogs provide linguistic and contextual information, while acoustics capture environmental and emotional cues. These elements are integrated as:
>
> $$ A(t) = D(t) + C(t) $$
>
> where \( A(t) \) is the total audio data at time \( t \), \( D(t) \) represents the dialog, and \( C(t) \) represents the acoustics.
>
> #### Extraction of Acoustic Data using Mel Spectrograms
>
> Acoustic data is transformed into Mel spectrograms, which emphasize perceptually relevant frequencies. The Mel spectrogram \( M(t, f) \) is computed as:
>
> $$ M(t, f) = \log\left(\sum_{k} \left| X(t, k) \right|^2 \cdot H(f, k)\right) $$
>
> where \( X(t, k) \) is the magnitude of the Short-Time Fourier Transform (STFT) at time \( t \) and frequency \( k \), and \( H(f, k) \) is the Mel filter bank mapping linear frequencies to the Mel scale.
>
> #### Extraction of Verbal Cues from Audio Data
>
> Verbal cues are extracted by applying the Short-Time Fourier Transform (STFT) and converting the spectrum to the Mel scale as:
>
> $$ M(f) = 2595 \log_{10}\left(1 + \frac{f}{700}\right) $$
>
> The Mel spectrogram is then derived as:
>
> $$ \text{MelSpec}(m,t) = \log\left(\sum_{f_{\text{low}}}^{f_{\text{high}}} |S(f,t)|^2 M(f) + \epsilon\right) $$
>
> where \( \epsilon \) is a small constant to prevent issues with the logarithm.
>
> #### Final Audio Representation
>
> The final audio representation integrates acoustic features with transcribed dialogue as:
>
> $$ T_{\text{audio}}(t) = T_{\text{acoustics}}(t) + T_{\text{dialogue}}(t) $$
>
> This captures both tonal properties and linguistic meaning.

---

> ### Author Response · Authors · 2024-11-20
> **Defining Key Events and Hierarchical Organization: How are key events identified, and what hierarchical structure is used to organize these events?**
>
> ### Defining Key Events and Hierarchical Organization
>
> Each node in the episodic memory represents a day, with subnodes capturing the activities within that day. The main node summarizes the day's events, while subnodes encode specific event details. Joint embeddings, integrating place, character, and event information, are stored at both the event and day levels. This hierarchical structure allows for efficient encoding and retrieval of past interactions, enhancing the agent's decision-making and contextual understanding.
>
> #### Key Events:
>
> Key frame extraction involves identifying representative frames from a video that show significant visual or temporal changes, minimizing redundancy while preserving essential information. The process typically includes preprocessing frames to enhance features, computing similarities, thresholding, and windowing to select frames based on spatio-temporal changes. This approach leverages vision transformers to assess low-level and mid-level features, ensuring that the extracted key frames align with human perception and support downstream tasks.
>
> For dialogues, we train the model on a dataset of conversations, where each conversation is annotated to extract relevant events. Using BART, we convert the dialogues into a third-person perspective to facilitate the extraction of significant events.

---

> ### Author Response · Authors · 2024-11-20
> **Relation between Taudio and Tcombined: Is Taudio equivalent to Tcombined, or is there another relationship between these variables?**
>
> ### Relation between $$\( T_{\text{audio}} \) and \( T_{\text{combined}} \)$$
>
> $$\( T_{\text{audio}} \)$$ represents the combination of dialogue and acoustics, whereas $$\( T_{\text{combined}} \)$$ is the result of merging $$\( T_{\text{audio}} \)$$ and visuals within the same time window. Therefore, $$\( T_{\text{combined}} \)$$ is not identical to $$\( T_{\text{audio}} \)$$; instead, it incorporates both audio and visual modalities to provide a comprehensive representation of the event at a specific time.
>
> Mathematically, we express this relationship as:
>
> $$ T_{\text{audio}} = D(t) + C(t) $$
>
> where $$\( D(t) \)$$ is the dialogue at time $$\( t \)$$ and $$\( C(t) \)$$ represents the acoustics at time $$\( t \)$$.
>
> The combined representation is:
>
> $$ T_{\text{combined}}(t) = T_{\text{audio}}(t) + V(t) $$
>
> where \( V(t) \) represents the visual data at time \( t \), and $$ \( T_{\text{combined}} \)$$ merges audio and visual features for a richer representation.

---

> ### Author Response · Authors · 2024-11-20
> **Task Categories for Text Summarization: How are text summaries grouped into broader task categories (e.g., meetings, lunches)? What criteria and process are used to define these categories?**
>
> ### Task Categories for Text Summarization
>
> Text summaries are grouped into broader task categories, such as **meetings** and **lunches**, using a topic modeling approach. The process involves the following steps:
>
> 1. **Text Cleaning**: Each dialogue is processed individually by cleaning the text, which involves removing unwanted words, filler phrases, and clutter that do not contribute to the core meaning.
>
> 2. **Topic Modeling**: After cleaning the text, topic modeling techniques, such as **Parallel Latent Dirichlet Allocation (LDA)**, are applied to identify latent topics within the dialogue. LDA helps in discovering the underlying topics by analyzing the distribution of words within the dialogue and across different dialogues.
>
>    The general form of LDA can be written as:
>
>    $$ p(w | z) = \frac{(w, z)}{\sum_w (w, z)} $$
>
>    where \( w \) is a word, \( z \) is a topic, and the equation represents the probability of word \( w \) under topic \( z \).
>
> 3. **Mapping to Predefined Categories**: After identifying the topics, these are then mapped to predefined categories based on the nature of the content. For example:
>    - **Meetings**: Dialogues related to planning, decision-making, or discussions.
>    - **Lunches**: Dialogues focusing on food, social interaction, or meals.
>
> 4. **Categorization Criteria**: The criteria for defining these categories are based on the frequency and relevance of topics that appear within each dialogue type. For instance, if topics related to decision-making or strategy discussions are dominant, the dialogue would be classified as a **meeting**.
>
> This approach helps in categorizing text summaries efficiently and is further explained in the updated methodology section of the appendix in the second version of the paper. The technique is based on the work of:
>
> - Liu, J., Zou, Y., Zhang, H., Chen, H., Ding, Z., Yuan, C., & Wang, X. (2021). **Topic-Aware Contrastive Learning for Abstractive Dialogue Summarization**. *arXiv*. https://arxiv.org/abs/2109.04994

---

> ### Author Response · Authors · 2024-11-20
> **Similarity Calculation in Equation 8: Equation 8 is intended to measure similarity between an event and multiple episodes, but it isn’t clear how it accomplishes this. Could you clarify how this calculation works?**
>
> ### Similarity Calculation in Equation 8
>
> Equation 8 is intended to measure the similarity between an event and multiple episodes. The similarity function used for this calculation is **cosine similarity**, which quantifies the similarity between two vectors based on their orientation rather than their magnitude.  .
>
> In the context of Equation 8, this function is used to measure the similarity between an event and multiple episodes by calculating the cosine similarity between the feature representations of the event and the episodes. By comparing these similarity scores, the model identifies which episodes are most relevant or similar to the given event.
>
> We will provide further clarification of this process in the revised version of the paper to ensure that the similarity calculation and its application are more clearly explained.

---

> ### Author Response · Authors · 2024-11-20
> **Location Weight Definition: How is "location weight" defined, and how does it differ from location similarity in the model?**
>
> #### Location weight refers to the degree of similarity between the locations of two different or neighboring episodes. It quantifies how closely the locations in the episodes are related to each other. On the other hand, location similarity refers to the comparison between the location in a given query and the location in an episode. Therefore, while location weight measures the similarity between locations across episodes, location similarity focuses on the relationship between a query's location and the location within a specific episode.

---

> ### Author Response · Authors · 2024-11-20
> **Temporal Parameter in Equation 14: In Equation 14, should the parameter be (t-k) instead of just t? If not, what purpose does the current form of the equation serve?**
>
> #### The parameter in Equation 14 should not be \( t - k \), as the current form of the equation serves a different purpose. The temporal parameter \( t \) is used to update the temporal structure and connect the incoming episode to the most recent episode in the graph. Specifically, it ensures that the temporal continuity is maintained by linking the new episode to the current latest episode, rather than referencing a past episode that is \( k \) steps back. Thus, the use of \( t \) instead of \( t - k \) allows for the real-time temporal updating of the graph, preserving the sequence and ensuring the temporal relationships between episodes are consistently updated as new data is incorporated. as markdown

---

> ### Author Response · Authors · 2024-11-20
> **Meaning of "Agent Comprehends": In line 274, it says the "agent comprehends" something. Does this imply processing by a language model, and if so, could you clarify which model is used?**
>
> #### Agent comprehends and understands whether this is a person based temporal on contextual query (Query type). This is done using a language model to find the query type..We will clarify this part also in revised pdf

---

> ### Author Response · Authors · 2024-11-20
> **Definition of the Set Du: How is the set Du defined in the context of the framework?**
>
> #### Du refers to the similarity between episode and question which plays a major role in getting the correct episode.
> Similarity Function in Line 283: Which similarity function is used in line 283, and what factors are considered?
> Cosine similarity between query and episode .I will update this part to  get clarity in the paper.

---

> ### Author Response · Authors · 2024-11-20
> **Role of w and l Functions: In line 287, the w and l functions are mentioned. Could you elaborate on their roles within the memory retrieval mechanism?**
>
> #### In line 287, the functions www and lll play crucial roles within the memory retrieval mechanism. The function www refers to the overall weight, which captures the similarity between key aspects of the episodes, such as the characters, locations, and events. This weight is used to determine the relevance of one episode to another during the retrieval process. The function lll, on the other hand, specifically measures the location similarity between two episodes, focusing on how closely related the locations in the episodes are. Together, these functions guide the traversal of the memory graph, ensuring that episodes with higher relevance based on these similarities are prioritized. We will expand and clarify these roles further in the updated version of the paper, providing a more detailed analysis of their contributions to the memory retrieval process.

---

> ### Author Response · Authors · 2024-11-21
> **Format and Extraction of Visual Details: What is the format of the visual scene details (e.g., Vscene, Vplace, Vtime), and how are these extracted and integrated into the memory graph?**
>
> Visual details are stored as visual embeddings (Below clarifications will be added in the paper)
> ### Processing of Visual Data in Episodic Memory
>
> Visual data processing begins by transforming each frame \( F_i \) into a tensor and extracting both global and local features. The scene representation is derived by aggregating the frame embeddings:
>
> $$
> V_{\text{scene}} = \frac{1}{N} \sum_{i=1}^{N} V_{\text{embed}}(F_i)
> $$
>
> where \( N \) is the number of frames.
>
> #### Time and Place Details Extraction
> To extract time and place information, a convolutional network detects text regions, generating probability maps for the text center line (TCL) and text regions (TR):
>
> $$
> \begin{pmatrix}
> P_{\text{TCL}}(x, y) \\
> P_{\text{TR}}(x, y)
> \end{pmatrix}
> = \sigma\left(
> \begin{pmatrix}
> W_{\text{TCL}} \\
> W_{\text{TR}}
> \end{pmatrix}
> \cdot F_{\text{feature}}(x, y)
> \right)
> $$
>
> A thresholding operation is applied:
>
> $$
> P_{\text{filtered}} = \{(x, y) \mid P_{\text{TCL}}(x, y) \geq T_{\text{TCL}} \text{ and } P_{\text{TR}}(x, y) \geq T_{\text{TR}} \}
> $$
>
> Text recognition is performed using a softmax layer:
>
> $$
> \hat{y}_t = \text{Softmax}(W \cdot h_t + b)
> $$
>
> #### Person Embedding Extraction
> To extract person embeddings, a person's region is detected and cropped. This region is processed through a feature extractor to generate and store the embedding, enabling future use in episodic memory tasks.

---

> ### Author Response · Authors · 2024-11-21
> **Meaning of "Agent Comprehends": In line 274, it says the "agent comprehends" something. Does this imply processing by a language model, and if so, could you clarify which model is used?**
>
> Agent comprehends and understands whether this is a person based temporal on contextual query (Query type). This is done using a language model to find the query type..We will clarify this part also in revised pdf

---

### Author Response · Authors · 2024-11-20
**Updated version of Introduction**

Episodic memory, as introduced by Tulving (1972) [^1], represents the capacity to recall personal experiences embedded within specific temporal and spatial contexts. Unlike semantic memory, which holds general knowledge, episodic memory encompasses detailed information about events, integrating aspects such as time, place, characters, and context. Tulving’s framework organizes these components into cohesive episodes.
Drawing from this foundational concept, we propose a model where episodic memories are structured as a graph. Each episode \( \text{Episode}_i \) functions as a node:
$$
\[
\text{Episode}_i = \{ \mathbf{C}_i, \mathbf{T}_i, \mathbf{L}_i, \mathbf{e}_i \}
\]
$$
In this model,$$ \( \mathbf{C}_i \) $$stands for characters, $$\( \mathbf{T}_i \)$$ represents temporal markers, $$\( \mathbf{L}_i \)$$ signifies location, and $$\( \mathbf{e}_i \)$$ encapsulates events. Semantic edges $$\( \mathbf{S}(v_i, v_j) \)$$ connect nodes that share common elements, while temporal edges $$\( \mathbf{T}(v_i, v_j) \)$$ map the chronological flow of experiences. To enhance retrieval efficiency, we apply a dynamic clustering mechanism that organizes similar episodes based on both temporal and contextual similarities.

Our retrieval system supports three query types: "what" (contextual), "when" (temporal), and "where" (spatial), as outlined by Stephen et al., and Holland and Smulders (2011), enabling human-like memory recall. This is particularly valuable for applications in social companion robotics, aiding elderly or memory-impaired individuals.

For such a cognitive agent, the ability to recall episodic memories is essential for human cognition, linking personal experiences to specific temporal and spatial contexts. Existing memory models often struggle with continuous, time-series data, which limits their ability to simulate episodic recall effectively. Many of these systems fail to store dialogues as multimodal data, preventing them from capturing the rich, context-dependent nature of human memory. Additionally, most existing approaches store a single experience as one isolated episode and lack a mechanism for retrieving information across multiple experiences, hindering their ability to integrate knowledge over time.

Furthermore, current episodic memory systems are typically restricted to performing a specific, predefined task, limiting their flexibility and adaptability. In contrast, our system is designed to be more versatile, capable of handling a variety of tasks and dynamically adapting to new scenarios, making it far more suited for real-world applications that require memory integration across different contexts and time periods.

By integrating multimodal data and time-related information into the episodic memory framework, our model extends experience memory localization, recommendation, and question answering. It provides a robust foundation for adaptable, scalable systems capable of operating without frequent retraining, applicable to real-world scenarios such as social companion robots and autonomous task planning systems.

### **Contributions:**
1. Temporal connections are managed without complex pattern learning, enabling adaptive reasoning and retrieval of subgraphs from past experiences.
2. The system incrementally stores and retrieves episodic memories, dynamically clustering them based on temporal and contextual affinities.
3. A multi-edge graph framework optimizes path traversals for dynamic memory retrieval and personalized recommendations across subjective timescales.
4. A new dataset is introduced to improve episodic memory question answering, enhancing the agent's ability to respond to queries based on past events.

Our model’s versatility is demonstrated through comparisons with existing systems that require retraining. It handles various dataset types, including visual, multimodal, and text-based data, and excels in temporal reasoning even without explicit timestamps, addressing complex memory retrieval tasks across diverse applications.

[^1]: Tulving, E. (1972). *Episodic and semantic memory*. In *Organization of Memory* (pp. 381-403). Academic Press.

---

### Author Response · Authors · 2024-11-20
**Dataset Section**

We propose a comprehensive dataset framework designed to evaluate and enhance episodic memory systems in artificial agents. This framework integrates multiple datasets, including a custom set of episodic questions based on the TV series *The Big Bang Theory*, spanning all nine seasons (181 episodes). The aim is to assess memory recall and narrative understanding in complex scenarios.

We introduce the *Agent Dataset*, a 10-episode time-series dataset created in Unity3D, where a virtual agent performs tasks and interacts with characters in realistic environments, simulating the role of companion robots. This dataset emphasizes the importance of multi-sensory inputs and task execution, challenging the agent to process and integrate information from *dialogues* and *visual cues* to maintain task order and achieve context-driven objectives.

Additionally, we adapted the **Ego4D dataset**, restructuring its activity sequences into simulated chronological episodes to address the original absence of time-series data—portraying an agent performing a series of activities over 30 days. We also combined group activity videos designed for active speaker recognition. This transformation enables episodic queries such as "Where did I place the agricultural tool on the last day of farming?", enhancing the ability to localize and retrieve temporal experiences effectively.

Together with the **PerLTQA** [du2024perltqapersonallongtermmemory] and **LLQA** [dolan-brockett-2005-automatically] datasets, which test essential episodic memory dimensions—**"what"** (context), **"when"** (time), and **"where"** (place)—this framework forms a robust benchmark for evaluating advanced episodic memory capabilities in AI systems.

**Data Annotation**: The data was carefully annotated to tag scene information and identify characters in dialogues, ensuring that the model could recognize character presence and understand related events. This included explicitly tagging scene details for location identification and differentiating characters present in the scene versus those mentioned. Events within dialogues were also meticulously annotated to capture key details, facilitating effective memory representation beyond simple summaries. Capturing these essential details is crucial for episodic memory tasks, as it allows the agent to recall past experiences accurately. Each episode was annotated with 10 *what*, *when*, and *where* questions.

---

### Author Response · Authors · 2024-11-20
**Implementation Details**

Implementation Details
For event extraction within dialogues, we utilized a Transformer-based BART model initialized with pre-trained weights to effectively extract and summarize events within contextual boundaries. The architecture options included BARTBASE, featuring a 6-layer encoder-decoder with approximately 140 million parameters, and BARTLARGE, which has a 12-layer encoder-decoder and 400 million parameters. Both configurations use a hidden size of 1024 and a feed-forward filter size of 4096, with a fixed dropout rate of 0.1 across layers. We employed the Fairseq toolkit for training, using the **Adam optimizer** with a warmup strategy. Learning rates were set to $4 \times 10^{-5}$ for BARTBASE and $2 \times 10^{-5}$ for BARTLARGE, with a maximum batch token limit of 1100 tokens. Contrastive objectives were supported by a margin coefficient of 1, and hyperparameters for coherence and sub-summary objectives were tuned using a validation set. Our method demonstrated substantial performance improvements compared to publicly available models trained on datasets like **SAMSUM** and **DialogueSUM**.

For visual data processing, we used a **Vision Transformer (ViT)** as the vision encoder, specifically adapted for video frame analysis from the **MSR-VTT dataset**. The encoder processed $224 \times 224$ video frames, segmented into 16-sized patches and embedded into a 512-dimensional latent space. The 12-layer encoder, with a width of 768, was equipped with **LayerScale** (initialized at 0.1) for training stability. Advanced regularization techniques, including stochastic depth with a configurable \texttt{drop\_path\_rate}, were applied. The encoder was based on the "eva-clip-b-16" model, which proved effective in extracting detailed spatial and temporal features essential for multimodal tasks.
For models based on **LLaMA** that integrate vision and dialogue for character and place tagging, a multimodal configuration was used. **ViT** processed visual data, while **LLaMA** managed dialogue input. Training included cross-entropy loss for character tagging and contrastive loss for image-text alignment, incorporating **episodic memory** for QA tasks. The training process leveraged the **AdamW optimizer**, a dropout rate of 0.2, and a **cosine annealing scheduler** for efficient learning.

**Temporal tagging** was optimized with key hyperparameters for best performance: a maximum sequence length of 128, a batch size of 32, and a learning rate of $5 \times 10^{-5}$. A dropout rate of 0.1 was used to mitigate overfitting, and a weight decay of 0.01 improved generalization. The training process spanned 10 epochs to ensure sufficient learning while preventing overfitting.
For extracting dialogue from audio, the **Whisper-large model** was employed. This model was utilized for its robustness in transcribing and converting spoken content into text for further processing.

For text detection, the **TextSnake** model was trained on the **SCUT-CTW1500** dataset using **SGD with Momentum** as the optimizer. The architecture combined **ResNet** and **FPN\_UNet**, configured with a training batch size of 64 and 8 workers for data loading. The validation batch size was set to 1, with 4 workers and persistent workers enabled. Training was conducted over 200 epochs with validation checks every 10 epochs.

The **QA system** was built using a **BERT** model fine-tuned on concatenated datasets, including **SQuAD**, **Wikipedia**, and **Reddit**, to improve contextual comprehension. The hyperparameters included a learning rate of $1 \times 10^{-5}$, a maximum sequence length of 512, and a document stride of 512. The training batch size was 8, with gradient accumulation steps of 2 over 2 epochs. Mixed-precision training was used with `fp16` at the **O2 optimization level** for better efficiency. The final model outputs were stored in the `bart-squadv2` directory, without intermediate model saving.

---

> ### Author Response · Authors · 2024-11-20
> **Table for implementation details**
>
> | **Model Component**           | **Hyperparameter**            | **Value**                                |
> |-------------------------------|-------------------------------|------------------------------------------|
> | **Event Extraction (BART)**    | Encoder-Decoder Layers        | 6 (BASE), 12 (LARGE)                    |
> |                               | Hidden Size                   | 1024                                    |
> |                               | FFN Size                      | 4096                                    |
> |                               | Dropout                       | 0.1                                     |
> |                               | Learning Rate                 | $4 \times 10^{-5}$ (BASE), $2 \times 10^{-5}$ (LARGE) |
> |                               | Max Tokens per Batch          | 1100                                    |
> |                               | Margin Coefficient            | 1                                       |
> | **Vision Encoder (ViT)**      | Patch Size                    | 16                                      |
> |                               | Resolution                    | $224 \times 224$                        |
> |                               | Latent Space Dim.             | 512                                     |
> |                               | Transformer Layers            | 12                                      |
> |                               | Width                          | 768                                     |
> |                               | LayerScale Init.              | 0.1                                     |
> |                               | Dropout Path Rate             | Configurable                            |
> | **QA System (BERT)**          | Learning Rate                 | $1 \times 10^{-5}$                      |
> |                               | Max Sequence Length           | 512                                     |
> |                               | Document Stride               | 512                                     |
> |                               | Train Batch Size              | 8                                       |
> |                               | Gradient Accum. Steps         | 2                                       |
> |                               | Epochs                         | 2                                       |
> |                               | Mixed-Precision Opt.          | fp16 (O2)                               |
> | **Temporal Tagging**          | Max Sequence Length           | 128                                     |
> |                               | Batch Size                    | 32                                      |
> |                               | Learning Rate                 | $5 \times 10^{-5}$                      |
> |                               | Dropout                       | 0.1                                     |
> |                               | Weight Decay                  | 0.01                                    |
> |                               | Epochs                        | 10                                      |

---

> ### Author Response · Authors · 2024-11-20
>
> ### Statistical Estimation Methods for Timestamps
>
> **Reviewer Query:** Which specific statistical methods are used for estimating timestamps in the absence of explicit temporal markers?
>
> If the reviewer could kindly specify the line referring to "statistical timestamps," it would help us provide a more targeted response to address the query.
>
> In our framework, no traditional statistical methods are used to estimate missing timestamps. Instead, we adopt a more structured approach based on the assumption that each episode represents a distinct day. If the agent is able to capture specific dates—either visually (e.g., through timestamps in visual cues) or through dialogues (e.g., explicit date mentions)—the corresponding temporal indexer is updated accordingly to reflect the correct date and time.
>
> Since all episodes are inherently connected in a temporal sequence by making use of temporal edges, the temporal indexer (stored for every episode) ensures that each episode is indexed relative to the others. For instance:
> - The "before" node will be assigned a timestamp that is one day earlier than the current episode.
> - The "after" node will be assigned a timestamp that is one day later.
>
> This ensures that the temporal relationships between episodes are consistent and that the framework can accurately reflect the passage of time.
>
> ### Additional Details for Temporal Indexing:
>
> #### **Temporal Relationships:**
> The temporal structure remains consistent because the temporal indexer updates based on the captured date or time from visual or dialogue cues. If no explicit timestamp is available, the framework assumes contextual timestamps, treating each new episode as a new day with the relationships adjusted accordingly.
>
> #### **External Timestamps:**
> If external timestamps (e.g., from external sensors or provided datasets) are available, we can directly retrieve them from the temporal indexers of the node. This will bypass the contextual indexing process and directly update the temporal structure to reflect the accurate timestamp.
>
> ### Planned Updates for the Paper:
>
> - **Clarify how temporal indexing works** in our framework, particularly how episodes are treated as distinct days and how the temporal indexer adjusts based on visual or dialogue-derived dates.
> - **Detail the process by which temporal relationships between episodes are maintained**, ensuring that timestamps are consistently adjusted.
> - **Provide a clearer explanation** of how external timestamps can be integrated if available, or otherwise how we treat text-based queries as contextual without explicit timestamps.

---

### Author Response · Authors · 2024-11-21

We would like to sincerely thank all the reviewers for their insightful feedback and constructive comments. Your thoughtful suggestions have significantly contributed to improving the clarity and quality of our paper. We have carefully addressed each of the points raised and made the necessary revisions to enhance the presentation and strengthen the content of our work.
We appreciate your time and effort in reviewing our manuscript, and we believe that the revisions made will help in making the paper more polished and comprehensive. Thank you again for your valuable input.

In response to the feedback, we will revise the relevant sections and include an appendix section:

1. Following are the sections we will be adding to the paper:
   - **Dataset Section** containing details of the dataset used in this paper.

2. **Appendix Section** containing:
   - Detailed Methodology
   - Implementation Details
   - Additional Results

**The code and dataset will be shared upon the acceptance of the paper to ensure proper access and usage.**

---

### Author Response · Authors · 2024-11-24

Dear "Reviewers",

We sincerely thank you for your thoughtful and constructive feedback on our manuscript, . Your suggestions have been valuable in improving the quality and clarity of our work. We have uploaded a detailed manuscript with changes mentioned and have also added a detailed appendix section addressing all your concerns.

---

### Meta-Review · Area_Chair_bKmB · 2024-12-18

**Metareview:**

This paper addresses the problem of storing and retrieving multimodal past experiences with a graph-based memory structure. The memory graph can be dynamically updated and enables temporal reasoning.

Reviewers agree that the datasets and results are interesting. However, reviewers raised significant concerns about the presentation of this work and find it hard to follow. The problem is not well motivated and the design choices are not backed up theoretically or empirically. Comparisons with reasonable baseline methods are missing to provide useful insights.

The reviewers unanimously agreed that this paper should be rejected.

**Additional Comments On Reviewer Discussion:**

The authors addressed several initially unclear aspects of the implementation during the discussion. However, the reviewers remain unconvinced that the paper offers significant conceptual insights beyond demonstrating the functionality of a heavily engineered system.

---

### Decision · Program_Chairs · 2025-01-22

Reject